# Fairness-Aware Estimation of Graphical Models

**Zhuoping Zhou,**[*] **Davoud Ataee Tarzanagh,**[*] **Bojian Hou,**[*] **Qi Long,**[†] **Li Shen**[†]
University of Pennsylvania
{zhuopinz@sas., tarzanaq@}upenn.edu
{bojian.hou, qlong, li.shen}@pennmedicine.upenn.edu

## Abstract

This paper examines the issue of fairness in the estimation of graphical models (GMs), particularly Gaussian, Covariance, and Ising models. These models play a vital role in understanding complex relationships in high-dimensional data. However, standard GMs can result in biased outcomes, especially when the underlying data involves sensitive characteristics or protected groups. To address this, we introduce a comprehensive framework designed to reduce bias in the estimation of GMs related to protected attributes. Our approach involves the integration of the *pairwise graph disparity error* and a tailored loss function into a *nonsmooth multi-objective optimization* problem, striving to achieve fairness across different sensitive groups while maintaining the effectiveness of the GMs. Experimental evaluations on synthetic and real-world datasets demonstrate that our framework effectively mitigates bias without undermining GMs' performance.

## 1 Introduction

Graphical models (GMs) are probabilistic models that use graphs to represent dependencies between random variables [34]. They are essential in domains such as gene expression [91], social networks [17], computer vision [33], and recommendation systems [8]. The capacity of GMs to handle complex dependencies makes them crucial across various data-intensive disciplines. Therefore, as our society's reliance on machine learning grows, ensuring the fairness of these models becomes increasingly paramount; see Section 1.1 for further discussions. While significant research has addressed fairness in supervised learning [29], the domain of unsupervised learning, particularly in the estimation of GMs, remains less explored.

We address the fair estimation of sparse GMs where the number of variables $P$ is much larger than the number of observations $N$ [22, 16, 43]. We focus on three types of GMs:

I. **Gaussian Graphical Model:** Rows $\mathbf{X}_{1:}, \ldots, \mathbf{X}_{N:}$ in the data matrix $\mathbf{X} \in \mathbb{R}^{N \times P}$ are i.i.d. from a multivariate Gaussian distribution $\mathcal{N}(\mathbf{0}, \boldsymbol{\Sigma})$. The *conditional independence* graph is determined by the sparsity of the inverse covariance matrix $\boldsymbol{\Sigma}^{-1}$, where $(\boldsymbol{\Sigma}^{-1})_{jj'} = 0$ indicates conditional independence between the $j$th and $j'$th variables.

II. **Gaussian Covariance Graph Model:** Rows $\mathbf{X}_{1:}, \ldots, \mathbf{X}_{N:}$ are i.i.d. from $\mathcal{N}(\mathbf{0}, \boldsymbol{\Sigma})$. The *marginal independence* graph is determined by the sparsity of the covariance matrix $\boldsymbol{\Sigma}$, where $\Sigma_{jj'} = 0$ indicates marginal independence between the $j$th and $j'$th variables.

III. **Binary Ising Graphical Model:** Rows $\mathbf{X}_{1:}, \ldots, \mathbf{X}_{N:}$ are binary vectors and i.i.d. with

$$p(\mathbf{x}; \boldsymbol{\Theta}) = (Z(\boldsymbol{\Theta}))^{-1} \exp \Big( \sum_{j=1}^{P} \theta_{jj} x_j + \sum_{1 \leq j < j' \leq P} \theta_{jj'} x_j x_{j'} \Big). \tag{1}$$

Here, $\boldsymbol{\Theta}$ is a symmetric matrix, and $Z(\boldsymbol{\Theta})$ normalizes the density. $\theta_{jj'} = 0$ indicates conditional independence between the $j$th and $j'$th variables. The sparsity pattern of $\boldsymbol{\Theta}$ reflects the conditional independence graph.

---

[*]Equal contribution
[†]Corresponding authors

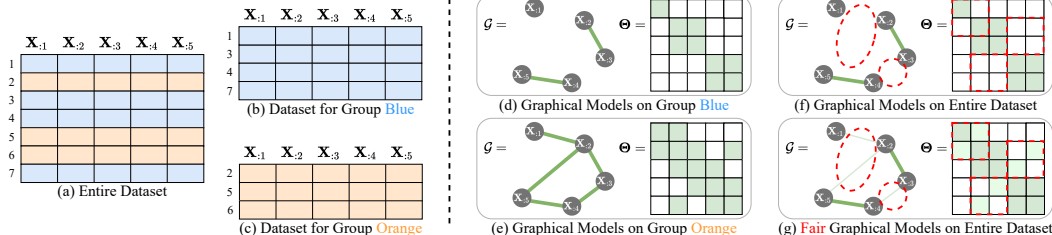

Figure 1: Illustration of a GM and its fair variant. (a) displays the entire dataset, split into Group Blue (b) and Group Orange (c). (d) and (e) show GMs for each group, detailing the relationships between variables. (f) uses a GM for the entire dataset. The fair model in (g) adjusts these relationships to ensure equitable representation and minimize biases in subgroup analysis.

In a data matrix $\mathbf{X} \in \mathbb{R}^{N \times P}$, each column corresponds to a node in a graph $\mathcal{G} = (\mathcal{V}, \mathcal{E})$, where $\mathcal{V} = \{1, 2, \ldots, P\}$ are vertices and $\mathcal{E} \subseteq \mathcal{V} \times \mathcal{V}$ are edges. Column $\mathbf{X}_{:i}$ ($i \in \{1, 2, \ldots, P\}$) is a vector of length $N$, representing the observations for the $i$-th variable across all $N$ samples. The graph $\mathcal{G}$, represented by the symmetric matrix $\mathbf{\Theta}$, has nonzero entries indicating edges and reflects the graph's independence properties. To obtain a sparse and interpretable graph estimate, we consider

$$\underset{\mathbf{\Theta}}{\text{minimize}} \; \mathcal{L}(\mathbf{\Theta}; \mathbf{X}) + \lambda \|\mathbf{\Theta}\|_1 \quad \text{subj. to} \quad \mathbf{\Theta} \in \mathcal{M}. \tag{2}$$

Here, $\mathcal{L}$ is a loss function; $\lambda \| \cdot \|_1$ is the $\ell_1$-norm regularization with parameter $\lambda > 0$; and $\mathcal{M}$ is a convex constraint subset of $\mathbb{R}^{P \times P}$. For example, in a Gaussian GM, $\mathcal{L}(\mathbf{\Theta}; \mathbf{X}) = -\log\det(\mathbf{\Theta}) + \text{trace}(\mathbf{S}\mathbf{\Theta})$, where $\mathbf{S} = n^{-1} \sum_{i=1}^{n} \mathbf{X}_{i:}^{\top} \mathbf{X}_{i:}$, and $\mathcal{M}$ is the set of $P \times P$ positive definite matrices.

## 1.1 Motivation

Our motivations for obtaining a fair GM estimation are summarized as follows. *i) Equitable Representation:* Standard group-specific GM models may improve accuracy for targeted groups but do not ensure fairness and can reinforce biases present in the data [48]. A unified approach considering the entire dataset is essential for mitigating biases and promoting fairness across all groups. *ii) Legal and Ethical Compliance:* Ethical and legal considerations [12] require explicit consent for processing sensitive attributes in model selection. Thus, constructing a fair estimation approach that adheres to fairness practices, uses data with consent, and excludes sensitive attribute information during deployment ensures privacy and legal compliance. *iii) Generalization across Groups:* A unified fair GM captures differences across groups without segregating the model, enhancing generalizability and preventing overfitting to a specific group [30], a risk in training separate models for each group.

For further discussion, we compare a GM with its proposed Fair variant, as illustrated in Figure 1. Panel (a) shows the entire dataset, divided into Group Blue and Group Orange in panels (b) and (c). Panels (d) and (e) detail the GM for each group, highlighting variable relationships. Panel (f) demonstrates a conventional GM applied to the full dataset, revealing a bias towards Group Blue. Panel (g) introduces a Fair GM, including modifications (red dashed lines) to reduce bias and ensure balanced representation. These adjustments correct relationships within the model, promoting fairness by preventing disproportionate favor towards any group. This illustration highlights the bias challenge in GMs and the steps Fair GMs take to ensure fair and equal modeling outcomes across groups.

## 1.2 Contributions

Our contributions are summarized as follows:

- ⋄ We propose a framework to mitigate bias in Gaussian, Covariance, and Ising models related to protected attributes. This is achieved by incorporating *pairwise graph disparity error* and a tailored loss function into a *nonsmooth multi-objective optimization* problem, striving to achieve fairness across different sensitive groups while preserving GMs performance.

- ⋄ We develop a proximal gradient method with non-asymptotic convergence guarantees for nonsmooth multi-objective optimization, applicable to Gaussian, Covariance, and Ising models (Theorems 6–8). To our knowledge, this is the first work providing a multi-objective proximal gradient method for GM estimation, in contrast to existing single-objective GM methods [3, 87, 13].

- ⋄ We provide extensive experiments to validate the effectiveness of our GM framework in mitigating bias while maintaining model performance on synthetic data, the Credit Dataset,

the Cancer Genome Atlas Dataset, Alzheimer's Disease Neuroimaging Initiative (ADNI), and the binary LFM-1b Dataset for recommender systems[3].

## 2 Related Work

**Estimation of Graphical Models.** The estimation of network structures from high-dimensional data [89, 51, 1, 84, 19, 84] is a well-explored domain with significant applications in biomedical and social sciences [44, 59, 25]. Given the challenge of parameter estimation with limited samples, sparsity is imposed via regularization, commonly through an $\ell_1$ penalty to encourage sparse network structures [22, 37, 25]. However, these approaches may overlook the complexity of real-world networks, which often have varying structures across scales, including densely connected subgraphs or communities [13, 26, 23]. Recent work extends beyond simple sparsity to estimate hidden communities within networks, reflecting homogeneity within and heterogeneity between communities [47]. This includes inferring connectivity and performing graph estimation when community information is known, as well as considering these tasks in the context of heterogeneous observations [42, 80, 24].

**Fairness.** Fairness research in machine learning has predominantly focused on supervised methods [11, 4, 15, 39, 92, 79, 28]. Our work broadens this scope to unsupervised learning, incorporating insights from [65, 77, 56, 9, 10]. Notably, [41] has developed algorithms for fair clustering using the Laplacian matrix. Our approach diverges by not presupposing any graph and Laplacian structures. The most relevant works to this study are [78, 93, 52, 53]. Specifically, [78] initiated the learning of fair GMs using an $\ell_1$-regularized pseudo-likelihood method for joint GMs estimation and fair community detection. [93, 94] proposed a fair spectral clustering model that integrates graph construction, fair spectral embedding, and discretization into a single objective function. Unlike these models, which assume community structures, our study formulates fair GMs without such assumption. Concurrently with this work, [52] proposed a regularization method for fair Gaussian GMs assuming the availability of node attributes. Their methodology significantly differs from ours, as we focus on developing three classes of fair GMs (Gaussian, Covariance, and Ising models) for imbalanced groups *without node attributes*, aiming to *automatically* ensure fairness through non-smooth multi-objective optimization.

## 3 Fair Estimation of Graphical Models

**Notation.** $\mathbb{R}^d$ denotes the $d$-dimensional real space, and $\mathbb{R}^d_+$ and $\mathbb{R}^d_{++}$ its positive and negative orthants. Vectors and matrices are in bold lowercase and uppercase letters (e.g., $\mathbf{a}$, $\mathbf{A}$), with elements $a_i$ and $a_{ij}$. Rows and columns of $\mathbf{A}$ are $\mathbf{A}_{i:}$ and $\mathbf{A}_{:j}$, respectively. For symmetric $\mathbf{A}$, $\mathbf{A} \succ 0$ and $\mathbf{A} \succeq 0$ denote positive definiteness and semi-definiteness. $\Lambda_i(\mathbf{A})$ is the $i$th smallest eigenvalue of $\mathbf{A}$. The matrix norms are defined as $\|\mathbf{A}\|_1 = \sum_{ij} |a_{ij}|$ and $\|\mathbf{A}\|_F = (\sum_{ij} |a_{ij}|^2)^{1/2}$. For any positive integer $n$, $[n] := \{1, \ldots, n\}$. Any notation is defined upon its first use and summarized in Table 3.

### 3.1 Graph Disparity Error

To evaluate the effects of joint GMs learning on different groups, we compare models trained on group-specific data with those trained on a combined dataset. Let a dataset $\mathbf{X}$ be divided into $K$ sensitive groups, with the data for group $k \in [K]$ represented as $\mathbf{X}_k \in \mathbb{R}^{N_k \times P}$, where $N_k$ is the sample size for group $k$, and $N = \sum_{k=1}^n N_k$. The performance of a GM, denoted by $\boldsymbol{\Theta}$, for group $k$ is measured by the loss function $\mathcal{L}(\boldsymbol{\Theta}; \mathbf{X}_k)$. Our goal is to find a global model $\boldsymbol{\Theta}^*$ that minimizes performance discrepancies across groups. We define graph disparity error to quantify fairness:

**Definition 1** (**Graph Disparity Error**). *Given a dataset* $\mathbf{X} \in \mathbb{R}^{N \times P}$ *with $K$ sensitive groups, where* $\mathbf{X}_k$ *represents the data for group $k \in [K]$, let*

$$\boldsymbol{\Theta}_k^* \in \underset{\boldsymbol{\Theta}_k \in \mathcal{M}}{\arg\min} \ \mathcal{L}(\boldsymbol{\Theta}_k; \mathbf{X}_k) + \lambda \|\boldsymbol{\Theta}_k\|_1. \tag{3}$$

*The graph disparity error for group $k$ is then:*

$$\mathcal{E}_k(\boldsymbol{\Theta}) := \mathcal{L}(\boldsymbol{\Theta}; \mathbf{X}_k) - \mathcal{L}(\boldsymbol{\Theta}_k^*; \mathbf{X}_k), \quad 1 \le k \le K. \tag{4}$$

This measures the loss difference between a global graph matrix $\boldsymbol{\Theta}$ and the optimal local graph matrix $\boldsymbol{\Theta}_k^*$ for each group's data $\mathbf{X}_k$. A fair GM, under Definition 1, seeks to balance $\mathcal{E}_k$ across all groups.

---

[3]Code is available at https://github.com/PennShenLab/Fair_GMs

---

**Algorithm 1** Fair Estimation of GMs (Fair GMs)

---

**Require:** Data Matrix $\boldsymbol{X} = \boldsymbol{X}_1 \cup \boldsymbol{X}_2 \cup \cdots \cup \boldsymbol{X}_K$; Parameters $\lambda > 0$, $\epsilon > 0$, $T > 0$, and $\ell > L$.

**S1.** Get local graph estimates $\{\boldsymbol{\Theta}_k^*\}_{k=1}^K$ using (3), and initialize global graph estimate $\boldsymbol{\Theta}^{(0)}$.
**S2. For** $t = 1$ **to** $T - 1$ **do**:
$\quad\quad \boldsymbol{\Theta}^{(t+1)} \leftarrow \mathbf{P}_\ell(\boldsymbol{\Theta}^{(t)})$, where $\mathbf{P}_\ell$ is obtained by solving Subproblem (9).

**Output**: Fair global graph estimate $\boldsymbol{\Theta}^{(t+1)}$.

---

**Definition 2** (**Fair GM**). *A GM with graph matrix $\boldsymbol{\Theta}^*$ is called fair if the graph disparity errors among different groups are equal, i.e.,*

$$\mathcal{E}_1(\boldsymbol{\Theta}^*) = \mathcal{E}_2(\boldsymbol{\Theta}^*) = \cdots = \mathcal{E}_K(\boldsymbol{\Theta}^*). \tag{5}$$

To address the imbalance in graph disparity error among all groups, we introduce the idea of pairwise graph disparity error, which quantifies the variation in graph disparity between different groups.

**Definition 3** (**Pairwise Graph Disparity Error**). *Let $\phi : \mathbb{R} \to \mathbb{R}_+$ be a penalty function such as $\phi(x) = \exp(x)$ or $\phi(x) = \frac{1}{2}x^2$. The pairwise graph disparity error for the group $k$ is defined as*

$$\Delta_k(\boldsymbol{\Theta}) := \sum_{s \in [K], s \neq k} \phi(\mathcal{E}_k(\boldsymbol{\Theta}) - \mathcal{E}_s(\boldsymbol{\Theta})). \tag{6}$$

The motivation for Definition 3 follows from the work of [35, 65, 95] in PCA and CCA. In our convergence analysis, we focus on smooth functions $\phi$, such as squared or exponential functions, while nonsmooth choices, such as $\phi(x) = |x|$, can be explored in the experimental evaluations.

### 3.2 Multi-Objective Optimization for Fair GMs

This section introduces a framework designed to mitigate bias in GMs (including Gaussian, Covariance, and Ising) related to protected attributes by incorporating pairwise graph disparity error into a nonsmooth multi-objective optimization problem. Smooth multi-objective optimization tackles fairness challenges in unsupervised learning [35, 95], proving particularly useful when decision-making involves multiple conflicting objectives.

We use *non-smooth multi-objective optimization* to balance two key factors: the loss in GMs and the pairwise graph disparity errors. To achieve this, let

$$f_1(\boldsymbol{\Theta}) = \mathcal{L}(\boldsymbol{\Theta}; \mathbf{X}), \qquad f_k(\boldsymbol{\Theta}) = \Delta_{k-1}(\boldsymbol{\Theta}), \qquad \text{for } 2 \leq k \leq K + 1, \tag{7a}$$

$$F_k(\boldsymbol{\Theta}) = f_k(\boldsymbol{\Theta}) + g(\boldsymbol{\Theta}), \qquad\qquad \text{for } 1 \leq k \leq M := K + 1, \tag{7b}$$

where $g(\boldsymbol{\Theta}) := \lambda \|\boldsymbol{\Theta}\|_1$ for some $\lambda > 0$.

Consequently, we propose the following multi-objective optimization problem for Fair GMs:

$$\underset{\boldsymbol{\Theta}}{\text{minimize}} \ \mathbf{F}(\boldsymbol{\Theta}) := [F_1(\boldsymbol{\Theta}), \ldots, F_M(\boldsymbol{\Theta})] \qquad \text{subj. to } \boldsymbol{\Theta} \in \mathcal{M}. \tag{8}$$

Here, $\mathcal{M}$ is a convex constraint subset of $\mathbb{R}^{P \times P}$ and $\mathbf{F} : \Omega \to \mathbb{R}^M$ is a multi-objective function.

**Assumption A.** *For some $L > 0$, all $\boldsymbol{\Theta}, \boldsymbol{\Phi} \in \mathcal{M}$, $k \in [M]$, $\|\nabla f_k(\boldsymbol{\Phi}) - \nabla f_k(\boldsymbol{\Theta})\|_F \leq L \|\boldsymbol{\Phi} - \boldsymbol{\Theta}\|_F$.*

Note that Assumption A holds for smooth $\phi$ functions such as squared or exponential, as specified in Definition 6, and when $\mathcal{L}$ is a smooth loss function. We demonstrate in Appendix C that this assumption holds for the Gaussian, Covariance, and Ising models studied in this work. To proceed, we provide the following definitions; see [20, 75, 73] for more details.

**Definition 4** (Pareto Optimality). *In Problem (8), a solution $\boldsymbol{\Theta}^* \in \mathcal{M}$ is Pareto optimal if there is no $\boldsymbol{\Theta} \in \mathcal{M}$ such that $\mathbf{F}(\boldsymbol{\Theta}) \preceq \mathbf{F}(\boldsymbol{\Theta}^*)$ and $\mathbf{F}(\boldsymbol{\Theta}) \neq \mathbf{F}(\boldsymbol{\Theta}^*)$. It is weakly Pareto optimal if there is no $\boldsymbol{\Theta} \in \mathcal{M}$ such that $\mathbf{F}(\boldsymbol{\Theta}) \prec \mathbf{F}(\boldsymbol{\Theta}^*)$.*

**Definition 5** (Pareto Stationary). *We define a point $\bar{\boldsymbol{\Theta}} \in \mathbb{R}^{P \times P}$ as Pareto stationary (or critical) if it satisfies the following condition:*

$$\max_{k \in [M]} F_k'(\bar{\boldsymbol{\Theta}}; \mathbf{D}) := \lim_{\alpha \to 0} \frac{F_k(\bar{\boldsymbol{\Theta}} + \alpha \mathbf{D}) - F_k(\bar{\boldsymbol{\Theta}})}{\alpha} \geq 0 \quad \text{for all} \quad \mathbf{D} \in \mathbb{R}^{P \times P}.$$

To solve Problem (8), we use the proximal gradient method and establish its convergence to a Pareto stationary point for the nonsmooth Problem (8). The procedure for our fairness-aware GMs (Fair GMs) is detailed in Algorithm 1. Given local graph estimates $\{\boldsymbol{\Theta}_k^*\}_{k=1}^K$ obtained in **S1.**, and $\ell > L$, where $L$ is a Lipschitz constant defined in Assumption A, the update of the global fair graph estimate $\boldsymbol{\Theta}$ is produced in **S2.** by solving:

$$\mathbf{P}_\ell(\boldsymbol{\Theta}) := \underset{\boldsymbol{\Phi} \in \mathcal{M}}{\arg\min} \, \varphi_\ell(\boldsymbol{\Phi}; \boldsymbol{\Theta}), \quad \text{with} \tag{9a}$$

$$\varphi_\ell(\boldsymbol{\Phi}; \boldsymbol{\Theta}) := \max_{k \in [M]} \left\langle \nabla f_k(\boldsymbol{\Theta}), \boldsymbol{\Phi} - \boldsymbol{\Theta} \right\rangle + g(\boldsymbol{\Phi}) - g(\boldsymbol{\Theta}) + \frac{\ell}{2} \|\boldsymbol{\Phi} - \boldsymbol{\Theta}\|_F^2. \tag{9b}$$

Note that the convexity of $g(\boldsymbol{\Theta}) = \lambda\|\boldsymbol{\Theta}\|_1$ ensures a unique solution for Problem (9). We provide a simple yet efficient approach to solve Subproblem (9) through its dual in Appendix B. In addition, Proposition 11 in Appendix B characterizes the weak Pareto optimality for Problem (8).

## 3.3 Theoretical Analysis

We apply Algorithm 1 to Gaussian, Covariance, and Ising models and provide theoretical guarantees.

**Fair Graphical Lasso (Fair GLasso).** Consider $\mathbf{X}_{1:}, \ldots, \mathbf{X}_{N:}$ as i.i.d. samples from $\mathcal{N}(\mathbf{0}, \boldsymbol{\Sigma})$. In the GLasso method [22], the loss is defined as $\mathcal{L}_G(\boldsymbol{\Theta}; \mathbf{X}) := -\log\det(\boldsymbol{\Theta}) + \text{trace}(\mathbf{S}\boldsymbol{\Theta})$, where $\boldsymbol{\Theta}$ is constrained to the set $\mathcal{M} = \{\boldsymbol{\Theta} : \boldsymbol{\Theta} \succ \mathbf{0}, \boldsymbol{\Theta} = \boldsymbol{\Theta}^\top\}$ and $\mathbf{S} = n^{-1}\sum_{i=1}^n \mathbf{X}_{i:}\mathbf{X}_{i:}^\top \in \mathbb{R}^{N \times N}$ is the empirical covariance matrix of $\mathbf{X}$. Extending this to fair GLasso and following (8), the multi-objective optimization problem is formulated as:

$$\underset{\boldsymbol{\Theta}}{\text{minimize}} \quad \mathbf{F}(\boldsymbol{\Theta}) = [\mathcal{L}_G(\boldsymbol{\Theta}; \mathbf{X}) + \lambda\|\boldsymbol{\Theta}\|_1, F_2(\boldsymbol{\Theta}), \cdots, F_M(\boldsymbol{\Theta})]$$

$$\text{subj. to} \quad \boldsymbol{\Theta} \in \mathcal{M} = \{\boldsymbol{\Theta} : \boldsymbol{\Theta} \succ \mathbf{0}, \boldsymbol{\Theta} = \boldsymbol{\Theta}^\top\}. \tag{Fair GLasso}$$

**Assumption B.** *Let $\mathcal{N}^*$ be the set of weakly Pareto optimal points for (8), and $\Omega_{\mathbf{F}}(\boldsymbol{\alpha}) := \{\boldsymbol{\Theta} \in \mathcal{S} \mid \mathbf{F}(\boldsymbol{\Theta}) \preceq \boldsymbol{\alpha}\}$ denote the the level set of $\mathbf{F}$ for $\boldsymbol{\alpha} \in \mathbb{R}^M$. For all $\boldsymbol{\Theta} \in \Omega_{\mathbf{F}}(\mathbf{F}(\boldsymbol{\Theta}^{(0)}))$, there exists $\boldsymbol{\Theta}^* \in \mathcal{N}^*$ such that $\mathbf{F}(\boldsymbol{\Theta}^*) \preceq \mathbf{F}(\boldsymbol{\Theta})$ and*

$$R := \sup_{\mathbf{F}^* \in \mathbf{F}(\mathcal{N}^* \cap \Omega_{\mathbf{F}}(\mathbf{F}(\boldsymbol{\Theta}^0)))} \quad \inf_{\boldsymbol{\Theta} \in \mathbf{F}^{-1}(\{\mathbf{F}^*\})} \|\boldsymbol{\Theta} - \boldsymbol{\Theta}^{(0)}\|_F^2 < \infty.$$

This assumption is satisfied when $\Omega_F(\mathbf{F}(\boldsymbol{\Theta}^{(0)}))$ is bounded [75, 73]. When $M = 1$, it holds if the problem has at least one optimal solution. If $\Omega_F(\mathbf{F}(\boldsymbol{\Theta}^{(0)}))$ is bounded, Assumption B also holds, such as when $F_k$ is strongly convex for some $k \in [M]$.

**Theorem 6.** *Suppose Assumptions A and B hold. Let $\{\boldsymbol{\Theta}^{(t)}\}_{t \geq 1}$ be the sequence generated by Algorithm 1 for solving* (Fair GLasso). *Then,*

$$\sup_{\boldsymbol{\Theta} \in \mathcal{M}} \min_{k \in [M]} \left\{ F_k\left(\boldsymbol{\Theta}^{(t)}\right) - F_k(\boldsymbol{\Theta}) \right\} \leq \frac{\ell R}{2t}.$$

**Fair Covariance Graph (Fair CovGraph).** For the Fair CovGraph, we assume $\mathbf{X}_{1:}, \ldots, \mathbf{X}_{N:}$ are i.i.d. samples from $\mathcal{N}(\mathbf{0}, \boldsymbol{\Sigma})$. We use a sparse estimator for the covariance matrix, ensuring it remains positive definite and specifies the marginal independence graph. Following [62], we define the estimator's loss function as $\mathcal{L}_C(\boldsymbol{\Sigma}, \mathbf{X}) := \frac{1}{2}\|\boldsymbol{\Sigma} - \mathbf{S}\|_F^2 - \tau\log\det(\boldsymbol{\Sigma})$ with $\tau > 0$. Building on this and using (7) and (8), for some nonnegative constants $\gamma_C$ and $\lambda$, we introduce the Fair CovGraph optimization problem, formulated as follows:

$$\underset{\boldsymbol{\Sigma}}{\text{minimize}} \quad \mathbf{F}(\boldsymbol{\Sigma}) = [F_1(\boldsymbol{\Sigma}), F_2(\boldsymbol{\Sigma}), \ldots, F_M(\boldsymbol{\Sigma})]$$

$$\text{subj. to} \quad \boldsymbol{\Sigma} \in \mathcal{M} = \{\boldsymbol{\Sigma} : \boldsymbol{\Sigma} \succ \mathbf{0}, \boldsymbol{\Sigma} = \boldsymbol{\Sigma}^\top\}. \tag{Fair CovGraph}$$

Here, following (8), we have $f_1(\boldsymbol{\Sigma}) = \mathcal{L}_C(\boldsymbol{\Sigma}; \mathbf{X})$ and $f_k(\boldsymbol{\Sigma}) = \Delta_{k-1}(\boldsymbol{\Sigma})$ for $2 \leq k \leq K+1$. Also, $F_1(\boldsymbol{\Sigma}) = f_1(\boldsymbol{\Sigma}) + \lambda\|\boldsymbol{\Sigma}\|_1$ and $F_k(\boldsymbol{\Sigma}) = f_k(\boldsymbol{\Sigma}) + \lambda\|\boldsymbol{\Sigma}\|_1 + \gamma_C\|\boldsymbol{\Sigma}\|_F^2$ for $2 \leq k \leq M = K+1$.

The parameter $\gamma_C$ is used to convexify (Fair CovGraph) and is crucial for ensuring the convergence of Algorithm 1. The following theorem establishes the convergence of Algorithm 1 for (Fair CovGraph).

**Theorem 7.** *Under conditions similar to Theorem 6, by replacing $\boldsymbol{\Theta}$ with $\boldsymbol{\Sigma}$ and $\mathcal{L}_G(\boldsymbol{\Theta}; \mathbf{X})$ with $\mathcal{L}_C(\boldsymbol{\Sigma}, \mathbf{X})$, for the sequence $\{\boldsymbol{\Sigma}^{(t)}\}_{t \geq 1}$ generated by Algorithm 1 applied to* (Fair CovGraph)*, and for $\gamma_C \geq \max\{0, -\Lambda_{\min}(\nabla^2 f_k(\boldsymbol{\Sigma}))\}$ for all $k \in [K]$, we have:*

$$\sup_{\boldsymbol{\Sigma} \in \mathcal{M}} \min_{k \in [M]} \left\{ F_k\left(\boldsymbol{\Sigma}^{(t)}\right) - F_k\left(\boldsymbol{\Sigma}\right) \right\} \leq \frac{\ell R}{2t}.$$

**Fair Binary Ising Network (Fair BinNet).** In this section, we focus on the binary Ising Markov random field as described by [58]. The model considers binary-valued, i.i.d. samples with probability density function defined in (1). Following [31], we consider the following loss function:

$$\mathcal{L}_I(\boldsymbol{\Theta}; \mathbf{X}) = -\sum_{j=1}^{P}\sum_{j'=1}^{P} \theta_{jj'}(\mathbf{X}^\top \mathbf{X})_{jj'} + \sum_{i=1}^{N}\sum_{j=1}^{P} \log\left(1 + \exp\left(\theta_{jj} + \sum_{j' \neq j} \theta_{jj'} x_{ij'}\right)\right). \tag{10}$$

Given some nonnegative constants $\gamma_I$ and $\lambda$, the Fair BinNet objective is defined as:

$$\operatorname*{minimize}_{\boldsymbol{\Theta}} \quad \mathbf{F}(\boldsymbol{\Theta}) = [F_1(\boldsymbol{\Theta}), F_2(\boldsymbol{\Theta}), \ldots, F_M(\boldsymbol{\Theta})]$$

$$\text{subj. to} \quad \boldsymbol{\Theta} \in \mathcal{M} = \{\boldsymbol{\Theta} : \boldsymbol{\Theta} = \boldsymbol{\Theta}^\top\}. \tag{Fair BinNet}$$

Here, following (8), we have $f_1(\boldsymbol{\Theta}) = \mathcal{L}_I(\boldsymbol{\Theta}; \mathbf{X})$, and $f_k(\boldsymbol{\Theta}) = \Delta_{k-1}(\boldsymbol{\Theta})$ for $2 \leq k \leq K+1$. Also, $F_1(\boldsymbol{\Theta}) = f_1(\boldsymbol{\Theta}) + \lambda\|\boldsymbol{\Theta}\|_1$, and $F_k(\boldsymbol{\Theta}) = f_k(\boldsymbol{\Theta}) + \lambda\|\boldsymbol{\Theta}\|_1 + \gamma_I\|\boldsymbol{\Theta}\|_F^2$ for $2 \leq k \leq M = K+1$.

The parameter $\gamma_I$ convexifies Problem (Fair BinNet) and ensures Algorithm 1 converges. The following theorem establishes the convergence of Algorithm 1 for Problem (Fair BinNet).

**Theorem 8.** *Suppose Assumptions A and B hold, and that $\gamma_I \geq \max\{0, -\Lambda_{\min}(\nabla^2 f_k(\boldsymbol{\Theta}))\}$ for all $k \in [K]$. Then, the sequence $\{\boldsymbol{\Theta}^{(t)}\}_{t \geq 1}$ generated by Algorithm 1 for* (Fair BinNet) *satisfies*

$$\sup_{\boldsymbol{\Theta} \in \mathcal{M}} \min_{k \in [M]} \left\{ F_k\left(\boldsymbol{\Theta}^{(t)}\right) - F_k\left(\boldsymbol{\Theta}\right) \right\} \leq \frac{\ell R}{2t}.$$

**Remark 9** (Iteration Complexity of Algorithm 1)**.** *Theorems 6, 7, and 8 establish the global convergence rates of $O(1/t)$ for Algorithm 1 for Gaussian, Covariance, and Ising models, respectively. In contrast to Theorem 6, Theorems 7 and 8 necessitate the inclusion of an additional convex regularization term with parameters $\gamma_C$ and $\gamma_I$, respectively, to achieve Pareto optimality.*

**Remark 10** (Computational Complexity of Algorithm 1)**.** *Given the iteration complexity to achieve $\epsilon$-accuracy is $O(\epsilon^{-1})$, the overall time complexity of our optimization procedure becomes $O\left(\epsilon^{-1}\max(NP^2, P^3)\right)$. Assuming a small number of groups ($K << N, P, 1/\epsilon$), the complexity aligns with that of standard proximal gradient methods used for covariance and inverse covariance estimation, making it feasible for large $N$ and $P$. To further support the theoretical analysis, sensitivity analysis experiments are conducted to investigate the impact of varying $P$, $N$, $K$, and group imbalance on the performance of the proposed methods. Note that the complexity of Algorithm 1 applied to (Fair BinNet) depends on the choice of subproblem solver (e.g., first or second order) due to the nonlinearity of (10). Further experiments and discussions are detailed in Appendices D.6-D.9.*

## 4 Experiment

### 4.1 Experimental Setup

**Baseline.** The Iterative Shrinkage-Thresholding Algorithm (ISTA) is widely used for sparse inverse covariance estimation [60] due to its simplicity and efficiency. We adapt ISTA for the Covariance and Ising models and use them as a baseline to compare with our proposed Fair GMs. Note that our Fair GMs reduce to ISTA for Gaussian, Covariance, and Ising models if $M = 1$ in (8). The detailed ISTA algorithm used in this study is provided in Appendix D for reference.

**Parameters and Architecture.** The initial iterate $\boldsymbol{\Theta}^{(0)}$ is chosen based on the highest graph disparity error among local graphs. This initialization can improve fairness by minimizing larger disparity errors. The $\ell_1$-norm coefficient $\lambda$ is fixed for each dataset, searched over a grid in $\{1e-5, \ldots, 0.01, \ldots, 0.1, 1\}$. Tolerance $\epsilon$ is set to $1e-5$, with a maximum of $1e+7$ iterations. The initial value of $\ell$ is $1e-2$, undergoing a line search at each iteration $t$ with a decay rate of $0.1$.

Table 1: Numerical outcomes in terms of PCEE. The last row calculates the difference in PCEE between the two groups: the smaller, the better, and the best value is in bold.

| Group | Std. GLasso | Fair GLasso | Std. CovGraph | Fair CovGraph | Std. BinNet | Fair BinNet |
|---|---|---|---|---|---|---|
| 1 | 0.7491 | 0.7538 | 0.8537 | 0.8750 | 0.4138 | 0.9540 |
| 2 | 0.8479 | 0.8108 | 0.9502 | 0.9357 | 0.8974 | 0.8974 |
| **Difference** | 0.0987 | **0.0569** | 0.0965 | **0.0607** | 0.4836 | **0.0566** |

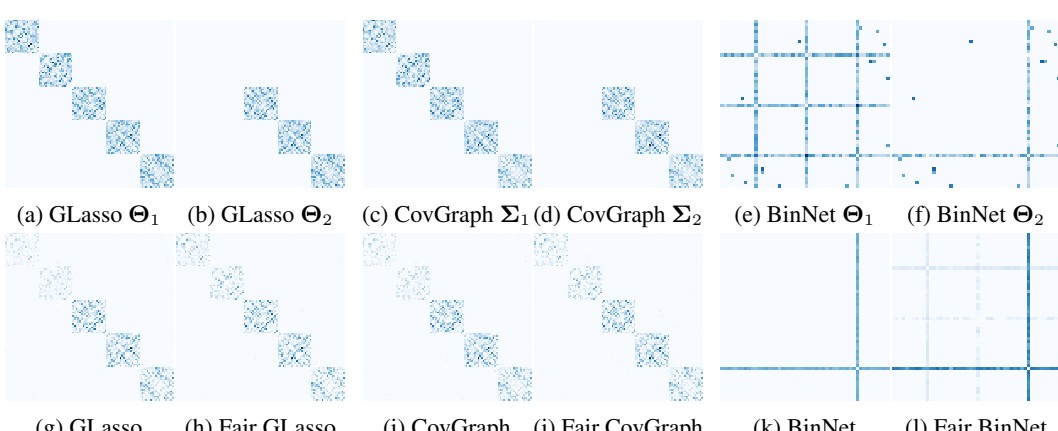

(a) GLasso $\boldsymbol{\Theta}_1$   (b) GLasso $\boldsymbol{\Theta}_2$   (c) CovGraph $\boldsymbol{\Sigma}_1$ (d) CovGraph $\boldsymbol{\Sigma}_2$   (e) BinNet $\boldsymbol{\Theta}_1$   (f) BinNet $\boldsymbol{\Theta}_2$

(g) GLasso   (h) Fair GLasso   (i) CovGraph   (j) Fair CovGraph   (k) BinNet   (l) Fair BinNet

Figure 2: Comparison of original graphs utilized in synthetic data creation for two groups, graph reconstruction using standard GMs, and fair graph reconstruction via Fair GMs. The diagonal elements are set to zero to enhance the visibility of the off-diagonal pattern.

**Evaluation Criteria.** In our experiments, we introduce three metrics to evaluate the performance of our methods and the baseline methods:

1. Value of the objective function of GM: $F_1 := \mathcal{L}(\boldsymbol{\Theta}; \mathbf{X}) + \lambda \|\boldsymbol{\Theta}\|_1$.
2. Summation of pairwise graph disparity error for fairness: $\Delta := \sum_{k=1}^{K} \Delta_k$.
3. Proportion of correctly estimated edges:
$$\text{PCEE} := \big( \sum_{j,j' \in [P]} \mathbf{1}\{\hat{\boldsymbol{\Theta}}_{jj'} \geq \lambda \text{ and } |\boldsymbol{\Theta}_{jj'}| \geq \lambda\} \big) / \big( \sum_{j,j' \in [P]} \mathbf{1}\{|\boldsymbol{\Theta}_{jj'}| \geq \lambda\} \big),$$

where $\mathbf{1}\{\cdot\}$ is the indicator function, $\boldsymbol{\Theta}$ and $\hat{\boldsymbol{\Theta}}$ are the groundtruth and estimated graph.

### 4.2 Simulation Study of Fair GLasso and CovGraph

In the simulation, we construct two $100 \times 100$ block diagonal covariance matrices, $\boldsymbol{\Sigma}_1$ and $\boldsymbol{\Sigma}_2$ (Figures 2c and 2d). These matrices correspond to two sensitive groups and are created following the rigorous process in Appendix D.2. Each graph has three consistent diagonal blocks, with Group 1 also featuring two distinct blocks indicating bias. For each group, we derive the ground truth graphs by $\boldsymbol{\Theta}_1 = \boldsymbol{\Sigma}_1^{-1}$ and $\boldsymbol{\Theta}_2 = \boldsymbol{\Sigma}_2^{-1}$ (Figures 2a and 2b). Datasets are generated from normal distributions: 1000 samples from $\mathcal{N}(\mathbf{0}, \boldsymbol{\Sigma}_1)$ for the first group, and 1000 samples from $\mathcal{N}(\mathbf{0}, \boldsymbol{\Sigma}_2)$ for the second.

**Results.** Figure 2g shows the global graph derived using Standard GLasso on the entire dataset, where the two top-left blocks are not distinctly marked, suggesting bias towards $\boldsymbol{\Theta}_2$. In contrast, Figure 2h shows a graph from our method that enhances block visibility, reducing bias. This improvement is supported by the results in Table 1, where the PCEE difference of Fair GLasso is smaller than that of Standard GLasso. Comparable efficacy in bias reduction for CovGraph is shown in Figures 2c, 2d, 2i, 2j, and Table 1, demonstrating our methods' effectiveness in achieving fairness.

### 4.3 Simulation Study of Fair BinNet

We provided two simulation networks: $\boldsymbol{\Theta}_1$ for Group 1 with $P = 50$ nodes and three hub nodes, and $\boldsymbol{\Theta}_2$ for Group 2 with one hub node (see Appendix D.3 for details). Adjacency matrices are shown in Figures 2e and 2f. We generate $N_1 = 500$ and $N_2 = 1000$ observations via Gibbs sampling, updating each variable $x_j^{(t+1)}$ at iteration $t + 1$ using the Bernoulli distribution: $x_j^{(t+1)} \sim$

Bernoulli$(z_\theta/(1+z_\theta))$, where $z_\theta = \exp(\theta_{jj} + \sum_{j' \neq j} \theta_{jj'} x_{j'}^{(t)})$. The first 10,000 iterations are designated as the burn-in period to ensure statistical independence among observations. Finally, observations are collected at every 100th iteration.

**Results.** Figure 2k illustrates the global graph from Standard BinNet, which is predominantly biased towards $\Theta_2$ by identifying only one hub node. In contrast, Figure 2l, derived from Fair BinNet, presents a more balanced structure. While this improvement might not be visually evident, the quantitative results in Table 1 and Table 2 confirm it. Table 1 reveals that PCEE for Group 1 improved significantly, increasing from 0.4444 to 0.7485. Conversely, PCEE for Group 2 exhibited a decrease from 0.9481 to 0.7662. This convergence in performance metrics between the two groups indicates a more balanced distribution of predictive errors, thus enhancing the overall fairness of the model.

## 4.4  Application of Fair GLasso to Gene Regulatory Network

We apply GLasso to analyze RNA-Seq data from TCGA, focusing on lung adenocarcinoma. The data includes expression levels of 60,660 genes from 539 patients. From these, 147 KEGG pathway genes [36] are selected to construct a gene regulatory network. GLasso reveals conditional dependencies, aiding in understanding cancer genetics and identifying therapeutic targets. However, initially, this method, without accounting for sex-based differences, risks overlooking critical biological disparities, potentially skewing drug discovery and health outcomes across genders. Therefore, we divide the patient cohort into two groups based on sex: 248 males and 291 females. This stratification enables the use of Fair GLasso, which creates a more equitable gene regulatory network by accounting for these differences. The parameter $\lambda$ is set to 0.03 for this experiment. Additionally, each variable is normalized to achieve a zero mean and a unit variance.

**Results.** The gene networks identified by GLasso and Fair GLasso are presented in Figures 3a-3b. GLasso identified several hub nodes, including NCOA1, BRCA1, FGF8, AKT1, NOTCH4, and CSNK1A1L. In contrast, Fair GLasso uniquely detected PIK3CD, suggesting its potential relevance in capturing sex-specific differences in lung adenocarcinoma. Although direct evidence linking PIK3CD exclusively to sex-specific traits in cancer is limited, this finding aligns with recent insights into sex-specific regulatory mechanisms in cancer [63, 45]. PIK3CD is a key component of the PI3K/Akt signaling pathway, which is involved in cell regulation and frequently implicated in various malignancies. The identification of PIK3CD by Fair GLasso demonstrates its potential to uncover biologically relevant genes that may be overlooked in conventional analyses, enhancing our understanding of lung adenocarcinoma and facilitating the development of personalized therapies.

## 4.5  Application of Fair GLasso to Amyloid / Tau Accumulation Network

The performance of GLasso and Fair GLasso is evaluated using AV45 and AV1451 PET imaging data from the Alzheimer's Disease Neuroimaging Initiative (ADNI) [85, 86], focusing on amyloid-$\beta$ and tau deposition in the brain. The dataset includes standardized uptake value ratios of AV45 and AV1451 tracers in 68 brain regions, as defined by the Desikan-Killiany atlas [14], collected from 1,143 participants. An amyloid (or tau) accumulation network [68] is constructed to investigate the pattern of amyloid (or tau) accumulation. GLasso and Fair GLasso are used to uncover conditional dependencies between brain regions, providing insights into Alzheimer's disease progression and identifying potential biomarkers for early diagnosis and treatment response monitoring. To examine the influence of sensitive attributes on the network structure, marital status, and race are incorporated as exemplary sensitive attributes due to their reported association with dementia risk [71, 49]. Comprehensive details regarding the experiments, results, and analysis are provided in Appendix 4.5.

## 4.6  Application of Fair CovGraph to Credit Datasets

The performance of Fair CovGraph is evaluated using the Credit Datasets [90] from the UCI Machine Learning Repository [2]. These datasets have been previously used in research on Fair PCA [55, 83], which shows potential for improvement through sparse covariance estimation. The dataset composition is detailed in Table 5 in Appendix D.5, with categorizations based on gender, marital status, and education level. For this experiment, the parameters $\tau$ and $\lambda$ are set to 0.01 and 0.1, respectively. Each variable in the dataset is standardized to have a mean of zero and a variance of one. As shown in Table 2, our Fair CovGraph achieves a 53.75% increase in fairness with only a 0.42% decrease in the graph objective, demonstrating the strong ability of our method to attain fairness.

Table 2: Outcomes in terms of the value of the objective function ($F_1$), the summation of the pairwise graph disparity error ($\Delta$), and the average computation time in seconds ($\pm$ standard deviation) from 10 repeated experiments. "$\downarrow$" means the smaller, the better, and the best value is in bold. Note that both $F_1$ and $\Delta$ are deterministic.

| Dataset | $F_1 \downarrow$ | | $\%F_1 \uparrow$ | $\Delta \downarrow$ | | $\%\Delta \uparrow$ | Runtime $\downarrow$ | |
|---|---|---|---|---|---|---|---|---|
| | GM | Fair GM | | GM | Fair GM | | GM | Fair GM |
| Simulation (GLasso) | **97.172** | 97.443 | -0.28% | 7.8149 | **0.6237** | +92.02% | **0.395 ($\pm$ 0.24)** | 32.32 ($\pm$ 1.5) |
| Simulation (CovGraph) | **14.319** | 14.484 | -1.15% | 5.2627 | **0.3889** | +92.61% | **0.254 ($\pm$ 0.05)** | 12.58 ($\pm$ 0.3) |
| Simulation (BinNet) | **34.363** | 34.362 | -0.00% | $1\times10^{-6}$ | **0.0000** | +100.0% | **0.536 ($\pm$ 0.15)** | 3.29 ($\pm$ 0.48) |
| TCGA Dataset | **127.96** | 128.11 | -0.11% | 2.5875 | **0.0742** | +97.13% | **8.468 ($\pm$ 1.17)** | 63.72 ($\pm$ 5.9) |
| Credit Dataset | **9.2719** | 9.3110 | -0.42% | 0.5436 | **0.2513** | +53.76% | **0.256 ($\pm$ 0.08)** | 64.20 ($\pm$ 1.8) |
| LFM-1b Dataset | 87.531 | **87.138** | +0.45% | 0.0040 | **0.0001** | +96.60% | **0.669 ($\pm$ 0.19)** | 41.19 ($\pm$ 3.5) |

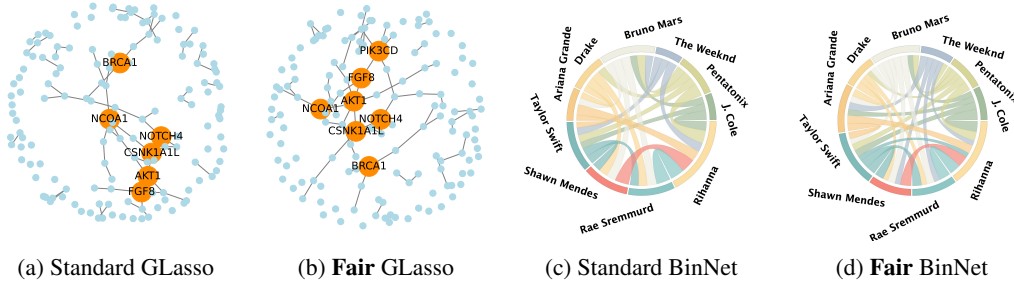

(a) Standard GLasso  (b) **Fair** GLasso  (c) Standard BinNet  (d) **Fair** BinNet

Figure 3: (a)-(b) Comparison of graphs generated by standard GLasso and Fair GLasso on TCGA Dataset. Week edges are removed for visibility, and hub nodes that own at least 4 edges are highlighted. (c)-(d) Comparison of sub-graphs generated by standard BinNet and Fair BinNet on LFM-1b Dataset. Fair BinNet provides a more diversified recommendation network.

## 4.7 Application of Fair BinNet to Music Recommendation Systems

LFM-1b Dataset[4] contains over one billion listening events intended for use in recommendation systems [66]. In this experiment, we use the user-artist play counts dataset to construct a recommendation network of artists. Our analysis focuses on 80 artists intersecting the 2016 Billboard Artist 100 and 1,807 randomly selected users who listened to at least 400 songs. We transform the play counts into binary datasets for BinNet models, setting play counts above 0 to 1 and all others to 0.

This experiment examines male and female categories, stratifying the dataset into two groups with 1,341 and 466 samples, respectively. We set the BinNet models' parameter, $\lambda$, to $1e-5$.

**Results.** Figures 3c-3d show the recommendation networks of the 2016 Billboard Top 10 popular music artists based on BinNet's and Fair BinNet's outputs. The comparative analysis reveals that Fair BinNet provides a more diversified recommendation network, particularly for the artist The Weeknd. Enhancing fairness fosters cross-group musical preference exchange, breaks the echo chamber effect, and broadens users' exposure to potentially intriguing music, enhancing the user-friendliness of the music recommendation system.

## 4.8 Trade-off Analysis

In fairness studies, the trade-off between fairness and model accuracy presents a fundamental challenge. An effective fair method should achieve equitable outcomes while maintaining strong accuracy performance. We evaluate this balance by analyzing the percentage changes in both accuracy and fairness metrics. Specifically, we define these changes as: $\%F_1 = -\frac{F_1 \text{ of Fair GM} - F_1 \text{ of GM}}{F_1 \text{ of GM}} \times 100\%$, and $\%\Delta = -\frac{\Delta \text{ of Fair GM} - \Delta \text{ of GM}}{\Delta \text{ of GM}} \times 100\%$.

Our empirical results (Tables 2, 4, and Figure 4) demonstrate that Fair GMs substantially reduce disparity error, thereby improving fairness, while incurring only minimal degradation in the objective function's value. This favorable trade-off validates the effectiveness of our approach. However, Fair GMs do face computational challenges, primarily stemming from two sources: local graph computation and multi-objective optimization.

---

[4]Available at http://www.cp.jku.at/datasets/LFM-1b/.

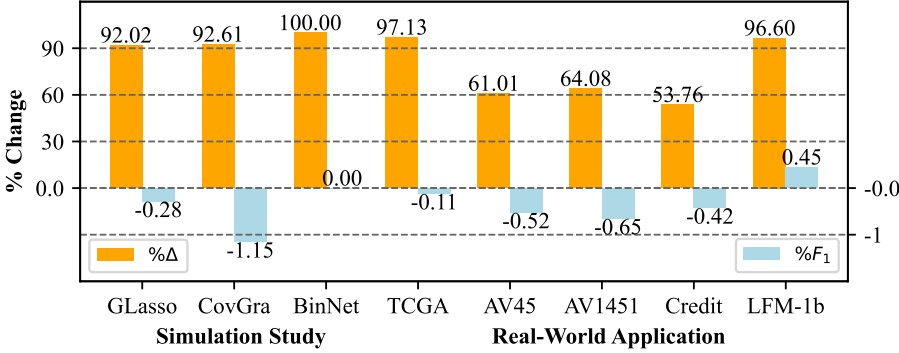

Figure 4: Percentage change from GMs to Fair GMs (results from Tables 2 and 4). $\%F_1$ is slight, while $\%\Delta$ changes are substantial, signifying fairness improvement without serious accuracy sacrifice.

To address these limitations, we propose several solutions. The local graph learning phase can be accelerated using advanced graphical model algorithms such as QUIC [32], SQUIC [7], PISTA [69], GISTA [60], OBN [57], or ALM [67]. Moreover, to mitigate the increased complexity from multiple objectives, we introduce a stochastic objective selection strategy, randomly sampling a subset of objectives in each iteration. This approach effectively reduces computational overhead while maintaining model fairness and performance. To validate these computational considerations, we conducted additional experiments using GLasso, with detailed results presented in the Appendix D.10.

## 5   Conclusion

In this paper, we tackle fairness in graphical models (GMs) such as Gaussian, Covariance, and Ising models, which interpret complex relationships in high-dimensional data. Standard GMs exhibit bias with sensitive group data. We propose a framework incorporating a pairwise graph disparity error term and a custom loss function into a nonsmooth multi-objective optimization. This approach enhances fairness without compromising performance, validated by experiments on synthetic and real-world datasets. However, it increases computational complexity and may be sensitive to the choice of loss function and balancing multiple objectives. Future research can include:

**F1.** Integrating our Fair GMs approach with supervised methods for downstream tasks, including spectral clustering [82], graph regularized dimension reduction [76].

**F2.** Developing novel group fairness notions based on sensitive attributes within our nonsmooth multi-objective optimization framework.

**F3.** Extending fairness to ordinal data models, which are crucial for socioeconomic and health-related applications [27], neighborhood selection [50], and partial correlation estimation [40].

Despite some limitations of Fair GMs for larger group sizes, this work demonstrates the potential of *nonsmooth multi-objective optimization* as a powerful tool for mitigating biases and promoting fairness in *high-dimensional* graph-based machine learning, contributing to the development of more equitable and responsible AI systems across a wide range of domains.

## 6   Acknowledgements

This work was supported in part by the NIH grants U01 AG066833, U01 AG068057, U19 AG074879, RF1 AG068191, RF1 AG063481, R01 LM013463, P30 AG073105, U01 CA274576, RF1-AG063481, R01-AG071174, and U01-CA274576. The ADNI data were obtained from the Alzheimer's Disease Neuroimaging Initiative database (https://adni.loni.usc.edu), funded by NIH U01 AG024904. The authors thank Laura Balzano and Alfred O. Hero for their helpful suggestions and discussions.

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

# Contents

# A    Summary of the Notations

Table 3: Summary of the Notations

| Notation | Description |
|---|---|
| $\mathbf{1}\{\cdot\}$ | Indicator function |
| $\|\mathbf{A}\|_1$ | $\ell_1$–norm: $\sum_{ij} |a_{ij}|$ |
| $\|\mathbf{A}\|_F$ | Frobenius norm: $(\sum_{ij} |a_{ij}|^2)^{1/2}$ |
| $[M]$ | The set $\{1, 2, \ldots, M\}$ |
| $\Lambda_i(\mathbf{A})$ | $i$th eigenvalue of $\mathbf{A}$ |
| $\Lambda_{\min}(\mathbf{A})$ | The smallest eigenvalue of $\mathbf{A}$ |
| $M_{\alpha h}(x)$ | Moreau envelope: $\min_y \left[ h(y) + \frac{1}{2\alpha}\|x - y\|^2 \right]$ |
| $\mathbf{prox}_{\alpha h}(x)$ | Proximal operator: $\arg\min_y \left[ h(y) + \frac{1}{2\alpha}\|x - y\|^2 \right]$ |
| $\eta_{\frac{1}{\ell}\lambda}(x)$ | Soft thresholding operator: $\mathrm{sign}(x) \max(|x| - \frac{1}{\ell}\lambda, 0)$ |
| $L$ | Lipschitz constant |
| $P$ | Number of variables in the data matrix |
| $N$ | Number of observations in the data matrix |
| $N^k$ | Number of observations in the $k$th group data matrix |
| $K$ | Number of sensitive groups in the data matrix |
| $M$ | Number of objectives in the multi-objective optimization problem |
| $t$ | Current iteration of Algorithm 1 |
| $\lambda$ | Hyper-parameter of the $\ell_1$–regularization term |
| $\gamma_C$ | Hyper-parameter of the convex regularization term in (Fair CovGraph) |
| $\gamma_I$ | Hyper-parameter of the convex regularization term in (Fair BinNet) |
| $\ell$ | Selected constant $> L$ |
| $\mathbf{X}$ | Data matrix |
| $\mathbf{X}_k$ | Data matrix of $k$th group |
| $\mathbf{X}_{i:}$ | Observations in the data matrix |
| $\mathbf{S}$ | Sample covariance matrix: $n^{-1}\sum_{i=1}^n \mathbf{X}_{i:}\mathbf{X}_{i:}^\top$ |
| $\boldsymbol{\Sigma}$ | Covariance matrix |
| $\boldsymbol{\Phi}$ | Conditional independence graph (inverse covariance matrix) |
| $\boldsymbol{\Theta}$ | Conditional independence graph (inverse covariance matrix) |
| $\boldsymbol{\Theta}_k$ | Conditional independence graph of $k$th group |
| $\boldsymbol{\Theta}^*$ | Optimal conditional independence graph |
| $\mathcal{L}$ | General Loss function |
| $\mathcal{L}_G$ | Loss function of GLasso: $-\log\det(\boldsymbol{\Theta}) + \mathrm{trace}(\mathbf{S}\boldsymbol{\Theta})$ |
| $\mathcal{L}_C$ | Loss function of CovGraph: $\frac{1}{2}\|\boldsymbol{\Sigma} - \mathbf{S}\|_F^2 - \tau\log\det(\boldsymbol{\Sigma})$ |
| $\mathcal{L}_I$ | Loss function of BinNet; refer to (10) |
| $\mathcal{E}_k$ | Graph disparity error of $k$th group |
| $\Delta_k$ | Pairwise graph disparity error of $k$th group |
| $\Delta$ | Summation of pairwise graph disparity error: $\sum_{k=1}^K \Delta_k$ |
| $\varphi_\ell$ | $\max_{k=1,\ldots,M} \langle \nabla f_k(\boldsymbol{\Theta}), \boldsymbol{\Phi} - \boldsymbol{\Theta} \rangle + g(\boldsymbol{\Phi}) - g(\boldsymbol{\Theta}) + \frac{\ell}{2}\|\boldsymbol{\Phi} - \boldsymbol{\Theta}\|_F^2$ |
| $\mathbf{F}$ | Objective function of the multi-objective optimization problem |
| $F_k$ | $k$th objective in the multi-objective optimization problem |
| $\mathbf{P}_\ell$ | solutions of the min-max problem for each $\ell$: $\arg\min_{\boldsymbol{\Phi}\in\mathcal{M}} \varphi_\ell(\boldsymbol{\Phi}; \boldsymbol{\Theta})$ |
| $R$ | $\sup_{\mathbf{F}^*\in\mathbf{F}(\mathcal{N}^*\cap\Omega_\mathbf{F}(\mathbf{F}(\boldsymbol{\Theta}^0)))} \quad \inf_{\boldsymbol{\Theta}\in\mathbf{F}^{-1}(\{\mathbf{F}^*\})} \left\|\boldsymbol{\Theta} - \boldsymbol{\Theta}^{(0)}\right\|_F^2$ |
| $\mathcal{M}$ | Convex constraint subset of $\mathbb{R}^{P\times P}$ |
| $\mathcal{C}$ | Standard simplex: $\left\{ \boldsymbol{\rho}\in\mathbb{R}^M : \sum_{k=1}^M \rho_k = 1, \ \rho_k \in [0,1] \ \forall k\in[M] \right\}$ |

# B  Addendum to Section 3

## B.1  Dual Reformulation and Computation of Subproblem (9)

In this section, we provide a dual method for solving the Subproblem (9) defined as:

$$\min_{\boldsymbol{\Phi} \in \mathcal{M}} \quad \varphi_\ell\left(\boldsymbol{\Phi}; \boldsymbol{\Theta}\right),$$

$$\text{with} \quad \varphi_\ell\left(\boldsymbol{\Phi}; \boldsymbol{\Theta}\right) = \max_{k \in \{1,\ldots,M\}} \langle \nabla f_k(\boldsymbol{\Theta}), \boldsymbol{\Phi} - \boldsymbol{\Theta} \rangle + g(\boldsymbol{\Phi}) - g(\boldsymbol{\Theta}) + \frac{\ell}{2} \|\boldsymbol{\Phi} - \boldsymbol{\Theta}\|_F^2, \quad (11)$$

for all $\ell > L$ where $L$ is defined in Assumption A.

For simplicity, let

$$\psi_{k,\ell}\left(\boldsymbol{\Phi}; \boldsymbol{\Theta}\right) = \langle \nabla f_k(\boldsymbol{\Theta}), \boldsymbol{\Phi} - \boldsymbol{\Theta} \rangle + g(\boldsymbol{\Phi}) - g(\boldsymbol{\Theta}) + \frac{\ell}{2} \|\boldsymbol{\Phi} - \boldsymbol{\Theta}\|_F^2. \quad (12)$$

By considering the standard simplex,

$$\mathcal{C} := \left\{ \boldsymbol{\rho} \in \mathbb{R}^M : \sum_{k=1}^M \rho_k = 1, \ \rho_k \in [0,1], \ \forall k \in [M] \right\}, \quad (13)$$

we reformulate (11) as

$$\min_{\boldsymbol{\Phi} \in \mathcal{M}} \max_{\boldsymbol{\rho} \in \mathcal{C}} \sum_{k=1}^M \rho_k \psi_{k,\ell}\left(\boldsymbol{\Phi}; \boldsymbol{\Theta}\right). \quad (14)$$

By leveraging the convexity of $\mathcal{M}$, the compactness and convexity of $\mathcal{C}$, and the convexity-concavity property of $\sum_{k=1}^M \rho_k \psi_{k,\ell}\left(\boldsymbol{\Phi}; \boldsymbol{\Theta}\right)$ with respect to $\boldsymbol{\Phi}$ and $\boldsymbol{\rho}$, respectively, we can invoke *Sion's minimax theorem* to reformulate (14) as follows:

$$\max_{\boldsymbol{\rho} \in \mathcal{C}} \min_{\boldsymbol{\Phi} \in \mathcal{M}} \sum_{k=1}^M \rho_k \psi_{k,\ell}\left(\boldsymbol{\Phi}; \boldsymbol{\Theta}\right). \quad (15)$$

Expanding on the definition of $\psi_{k,\ell}$, we arrive at the following expression:

$$\max_{\boldsymbol{\rho} \in \mathcal{C}} \min_{\boldsymbol{\Phi} \in \mathcal{M}} \sum_{k=1}^M \rho_k \psi_{k,\ell}\left(\boldsymbol{\Phi}; \boldsymbol{\Theta}\right)$$

$$= \max_{\boldsymbol{\rho} \in \mathcal{C}} \min_{\boldsymbol{\Phi} \in \mathcal{M}} \left[ g\left(\boldsymbol{\Phi}\right) + \frac{\ell}{2} \left\| \boldsymbol{\Phi} - \left( \boldsymbol{\Theta} + \frac{1}{\ell} \sum_{k=1}^M \rho_k \nabla f_k\left(\boldsymbol{\Theta}\right) \right) \right\|_F^2 \right]$$

$$- \frac{1}{2\ell} \left\| \sum_{k=1}^M \rho_k \nabla f_k\left(\boldsymbol{\Theta}\right) \right\|_F^2 - g\left(\boldsymbol{\Theta}\right)$$

$$= \max_{\boldsymbol{\rho} \in \mathcal{C}} \ \ell M_{\frac{1}{\ell}g}\left( \boldsymbol{\Theta} - \frac{1}{\ell} \sum_{k=1}^M \rho_k \nabla f_k\left(\boldsymbol{\Theta}\right) \right)$$

$$- \frac{1}{2\ell} \left\| \sum_{k=1}^M \rho_k \nabla f_k\left(\boldsymbol{\Theta}\right) \right\|_F^2 - g\left(\boldsymbol{\Theta}\right), \quad (16)$$

where Moreau envelope $M_{\alpha h}(x)$ and the proximal operator are defined as

$$M_{\alpha h}(x) := \min_y \left[ h(y) + \frac{1}{2\alpha} \|x - y\|^2 \right], \quad (17a)$$

$$\mathbf{prox}_{\alpha h}(x) := \arg\min_y \left[ h(y) + \frac{1}{2\alpha} \|x - y\|^2 \right]. \quad (17b)$$

Problem (16) is equivalent to the dual problem:

$$\max_{\boldsymbol{\rho} \in \mathbb{R}^M} \ \omega(\boldsymbol{\rho}) \qquad \text{subj. to} \qquad \boldsymbol{\rho} \succeq \mathbf{0} \ \text{and} \ \sum_{k=1}^M \rho_k = 1, \quad (18a)$$

where

$$\omega(\boldsymbol{\rho}) = \ell M_{\frac{1}{\ell}g}\left(\boldsymbol{\Theta} - \frac{1}{\ell}\sum_{k=1}^{M}\rho_k \nabla f_k(\boldsymbol{\Theta})\right) - \frac{1}{2\ell}\left\|\sum_{k=1}^{M}\rho_k \nabla f_k(\boldsymbol{\Theta})\right\|_F^2 - g(\boldsymbol{\Theta}). \tag{18b}$$

Upon solving the dual Problem (18), the optimal solution $\boldsymbol{\Phi}^*$ of (11) is obtained through:

$$\boldsymbol{\Phi}^* = \mathbf{prox}_{\frac{1}{\ell}g}\left(\boldsymbol{\Theta} - \frac{1}{\ell}\sum_{k=1}^{M}\rho_k \nabla f_k(\boldsymbol{\Theta})\right). \tag{19}$$

In the implementation, for the given $g(\boldsymbol{\Theta}) = \lambda\|\boldsymbol{\Theta}\|_1$, $\mathbf{prox}_{\frac{1}{\ell}g}$ is computed using soft thresholding $\eta_{\frac{1}{\ell}\lambda}$, as shown below:

$$\left(\eta_{\frac{1}{\ell}\lambda}(\mathbf{x})\right)_j = \mathrm{sign}(x_j)\max\left(|x_j| - \frac{1}{\ell}\lambda, 0\right). \tag{20}$$

To provide a clear and logical summary of the iterative update process in Algorithm 1, we proceed as follows: At each iteration $t$, the update for $\boldsymbol{\Theta}^{(t+1)}$ is performed by inputting $\boldsymbol{\Theta}^{(t)}$ and solving the Subproblem (11). This is achieved by utilizing the `scipy.optimize.minimize` function with the `method="trust-constr"` option to solve the dual problem. Specifically, for given constants $\ell > L$ and $\lambda > 0$, and the calculated $\boldsymbol{\rho}^{(t)} \in \mathcal{C}$ at the $t$th iteration, the update rule for $\boldsymbol{\Theta}^{(t+1)}$ is given by:

$$\boldsymbol{\Theta}^{(t+1)} = \eta_{\frac{1}{\ell}\lambda}\left(\boldsymbol{\Theta}^{(t)} - \frac{1}{\ell}\sum_{k=1}^{M}\rho_k^{(t)}\nabla f_k\left(\boldsymbol{\Theta}^{(t)}\right)\right), \tag{21}$$

which incorporates the proximal operator and the weighted sum of gradients. Through solving Subproblem (9), the following proposition characterizes the weak Pareto optimality in the context of multi-objective optimization Problem (8):

**Proposition 11.** *Let $\omega_\ell(\boldsymbol{\Theta}) := \min_{\boldsymbol{\Phi}\in\mathcal{M}}\varphi_\ell(\boldsymbol{\Phi};\boldsymbol{\Theta})$ and $\mathbf{P}_\ell$ be defined as in (9). Then,*

*(i)    The following conditions are equivalent:*
> *(a) $\boldsymbol{\Theta}$ is a weakly Pareto optimal point;*
> *(b) $\mathbf{P}_\ell(\boldsymbol{\Theta}) = \boldsymbol{\Theta}$;*
> *(c) $\omega_\ell(\boldsymbol{\Theta}) = 0$.*

*(ii)   The mappings $\mathbf{P}_\ell$ and $\omega_\ell$ are both continuous.*

*Proof.* The proof follows from [73, Lemma 3.2] and the convexity of $f_k$. The detailed convexity analyses for Fair GLasso, Fair CovGraph, and Fair BinNet are provided in Sections C.2, C.3, and C.4, respectively. $\quad\square$

As demonstrated in the analysis of Subproblem (9) in the beginning of this section, the proposition implies that the descent direction is the minimum norm matrix within the convex hull of the gradients of all objectives. Furthermore, this direction is non-increasing with respect to each individual objective function. This property ensures that the chosen descent direction simultaneously minimizes the overall norm while guaranteeing non-increasing behavior for each objective, thereby facilitating the optimization process in a multi-objective setting.

### B.2    Subproblem Solver for Fair GMs

### B.2.1    Fair GLasso

To update $\boldsymbol{\Theta}^{(t)}$ in Algorithm 1 applied to (Fair GLasso), the iterative update formula in Equation (21) is used at each iteration. The gradients for the functions $f_1$ and $\{f_{k+1}\}_{k=1}^{M-1}$ are computed as follows:

The gradient of $f_1$ with respect to $\boldsymbol{\Theta}$ is given by:

$$\nabla f_1(\boldsymbol{\Theta}) = \mathbf{S} - \boldsymbol{\Theta}^{-1}, \tag{22}$$

where $\mathbf{S}$ is the sample covariance matrix and $\boldsymbol{\Theta}^{-1}$ is the inverse of the precision matrix $\boldsymbol{\Theta}$.

The gradient of $f_{k+1}$ for $k = 1, \ldots, M-1$ with respect to $\boldsymbol{\Theta}$ is given by:

$$\nabla f_{k+1}(\boldsymbol{\Theta}) = \sum_{s \in [K], s \neq k} \left( \text{trace}(\mathbf{S}_k \boldsymbol{\Theta}) - \text{trace}(\mathbf{S}_s \boldsymbol{\Theta}) + \log \det(\boldsymbol{\Theta}_k^*) \right.$$
$$\left. - \text{trace}(\mathbf{S}_k \boldsymbol{\Theta}_k^*) - \log \det(\boldsymbol{\Theta}_s^*) + \text{trace}(\mathbf{S}_s \boldsymbol{\Theta}_s^*) \right) (\mathbf{S}_k - \mathbf{S}_s), \tag{23}$$

where $K$ is the number of groups, $\mathbf{S}_k$ and $\mathbf{S}_s$ are the sample covariance matrices for groups $k$ and $s$, respectively, and $\boldsymbol{\Theta}_k^*$ and $\boldsymbol{\Theta}_s^*$ are the optimal precision matrices for groups $k$ and $s$ obtained by solving the group-specific GLasso problems.

### B.2.2   Fair CovGraph

To refine the iterative update rule for estimating the fair covariance matrix $\boldsymbol{\Sigma}$ in (Fair GLasso) using Algorithm 1, the gradients for $f_1$ and the set $\{f_{k+1}\}_{k=1}^{M-1}$ are computed as follows:

The gradient of $f_1$ with respect to $\boldsymbol{\Sigma}$ is given by:

$$\nabla f_1(\boldsymbol{\Sigma}) = \boldsymbol{\Sigma} - \mathbf{S} - \tau \boldsymbol{\Sigma}^{-1}, \tag{24}$$

where $\mathbf{S}$ is the pooled sample covariance matrix, $\tau$ is the regularization parameter, and $\boldsymbol{\Sigma}^{-1}$ is the inverse of the covariance matrix $\boldsymbol{\Sigma}$.

The gradient of $f_{k+1}$ for $k = 1, \ldots, M-1$ with respect to $\boldsymbol{\Sigma}$ is given by:

$$\nabla f_{k+1}(\boldsymbol{\Sigma}) = \sum_{s \in [K], s \neq k} \left( \frac{1}{2} \|\boldsymbol{\Sigma} - \mathbf{S}_k\|_F^2 - \frac{1}{2} \|\boldsymbol{\Sigma} - \mathbf{S}_s\|_F^2 + \tau \log \det(\boldsymbol{\Sigma}_k^*) \right.$$
$$\left. - \frac{1}{2} \|\boldsymbol{\Sigma}_k^* - \mathbf{S}_k\|_F^2 - \tau \log \det(\boldsymbol{\Sigma}_s^*) + \frac{1}{2} \|\boldsymbol{\Sigma}_s^* - \mathbf{S}_s\|_F^2 \right) (\mathbf{S}_s - \mathbf{S}_k), \tag{25}$$

where $K$ is the number of groups, $\mathbf{S}_k$ and $\mathbf{S}_s$ are the sample covariance matrices for groups $k$ and $s$, respectively, $\boldsymbol{\Sigma}_k^*$ and $\boldsymbol{\Sigma}_s^*$ are the optimal covariance matrices for groups $k$ and $s$ obtained by solving the group-specific CovGraph problems.

### B.2.3   Fair BinNet

To simplify, denote

$$z_\theta = \exp \left( \theta_{jj} + \sum_{j' \neq j} \theta_{jj'} x_{ij'} \right), \quad \text{and} \quad z_\phi = \exp \left( \phi_{jj} + \sum_{j' \neq j} \phi_{jj'} x_{ij'} \right).$$

The gradients of the objectives of Fair BinNet that are utilized in the iterative update formula (21) are computed as follows:

$$(\nabla f_1(\boldsymbol{\Theta}))_{jj} = \left( \frac{\partial \mathcal{L}}{\partial \boldsymbol{\Theta}} (\boldsymbol{\Theta}, \mathbf{X}) \right)_{jj} = -(\mathbf{X}^T \mathbf{X})_{jj} + \sum_{i=1}^N \frac{z_\theta}{1 + z_\theta},$$

$$(\nabla f_1(\boldsymbol{\Theta}))_{jj'} = \left( \frac{\partial \mathcal{L}}{\partial \boldsymbol{\Theta}} (\boldsymbol{\Theta}, \mathbf{X}) \right)_{jj'} = -(\mathbf{X}^T \mathbf{X})_{jj'} + \sum_{i=1}^N \frac{x_{ij'} z_\theta}{1 + z_\theta},$$

$$\nabla f_{k+1}(\boldsymbol{\Theta}) = \sum_{s \in [K], s \neq k} ((\mathcal{L}(\boldsymbol{\Theta}; \mathbf{X}_k) - \mathcal{L}(\boldsymbol{\Theta}_k^*; \mathbf{X}_k)) - (\mathcal{L}(\boldsymbol{\Theta}; \mathbf{X}_s) \tag{26}$$

$$- \mathcal{L}(\boldsymbol{\Theta}_s^*; \mathbf{X}_s))) \left( \frac{\partial \mathcal{L}}{\partial \boldsymbol{\Theta}} (\boldsymbol{\Theta}, \mathbf{X}_k) - \frac{\partial \mathcal{L}}{\partial \boldsymbol{\Theta}} (\boldsymbol{\Theta}, \mathbf{X}_s) \right).$$

Here, $\mathcal{L}(\boldsymbol{\Theta}, \mathbf{X}) = f_1(\boldsymbol{\Theta})$ is the negative log-likelihood of the Ising model, where $\boldsymbol{\Theta} \in \mathbb{R}^{P \times P}$ is the interaction matrix, $\mathbf{X} \in \{0, 1\}^{N \times P}$ is the binary data matrix with $N$ samples and $P$ variables, and $\theta_{jj'}$ denotes the $(j, j')$-th element of $\boldsymbol{\Theta}$.

# C  Addendum to Section 3.3

## C.1  Auxiliary Lemmas

**Lemma 12.** *Let $\{\Theta^{(t)}\}$ be generated by Algorithm 1. Then for all $k = 1, \ldots, M$, we have*

$$F_k\left(\Theta^{(t+1)}\right) \leq F_k\left(\Theta^{(t)}\right). \tag{27}$$

*Proof.* Let $\varphi_\ell\left(\Phi, \Theta\right)$ be defined as in (9). Following the proof of [73, Lemma 4.1], we have

$$\varphi_\ell\left(\Theta^{(t+1)}, \Theta^{(t)}\right) \leq -\ell\|\Theta^{(t+1)} - \Theta^{(t)}\|_F^2. \tag{28}$$

If $\ell > L$, using the descent lemma [5, Proposition A.24], for all $k = 1, \ldots, M$, we obtain

$$F_k\left(\Theta^{(t+1)}\right) - F_k\left(\Theta^{(t)}\right) \leq \langle \nabla f_i\left(\Theta^{(t)}\right), \Theta^{(t+1)} - \Theta^{(t)}\rangle$$
$$+ g\left(\Theta^{(t+1)}\right) - g\left(\Theta^{(t)}\right) + \frac{\ell}{2}\|\Theta^{(t+1)} - \Theta^{(t)}\|_F^2. \tag{29}$$

Since the right-hand side of the above inequality is less than or equal to zero, it implies that

$$F_k\left(\Theta^{(t+1)}\right) \leq F_k\left(\Theta^{(t)}\right). \tag{30}$$

$\square$

**Lemma 13.** *Suppose Assumption A holds. Let $f_k$ and $g$ have convexity parameters $\mu_k \in \mathbb{R}_+$ and $\nu \in \mathbb{R}_+$, respectively, and define $\mu := \min_{k \in [M]} \mu_k$. Then, for all $\Theta \in \mathcal{M}$, we have*

$$\sum_{k=1}^{M} \rho_k^{(t)}\left(F_k\left(\Theta^{(t+1)}\right) - F_k(\Theta)\right) \leq \frac{\ell}{2}\left(\|\Theta^{(t)} - \Theta\|_F^2 - \|\Theta^{(t+1)} - \Theta\|_F^2\right)$$
$$- \frac{\nu}{2}\|\Theta^{(t+1)} - \Theta\|_F^2 - \frac{\mu}{2}\|\Theta^{(t)} - \Theta\|_F^2, \tag{31}$$

*where $\rho_k^{(t)}$ satisfies the following conditions:*

*1. There exists $\eta^{(t)} \in \partial g\left(\Theta^{(t+1)}\right)$ such that*

$$-\sum_{k=1}^{M} \rho_k^{(t)}\left(\nabla f_i\left(\Theta^{(t)}\right) + \eta^{(t)}\right) = \ell\left(\Theta^{(t+1)} - \Theta^{(t)}\right).$$

*2. $\boldsymbol{\rho}^{(t)} \in \mathcal{C}$ where $\mathcal{C}$ is defined in (13).*

*Proof.* Assumption A yields

$$F_k\left(\Theta^{(t+1)}\right) - F_k\left(\Theta^{(t)}\right) \leq \langle \nabla f_k(\Theta^{(t)}), \Theta^{(t+1)} - \Theta^{(t)}\rangle$$
$$+ g\left(\Theta^{(t+1)}\right) - g\left(\Theta^{(t)}\right) + \frac{\ell}{2}\|\Theta^{(t+1)} - \Theta^{(t)}\|_F^2. \tag{32}$$

From the convexity of $f_k$ and $g$, we have

$$F_k\left(\Theta^{(t+1)}\right) - F_k(\Theta)$$
$$= \left(F_k\left(\Theta^{(t+1)}\right) - F_k\left(\Theta^{(t)}\right)\right) + \left(F_k\left(\Theta^{(t)}\right) - F_k(\Theta)\right)$$
$$\leq \left(\langle \nabla f_k(\Theta^{(t)}), \Theta^{(t+1)} - \Theta^{(t)}\rangle + g\left(\Theta^{(t+1)}\right) - g\left(\Theta^{(t)}\right) + \frac{\ell}{2}\|\Theta^{(t+1)} - \Theta^{(t)}\|_F^2\right)$$
$$+ \left(\langle \nabla f_k(\Theta), \Theta^{(t)} - \Theta\rangle - \frac{\mu_i}{2}\|\Theta^{(t)} - \Theta\|_F^2 + g\left(\Theta^{(t)}\right) - g(\Theta)\right)$$
$$\leq \langle \nabla f_k(\Theta^{(t)}), \Theta^{(t+1)} - \Theta\rangle + g\left(\Theta^{(t+1)}\right)$$
$$- g(\Theta) - \frac{\mu}{2}\|\Theta^{(t)} - \Theta\|_F^2 + \frac{\ell}{2}\|\Theta^{(t+1)} - \Theta^{(t)}\|_F^2$$
$$\leq \langle \nabla f_k(\Theta^{(t)}) + \eta^{(t)}, \Theta^{(t+1)} - \Theta\rangle + \frac{\ell}{2}\|\Theta^{(t+1)}$$
$$- \Theta^{(t)}\|_F^2 - \frac{\mu}{2}\|\Theta^{(t)} - \Theta\|_F^2 - \frac{\nu}{2}\|\Theta^{(t+1)} - \Theta\|_F^2. \tag{33}$$

Condition 1 and Condition 2 yield

$$
\sum_{k=1}^{M} \rho_k^{(t)} \left( F_k \left( \boldsymbol{\Theta}^{(t+1)} \right) - F_k \left( \boldsymbol{\Theta} \right) \right) = \ell \langle \boldsymbol{\Theta}^{(t+1)} - \boldsymbol{\Theta}^{(t)}, \boldsymbol{\Theta}^{(t+1)} - \boldsymbol{\Theta} \rangle + \frac{\ell}{2} \| \boldsymbol{\Theta}^{(t+1)}
$$

$$
- \boldsymbol{\Theta}^{(t)} \|_F^2 - \frac{\mu}{2} \| \boldsymbol{\Theta}^{(t)} - \boldsymbol{\Theta} \|_F^2 - \frac{\nu}{2} \| \boldsymbol{\Theta}^{(t+1)} - \boldsymbol{\Theta} \|_F^2 \tag{34}
$$

$$
= \frac{\ell}{2} \left( \| \boldsymbol{\Theta}^{(t)} - \boldsymbol{\Theta} \|_F^2 - \| \boldsymbol{\Theta}^{(t+1)} - \boldsymbol{\Theta} \|_F^2 \right)
$$

$$
- \frac{\nu}{2} \| \boldsymbol{\Theta}^{(t+1)} - \boldsymbol{\Theta} \|_F^2 - \frac{\mu}{2} \| \boldsymbol{\Theta}^{(t)} - \boldsymbol{\Theta} \|_F^2.
$$

$\square$

### C.2    Proof of Theorem 6 for Fair GLasso

First, we present the convexity analysis and gradient Lipschitz continuity.

**Proposition 14** (Convexity of Fair GLasso). *Each $f_k$ for $k = 1, \ldots, M$ and $g$ defined in* (Fair GLasso) *of Fair GLasso are convex. Further, $f_1$ is strongly convex.*

*Proof.* First, consider $f_1$ in the first objective function of Fair GLasso:

$$
f_1(\boldsymbol{\Theta}) = -\log \det(\boldsymbol{\Theta}) + \operatorname{trace}(\mathbf{S}\boldsymbol{\Theta}), \tag{35}
$$

where $\boldsymbol{\Theta}$ is a positive definite matrix.

The gradient and Hessian of $f_1$ are, respectively:

$$
\nabla f_1(\boldsymbol{\Theta}) = \mathbf{S} - \boldsymbol{\Theta}^{-1}, \quad \mathbf{H}_{f_1} = \boldsymbol{\Theta}^{-1} \otimes \boldsymbol{\Theta}^{-1}. \tag{36}
$$

The positive definiteness of $\boldsymbol{\Theta}$ implies that $\boldsymbol{\Theta}^{-1}$ is also positive definite. Therefore, the Hessian $\mathbf{H}_{f_1}$, being the Kronecker product of $\boldsymbol{\Theta}^{-1}$ with itself, is positive definite. This establishes that the objective function $f_1$ is strongly convex.

Next, the functions $f_{k+1}$ for $k = 1, \ldots, M - 1$ are defined as:

$$
\begin{aligned}
f_{k+1}(\boldsymbol{\Theta}) &= \sum_{s \in [K], s \neq k} \phi \left( \mathcal{E}_k(\boldsymbol{\Theta}) - \mathcal{E}_s(\boldsymbol{\Theta}) \right) \\
&= \sum_{s \in [K], s \neq k} \frac{1}{2} \left( (\mathcal{L}(\boldsymbol{\Theta}; \mathbf{X}_k) - \mathcal{L}(\boldsymbol{\Theta}_k^*; \mathbf{X}_k)) - (\mathcal{L}(\boldsymbol{\Theta}; \mathbf{X}_s) - \mathcal{L}(\boldsymbol{\Theta}_s^*; \mathbf{X}_s)) \right)^2,
\end{aligned} \tag{37}
$$

which are convex due to the linearity of the trace operator in the loss function difference $\mathcal{L}(\boldsymbol{\Theta}; \mathbf{X}_k) - \mathcal{L}(\boldsymbol{\Theta}; \mathbf{X}_s) = \operatorname{trace}((\mathbf{S}_k - \mathbf{S}_s)\boldsymbol{\Theta})$, leading to a strong convexity parameter of 0.

In addition, $g(\boldsymbol{\Theta}) = \lambda \|\boldsymbol{\Theta}\|_1$ is identified as a closed, proper, and convex function. $\square$

**Proposition 15** (Gradient Lipschitz Continuity of Fair GLasso). *The gradients of $f_k$ for $k = 1, \ldots, M$ defined in* (Fair GLasso) *are Lipschitz continuous.*

*Proof.* First, we present the gradient and Hessian of functions $f_1$ and $\{f_{k+1}\}_{k=1}^{M-1}$ as follows:

$$
\begin{aligned}
f_1(\boldsymbol{\Theta}) = \quad & -\log \det(\boldsymbol{\Theta}) + \operatorname{trace}(\mathbf{S}\boldsymbol{\Theta}), \quad \nabla f_1(\boldsymbol{\Theta}) = \mathbf{S} - \boldsymbol{\Theta}^{-1}, \quad \mathbf{H}_{f_1} = \boldsymbol{\Theta}^{-1} \otimes \boldsymbol{\Theta}^{-1}; \\
f_{k+1}(\boldsymbol{\Theta}) = & \sum_{s \in [K], s \neq k} \frac{1}{2} \left( \operatorname{trace}(\mathbf{S}_k \boldsymbol{\Theta}) - \operatorname{trace}(\mathbf{S}_s \boldsymbol{\Theta}) + \log \det(\boldsymbol{\Theta}_k^*) \right. \\
& \left. - \operatorname{trace}(\mathbf{S}_k \boldsymbol{\Theta}_k^*) - \log \det(\boldsymbol{\Theta}_s^*) + \operatorname{trace}(\mathbf{S}_s \boldsymbol{\Theta}_s^*) \right)^2, \\
\nabla f_{k+1}(\boldsymbol{\Theta}) = & \sum_{s \in [K], s \neq k} \left( \operatorname{trace}(\mathbf{S}_k \boldsymbol{\Theta}) - \operatorname{trace}(\mathbf{S}_s \boldsymbol{\Theta}) + \log \det(\boldsymbol{\Theta}_k^*) \right. \\
& \left. - \operatorname{trace}(\mathbf{S}_k \boldsymbol{\Theta}_k^*) - \log \det(\boldsymbol{\Theta}_s^*) + \operatorname{trace}(\mathbf{S}_s \boldsymbol{\Theta}_s^*) \right) (\mathbf{S}_k - \mathbf{S}_s), \\
\mathbf{H}_{f_{k+1}}(\boldsymbol{\Theta}) = & \sum_{s \in [K], s \neq k} (\mathbf{S}_k - \mathbf{S}_s) \otimes (\mathbf{S}_k - \mathbf{S}_s).
\end{aligned}
$$

$$\tag{38}$$

Define $L_1 = \Lambda_{\max}(\mathbf{H}_{f_1})$ and $L_{k+1} = \Lambda_{\max}(\mathbf{H}_{f_{k+1}})$ for $k = 1, \ldots, M-1$. Given that $\{f_k\}_{k=1}^M$ are convex (as proven in Proposition 14) and twice differentiable, their gradients satisfy Lipschitz continuity with Lipschitz constants $\{L_k\}_{k=1}^M$. $\square$

Next, we present the proof for Theorem 6.

*proof for Theorem 6.* From Proposition 14 and Proposition 15, convexity and gradient Lipschitz continuity of objective functions $\{f_k\}_{k=1}^M$ are verified. Hence, Assumption A holds.

From Lemma 13 and the convexity of $f_i$ and $g$, for all $\boldsymbol{\Theta} \in \mathcal{M}$, we obtain

$$\sum_{k=1}^M \rho_k^{(t)} \left( F_k\left(\boldsymbol{\Theta}^{(t+1)}\right) - F_k(\boldsymbol{\Theta}) \right) \leq \frac{\ell}{2} \left( \|\boldsymbol{\Theta}^{(t)} - \boldsymbol{\Theta}\|_F^2 - \|\boldsymbol{\Theta}^{(t+1)} - \boldsymbol{\Theta}\|_F^2 \right). \tag{39}$$

Adding up the above inequality (39) from $t = 0$ to $t = \tilde{t}$, we have

$$\sum_{t=0}^{\tilde{t}} \sum_{k=1}^M \rho_k^{(t)} \left( F_k\left(\boldsymbol{\Theta}^{(t+1)}\right) - F_k(\boldsymbol{\Theta}) \right) \leq \frac{\ell}{2} \left( \|\boldsymbol{\Theta}^{(0)} - \boldsymbol{\Theta}\|_F^2 - \|\boldsymbol{\Theta}^{(\tilde{t}+1)} - \boldsymbol{\Theta}\|_F^2 \right) \tag{40}$$

$$\leq \frac{\ell}{2} \|\boldsymbol{\Theta}^{(0)} - \boldsymbol{\Theta}\|_F^2.$$

Lemma 12 implies that $F_k\left(\boldsymbol{\Theta}^{(\tilde{t}+1)}\right) \leq F_k\left(\boldsymbol{\Theta}^{(t+1)}\right)$ for all $t \leq \tilde{t}$ and

$$\sum_{t=0}^{\tilde{t}} \sum_{k=1}^M \rho_k^{(t)} \left( F_k\left(\boldsymbol{\Theta}^{(\tilde{t}+1)}\right) - F_k(\boldsymbol{\Theta}) \right) \leq \frac{\ell}{2} \|\boldsymbol{\Theta}^{(0)} - \boldsymbol{\Theta}\|_F^2. \tag{41}$$

Let $\bar{\rho}_k^{\tilde{t}} := \sum_{t=0}^{\tilde{t}} \rho_k^{(t)} / (\tilde{t}+1)$. Then, it follows that

$$\sum_{k=1}^M \bar{\rho}_k^{\tilde{t}} \left( F_k\left(\boldsymbol{\Theta}^{(\tilde{t}+1)}\right) - F_k(\boldsymbol{\Theta}) \right) \leq \frac{\ell}{2(\tilde{t}+1)} \|\boldsymbol{\Theta}^{(0)} - \boldsymbol{\Theta}\|_F^2. \tag{42}$$

Since $\bar{\rho}_k^{\tilde{t}} \geq 0$ and $\sum_{k=1}^M \bar{\rho}_k^{\tilde{t}} = 1$, we can conclude that

$$\min_{k \in [M]} \left( F_k\left(\boldsymbol{\Theta}^{(\tilde{t}+1)}\right) - F_k(\boldsymbol{\Theta}) \right) \leq \frac{\ell}{2(\tilde{t}+1)} \|\boldsymbol{\Theta}^{(0)} - \boldsymbol{\Theta}\|_F^2. \tag{43}$$

Now, following the proof of [75, Theorem 5.1] and using Assumption B, we obtain

$$\sup_{\mathbf{F}^* \in \mathbf{F}\left(\mathcal{N}^* \cap \Omega_{\mathbf{F}}\left(\mathbf{F}\left(\boldsymbol{\Theta}^{(0)}\right)\right)\right)} \inf_{\boldsymbol{\Theta} \in \mathbf{F}^{-1}(\{\mathbf{F}^*\})} \min_{k \in [M]} \left( F_k\left(\boldsymbol{\Theta}^{(\tilde{t}+1)}\right) - F_k(\boldsymbol{\Theta}) \right) \leq \frac{\ell R}{2(\tilde{t}+1)}, \tag{44}$$

$$\sup_{\mathbf{F}^* \in \mathbf{F}\left(\mathcal{N}^* \cap \Omega_{\mathbf{F}}\left(\mathbf{F}\left(\boldsymbol{\Theta}^{(0)}\right)\right)\right)} \min_{k \in [M]} \left( F_k\left(\boldsymbol{\Theta}^{(\tilde{t}+1)}\right) - F_k^* \right) \leq \frac{\ell R}{2(\tilde{t}+1)}, \tag{45}$$

$$\sup_{\boldsymbol{\Theta} \in \mathcal{N}^* \cap \Omega_{\mathbf{F}}\left(\mathbf{F}\left(\boldsymbol{\Theta}^{(0)}\right)\right)} \min_{k \in [M]} \left( F_k\left(\boldsymbol{\Theta}^{(\tilde{t}+1)}\right) - F_k(\boldsymbol{\Theta}) \right) \leq \frac{\ell R}{2(\tilde{t}+1)}. \tag{46}$$

The inequality $F_k\left(\boldsymbol{\Theta}^{(t)}\right) \leq F_k\left(\boldsymbol{\Theta}^{(0)}\right)$ from Lemma 12 implies that

$$\sup_{\boldsymbol{\Theta} \in \Omega_{\mathbf{F}}\left(\mathbf{F}\left(\boldsymbol{\Theta}^{(0)}\right)\right)} \min_{k \in [M]} \left( F_k\left(\boldsymbol{\Theta}^{(\tilde{t}+1)}\right) - F_k(\boldsymbol{\Theta}) \right) = \sup_{\boldsymbol{\Theta} \in \Omega_{\mathbf{F}}\left(\mathbf{F}\left(\boldsymbol{\Theta}^{\tilde{t}+1}\right)\right)} \min_{k \in [M]} \left( F_k\left(\boldsymbol{\Theta}^{(\tilde{t}+1)}\right) - F_k(\boldsymbol{\Theta}) \right)$$

$$= \sup_{\boldsymbol{\Theta} \in \mathcal{M}} \min_{k \in [M]} \left( F_k\left(\boldsymbol{\Theta}^{(\tilde{t}+1)}\right) - F_k(\boldsymbol{\Theta}) \right). \tag{47}$$

Moreover, from Assumption B that for all $\boldsymbol{\Theta} \in \Omega_{\mathbf{F}}\left(\mathbf{F}\left(\boldsymbol{\Theta}^{(0)}\right)\right)$, there exists $\boldsymbol{\Theta}^* \in \mathcal{N}^*$ such that $\mathbf{F}(\boldsymbol{\Theta}^*) \preceq \mathbf{F}(\boldsymbol{\Theta})$, it follows:

$$\sup_{\boldsymbol{\Theta} \in \mathcal{N}^* \cap \Omega_{\mathbf{F}}\left(\mathbf{F}\left(\boldsymbol{\Theta}^{(0)}\right)\right)} \min_{k \in [M]} \left( F_k\left(\boldsymbol{\Theta}^{(\tilde{t}+1)}\right) - F_k(\boldsymbol{\Theta}) \right)$$

$$= \sup_{\boldsymbol{\Theta} \in \Omega_{\mathbf{F}}\left(\mathbf{F}\left(\boldsymbol{\Theta}^{(0)}\right)\right)} \min_{k \in [M]} \left( F_k\left(\boldsymbol{\Theta}^{(\tilde{t}+1)}\right) - F_k(\boldsymbol{\Theta}) \right). \tag{48}$$

Therefore, from (44) and (48), we can conclude that

$$\sup_{\boldsymbol{\Theta} \in \mathcal{M}} \min_{k \in [M]} \{F_k \left(\boldsymbol{\Theta}^{(\tilde{t}+1)}\right) - F_k (\boldsymbol{\Theta})\} \leq \frac{\ell R}{2 (\tilde{t} + 1)}. \tag{49}$$

$\square$

### C.3 Proof of Theorem 7 for Fair CovGraph

First, we present the convexity analysis and gradient Lipschitz continuity.

**Proposition 16** (Convexity of Fair CovGraph). *Through incorporating a convex regularization term $\gamma_C \|\boldsymbol{\Theta}\|_F^2$ for some $\gamma_C \geq \max\{0, -\Lambda_{\min}(\nabla^2 f_k(\boldsymbol{\Theta}))\}$ into each $f_k$ for $k = 2, \ldots, M$, each $f_k$ for $k = 1, \ldots, M$ and $g$ defined in the multi-objective optimization problem* (Fair CovGraph) *of Fair CovGraph are guaranteed to be convex. In particular, $f_1$ is strongly convex.*

*Proof.* In Fair CovGraph (Fair CovGraph), the function $f_1$, its gradient, and Hessian are defined as follows:

$$f_1(\boldsymbol{\Sigma}) = \frac{1}{2}\|\boldsymbol{\Sigma} - \mathbf{S}\|_F^2 - \tau \log \det (\boldsymbol{\Sigma}),$$

$$\nabla f_1(\boldsymbol{\Sigma}) = \boldsymbol{\Sigma} - \mathbf{S} - \tau \boldsymbol{\Sigma}^{-1}, \quad \mathbf{H}_{f_1} = \boldsymbol{I}_{P^2} + \tau \boldsymbol{\Sigma}^{-1} \otimes \boldsymbol{\Sigma}^{-1}.$$

The positive definiteness of the covariance matrix $\boldsymbol{\Sigma}$ guarantees that the Hessian matrix $\mathbf{H}_{f_1}$ is also positive definite, establishing the strong convexity of the function $f_1$. Next, consider:

$$\mathcal{L}(\boldsymbol{\Sigma}; \mathbf{X}_k) - \mathcal{L}(\boldsymbol{\Sigma}; \mathbf{X}_s) = \frac{1}{2}\|\boldsymbol{\Sigma} - \mathbf{S}_k\|_F^2 - \frac{1}{2}\|\boldsymbol{\Sigma} - \mathbf{S}_s\|_F^2. \tag{50}$$

This difference is necessarily convex, as it is the difference between two convex functions.

To ensure the convexity of the functions $f_k$ for $k = 2, \ldots, M$, a convexity regularization term $\gamma_C \|\boldsymbol{\Theta}\|_F^2$ is added to $f_k$, denoted by $\tilde{f}_k$, where $\gamma_C$ is chosen to be $\gamma_C \geq \max\{0, -\Lambda_{\min}(\nabla^2 f_k(\boldsymbol{\Theta}))\}$ such that $\Lambda_{\min}(\mathbf{H}_{\tilde{f}_k}) \geq 0$ for $k = 2, \ldots, M$. This regularization term guarantees that the minimum eigenvalue of the Hessian matrix $\mathbf{H}_{\tilde{f}_k}$ is non-negative, thereby ensuring the convexity of $f_k$. Furthermore, the function $g(\boldsymbol{\Sigma})$ is a closed, proper, and convex function. $\square$

**Proposition 17** (Gradient Lipschitz Continuity of Fair CovGraph). *The gradients of $f_k$ for $k = 1, \ldots, M$ defined in* (Fair CovGraph) *are Lipschitz continuous.*

*Proof.* We detail the gradient and Hessian of functions $f_1$ and $\{f_{k+1}\}_{k=1}^{M-1}$ as follows:

$$\begin{aligned} f_1(\boldsymbol{\Sigma}) =& \frac{1}{2}\|\boldsymbol{\Sigma} - \mathbf{S}\|_F^2 - \tau \log \det (\boldsymbol{\Sigma}), \\ \nabla f_1(\boldsymbol{\Sigma}) =& \boldsymbol{\Sigma} - \mathbf{S} - \tau \boldsymbol{\Sigma}^{-1}, \quad \mathbf{H}_{f_1} = \boldsymbol{I}_{P^2} + \tau \boldsymbol{\Sigma}^{-1} \otimes \boldsymbol{\Sigma}^{-1}, \\ f_{k+1}(\boldsymbol{\Sigma}) =& \sum_{s \in [K], s \neq k} \frac{1}{2} \left( \frac{1}{2}\|\boldsymbol{\Sigma} - \mathbf{S}_k\|_F^2 - \frac{1}{2}\|\boldsymbol{\Sigma} - \mathbf{S}_s\|_F^2 + \tau \log \det(\boldsymbol{\Sigma}_k^*) \right. \\ & \left. - \frac{1}{2}\|\boldsymbol{\Sigma}_k^* - \mathbf{S}_k\|_F^2 - \tau \log \det(\boldsymbol{\Sigma}_s^*) + \frac{1}{2}\|\boldsymbol{\Sigma}_s^* - \mathbf{S}_s\|_F^2 \right)^2, \\ \nabla f_{k+1}(\boldsymbol{\Sigma}) =& \sum_{s \in [K], s \neq k} \left( \frac{1}{2}\|\boldsymbol{\Sigma} - \mathbf{S}_k\|_F^2 - \frac{1}{2}\|\boldsymbol{\Sigma} - \mathbf{S}_s\|_F^2 + \tau \log \det(\boldsymbol{\Sigma}_k^*) \right. \\ & \left. - \frac{1}{2}\|\boldsymbol{\Sigma}_k^* - \mathbf{S}_k\|_F^2 - \tau \log \det(\boldsymbol{\Sigma}_s^*) + \frac{1}{2}\|\boldsymbol{\Sigma}_s^* - \mathbf{S}_s\|_F^2 \right) (\mathbf{S}_s - \mathbf{S}_k), \\ \mathbf{H}_{f_{k+1}}(\boldsymbol{\Sigma}) =& \sum_{s \in [K], s \neq k} (\mathbf{S}_s - \mathbf{S}_k) \otimes (\mathbf{S}_s - \mathbf{S}_k). \end{aligned} \tag{51}$$

Then given that $f_1$ and $\frac{\partial f_1}{\partial \boldsymbol{\Sigma}}$ are Lipschitz continuous and bounded on the set $\{\boldsymbol{\Sigma} \in \mathcal{M} | \|\boldsymbol{\Sigma}\|_1 < \infty\}$, the function sequence $\{f_{k+1}\}_{k=1}^{M-1}$ is also Lipschitz continuous. $\square$

*Proof of Theorem 7.* Note that for

$$\gamma_C \geq \max\{0, -\Lambda_{\min}(\nabla^2 f_k(\boldsymbol{\Sigma}))\}, \tag{52}$$

the problem (Fair CovGraph) is convex. Now, from Proposition 16 and Proposition 17, convexity and gradient Lipschitz continuity of objective functions $\{f_k\}_{k=1}^M$ are verified. Then, the proof of Theorem 7 is a slightly modified version of the proof of Theorem 6. □

### C.4 Proof of Theorem 8 for Fair BinNet

First, we present the convexity analysis and gradient Lipschitz continuity.

**Proposition 18** (Convexity of Fair BinNet)**.** *In the multi-objective optimization Problem (Fair BinNet), the functions $f_1$ and $g$ are convex. Furthermore, by incorporating a convex regularization term $\gamma_I \|\boldsymbol{\Theta}\|_F^2$ for some $\gamma_I \geq |\min\{\frac{1}{2}\Lambda_{\min}(\nabla^2 f_k), 0\}|$ into each $f_k$ for $k = 2, \ldots, M$, the set of functions $\{f_k\}_{k=2}^M$ are ensured to be convex as well.*

*Proof.* The function $f_1$ for Fair BinNet is defined as:

$$f_1(\boldsymbol{\Theta}) = -\sum_{j=1}^P \sum_{j'=1}^P \theta_{jj'}(\mathbf{X}^T\mathbf{X})_{jj'} + \sum_{i=1}^N \sum_{j=1}^P \log\left(1 + \exp\left(\theta_{jj} + \sum_{j'\neq j} \theta_{jj'} x_{ij'}\right)\right). \tag{53}$$

To demonstrate the convexity of $f_1$, observe that $h(x) = \log(1+\exp(x))$ is convex and nondecreasing. Since convexity is preserved under linear combination and summation, $f_1$ is convex by construction. Also, $g(\boldsymbol{\Sigma})$ is a closed, proper, and convex function.

Consider $\tilde{f}_{k+1}(\boldsymbol{\Theta}) = f_{k+1}(\boldsymbol{\Theta}) + \gamma_I\|\boldsymbol{\Theta}\|_F^2$ for $k = 1, \ldots, M-1$, its Hessian matrix is given by:

$$\mathbf{H}_{\tilde{f}_{k+1}}(\boldsymbol{\Theta}) = \mathbf{H}_{f_{k+1}}(\boldsymbol{\Theta}) + 2\gamma_I \mathbf{I}_{P^2}. \tag{54}$$

If $\gamma_I$ is chosen to be $|\min\{\frac{1}{2}\Lambda_{\min}(\nabla^2 f_k), 0\}|$ such that $\gamma_I \mathbf{I}_{P^2}$ dominates any negative curvature in $\mathbf{H}_{f_{k+1}}(\boldsymbol{\Theta})$, then $\mathbf{H}_{\tilde{f}_{k+1}}(\boldsymbol{\Theta})$ will be positive semidefinite, leading the convexity of $\{\tilde{f}_{k+1}\}_{k=1}^M$.

□

**Proposition 19** (Gradient Lipschitz Continuity of Fair BinNet)**.** *The gradients of $f_k$ for $k = 1, \ldots, M$ defined in the multi-objective optimization Problem (Fair BinNet) are Lipschitz continuous.*

*Proof.* For notational simplicity, we introduce the following substitutions: utilize $\mathcal{L}(\boldsymbol{\Theta}, \mathbf{X})$ in place of $f_1(\boldsymbol{\Theta})$, denote $z_\theta = \exp\left(\theta_{jj} + \sum_{j'\neq j} \theta_{jj'} x_{ij'}\right)$, $z_\phi = \exp\left(\phi_{jj} + \sum_{j'\neq j} \phi_{jj'} x_{ij'}\right)$. Then, we proceed to evaluate the gradient of the function $f_1$ in the context of Fair BinNet as follows:

$$\begin{aligned}
\left(\frac{\partial\mathcal{L}}{\partial\boldsymbol{\Theta}}(\boldsymbol{\Theta}, \mathbf{X})\right)_{jj} &= (\nabla f_1(\boldsymbol{\Theta}))_{jj} = -(\mathbf{X}^T\mathbf{X})_{jj} + \sum_{i=1}^N \frac{z_\theta}{1 + z_\theta}, \\
\left(\frac{\partial\mathcal{L}}{\partial\boldsymbol{\Theta}}(\boldsymbol{\Theta}, \mathbf{X})\right)_{jj'} &= (\nabla f_1(\boldsymbol{\Theta}))_{jj'} = -(\mathbf{X}^T\mathbf{X})_{jj'} + \sum_{i=1}^N \frac{x_{ij'} z_\theta}{1 + z_\theta}.
\end{aligned} \tag{55}$$

Given that $h_1(x) = \exp(x)/(1 + \exp(x))$ is Lipschitz continuous with Lipschitz constant $0.25$, for any $\boldsymbol{\Theta}, \boldsymbol{\Phi} \in \mathcal{M}$,

$$\begin{aligned}
\|\nabla f_1(\boldsymbol{\Theta}) - \nabla f_1(\boldsymbol{\Phi})\|_F &\leq \sum_{j=1}^P \sqrt{\left(\sum_{i=1}^N \frac{z_\theta}{1 + z_\theta} - \sum_{i=1}^N \frac{z_\phi}{1 + z_\phi}\right)^2} \\
&\quad + \sum_{j=1}^P \sum_{j'=1, j'\neq j}^P \sqrt{\left(\sum_{i=1}^N \frac{x_{ij'} z_\theta}{1 + z_\theta} - \sum_{i=1}^N \frac{x_{ij'} z_\phi}{1 + z_\phi}\right)^2} \\
&\leq \sum_{j=1}^P \sum_{i=1}^N \left|\frac{z_\theta}{1 + z_\theta} - \frac{z_\phi}{1 + z_\phi}\right| + \sum_{j=1}^P \sum_{j'=1, j'\neq j}^P \sum_{i=1}^N \left|\frac{x_{ij'} z_\theta}{1 + z_\theta} - \frac{x_{ij'} z_\phi}{1 + z_\phi}\right|
\end{aligned}$$

$$\leq \sum_{j=1}^{P} \sum_{i=1}^{N} \left| \theta_{jj} - \phi_{jj} + \sum_{j' \neq j} (\theta_{jj'} - \phi_{jj'}) x_{ij'} \right|$$

$$+ (P-1) \times \sum_{j=1}^{P} \sum_{i=1}^{N} \left| \theta_{jj} - \phi_{jj} + \sum_{j' \neq j} (\theta_{jj'} - \phi_{jj'}) x_{ij'} \right|$$

$$\leq N \times P \times \sum_{j=1}^{P} \sum_{j'=1}^{P} |\theta_{jj'} - \phi_{jj'}|$$

$$\leq N \times P^2 \times \sqrt{\sum_{j=1}^{P} \sum_{j'=1}^{P} |\theta_{jj'} - \phi_{jj'}|^2}$$

$$= N \times P^2 \times \|\mathbf{\Theta} - \mathbf{\Phi}\|_F. \tag{56}$$

It follows that there exists $L_1 = N \times P^2 \in \mathbb{R}$ such that $\|\nabla f_1(\mathbf{\Theta}) - \nabla f_1(\mathbf{\Phi})\|_F \leq L_1 \|\mathbf{\Theta} - \mathbf{\Phi}\|_F$.

Subsequently, the gradients of functions $\{f_{k+1}\}_{k=1}^{M-1}$ in Fair BinNet are evaluated as:

$$\nabla f_{k+1}(\mathbf{\Theta}) = \sum_{s \in [K], s \neq k} ((\mathcal{L}(\mathbf{\Theta}; \mathbf{X}_k) - \mathcal{L}(\mathbf{\Theta}_k^*; \mathbf{X}_k))$$

$$- (\mathcal{L}(\mathbf{\Theta}; \mathbf{X}_s) - \mathcal{L}(\mathbf{\Theta}_s^*; \mathbf{X}_s))) \left( \frac{\partial \mathcal{L}}{\partial \mathbf{\Theta}}(\mathbf{\Theta}, \mathbf{X}_k) - \frac{\partial \mathcal{L}}{\partial \mathbf{\Theta}}(\mathbf{\Theta}, \mathbf{X}_s) \right). \tag{57}$$

Then given that $\mathcal{L}$ and $\frac{\partial \mathcal{L}}{\partial \mathbf{\Theta}}$ are Lipschitz continuous and bounded on the set $\{\mathbf{\Theta} \in \mathcal{M} | \|\mathbf{\Theta}\|_1 < \infty\}$, the function sequence $\{f_{k+1}\}_{k=1}^{M-1}$ is also Lipschitz continuous. $\qquad \square$

*Proof of Theorem 8 for Fair BinNet.* Building on Proposition 18 and Proposition 19, proof of Theorem 8 can be viewed as a nuanced adaptation of the proof presented in Theorem 6. $\qquad \square$

### C.5 Computational Complexity of FairGMs

The computational complexity of the fair GLasso and fair CovGraph algorithm depends on both the number of variables $P$ and the number of observations $N$. We aim to demonstrate that our algorithm has a complexity of $O\left(\frac{\max(NP^2, P^3)}{\epsilon}\right)$, which is similar to standard graph learning methods when $K << N, P$. This is applicable to our experimental results, where $K = 2, 8, 2,000 \leq N \leq 15,000$, and $5 \leq P \leq 120$. The computational complexity is primarily influenced by the following factors:

1. Number of Variables ($P$): The complexity scales as $O(P^3)$ due to matrix inversion for computing the gradient at each step.

2. Number of Observations ($N$): Computing the empirical covariance matrix from the data has a complexity of $O(NP^2)$.

3. Global Fair GMs Complexity: Considering factors 1 and 2, the complexity of each proximal gradient step applied to global fair GM is $O(\max(NP^2, P^3))$.

4. Local GMs Complexity: Applying factors 1 and 2 to group-specific data, the complexity of each local GM is $\max(N_k P^2, P^3)$ for all $k = 1, \ldots, K$. The total complexity of the local GMs is $\sum_{k=1}^{K} \max(N_k P^2, P^3)$.

As established in Theorem 6 and Theorem 8 for fair inverse covariance and covariance estimation, the iteration complexity of our algorithm to achieve $\epsilon$-accuracy is $O\left(\frac{1}{\epsilon}\right)$. Combining this result with the per-iteration complexity of the algorithm, the total time complexity of our optimization procedure is $O\left(\frac{\max(NP^2, P^3)}{\epsilon}\right)$. Including the Local GMs computation, the total time complexity of fair GMs is $O\left(\sum_{k=1}^{K} \max(N_k P^2, P^3) + \frac{\max(NP^2, P^3)}{\epsilon}\right)$.

Under the assumption that the number of groups is small (i.e., $K << N$, $K << P$, and $K << 1/\epsilon$), the complexity reduces to $O\left(\frac{\max(NP^2, P^3)}{\epsilon}\right)$. This complexity is of the same order as the complexity of running the proximal gradient method applied to covariance estimation and inverse covariance

estimation. Therefore, for large $N$ and $P$ and a small number of groups, the time complexity of our algorithm is comparable to the standard method. In addition to theoretical analysis, we also provide sensitivity analysis experiments on $P$, $N$, $K$, and group imbalance in Appendix D.6-D.9.

# D Addendum to Section 4

## D.1 Iterative Soft-Thresholding Algorithm (ISTA)

ISTA for sparse inverse covariance estimation is initially introduced by [60] and demonstrates a closed-form linear convergence rate. We adapt this approach and extend it to other GMs, utilizing it in the generation of both baseline and local graphs. Specifically, for a GM characterized by the loss function $\mathcal{L}(\boldsymbol{\Theta};\mathbf{X}) + \lambda\|\boldsymbol{\Theta}\|_1$, we employ the following detailed algorithm:

---

**Algorithm 2** ISTA for GMs

---

**Input**: Sample matrix $\mathbf{X}$, initial iterate $\boldsymbol{\Theta}^{(0)}$, maximum iteration $T$, step size $\zeta$, regularization parameter $\lambda$, tolerance $\epsilon$. Set $t = 0$.
**for** $t = 0, 1, \ldots, T-1$ **do**

   **Gradient Step**: $\boldsymbol{\Theta}^{(t+1)} \leftarrow \boldsymbol{\Theta}^{(t)} - \zeta\nabla\mathcal{L}\left(\boldsymbol{\Theta}^{(t)};\mathbf{X}\right)$

   **Soft-Thresholding Step**: $\boldsymbol{\Theta}^{(t+1)} \leftarrow \eta_{\zeta\rho}(\boldsymbol{\Theta}^{(t+1)}), \quad (\eta_{\zeta\rho}(\boldsymbol{\Theta}))_{jj'} = \text{sign}(\theta_{jj'})\max(|\theta_{jj'}| - \zeta\rho, 0)$

   **if** $\|\nabla\mathcal{L}\left(\boldsymbol{\Theta}^{(t+1)}, \mathbf{X}\right)\|_1 \leq \epsilon$ **then**
      Break
   **end if**
**end for**
**Output**: $\boldsymbol{\Theta}^{(t+1)}$

---

## D.2 Simulation Study of Fair GLasso

As a supplement to Section 4.2, we detail the process of generating $K$ block diagonal covariance matrices of dimensions $P \times P$, denoted as $\{\boldsymbol{\Sigma}_k\}_{k=1}^{K}$, each corresponding to distinct sensitive groups. The procedure is as follows:

1. Firstly, we assume that each $\boldsymbol{\Sigma}_k$ contains $Q$ blocks and $P$ is divisible by $Q$. For the first group, the covariance matrix $\boldsymbol{\Sigma}_1$ is constructed as a block diagonal matrix:

$$\boldsymbol{\Sigma}_1 = \begin{pmatrix} \mathbf{B}_1 & \cdots & \mathbf{0} \\ \vdots & \ddots & \vdots \\ \mathbf{0} & \cdots & \mathbf{B}_Q \end{pmatrix}, \tag{58}$$

   Here, each block $\mathbf{B}_q$ is a sub-matrix filled with values drawn from a normal distribution $\mathcal{N}(0.7, 0.2)$.

2. To ensure $\boldsymbol{\Sigma}_1$ is symmetric, it is adjusted to $(\boldsymbol{\Sigma}_1 + \boldsymbol{\Sigma}_1^{\top})/2$. To ensure it is positive definite, we further adjust $\boldsymbol{\Sigma}_1$ as:

$$\boldsymbol{\Sigma}_1 = \begin{bmatrix} \boldsymbol{v}_1 & \cdots & \boldsymbol{v}_P \end{bmatrix} \begin{bmatrix} \hat{\lambda}_1 & \cdots & 0 \\ \vdots & \ddots & \vdots \\ 0 & \cdots & \hat{\lambda}_P \end{bmatrix} \begin{bmatrix} \boldsymbol{v}_1^{\top} \\ \vdots \\ \boldsymbol{v}_P^{\top} \end{bmatrix}, \tag{59}$$

   where $\hat{\lambda}_j$ represents $\max(\lambda_j(\boldsymbol{\Sigma}_1), 10^{-5})$, and $\boldsymbol{v}_j$ is the corresponding eigenvector.

3. For each subsequent group ($k = 2, \ldots, K$), the covariance matrix $\boldsymbol{\Sigma}_k$ is initially set equal to $\boldsymbol{\Sigma}_{k-1}$. Then, two (one for sensitivity analysis) of its sub-matrices, which have not been altered yet, are reset to the identity matrix.

## D.3 Simulation Study of Fair BinNet

We specify the process of generating synthetic data for the simulation study in Section 4.3. This process adapts a hub node-based network as proposed by [72], aiming to generate a sequence of networks $\{\boldsymbol{\Theta}\}_{l=1}^{k}$. The process comprises the following steps:

Table 4: Outcomes in terms of the value of the objective function ($F_1$), the summation of the pairwise graph disparity error ($\Delta$), and the average computation time in seconds ($\pm$ standard deviation) from 10 repeated experiments. "$\downarrow$" means the smaller, the better, and the best value is in bold. These experiments are conducted on an Apple M2 Pro processor. Note that both $F_1$ and $\Delta$ are deterministic.

| Dataset | $F_1 \downarrow$ | | $\%F_1 \uparrow$ | $\Delta \downarrow$ | | $\%\Delta \uparrow$ | Runtime $\downarrow$ | |
|---|---|---|---|---|---|---|---|---|
| | GM | Fair GM | | GM | Fair GM | | GM | Fair GM |
| AV45 | **79.201** | 79.611 | -0.52% | 8.7626 | **3.4162** | +61.01% | **0.548 ($\pm$ 0.06)** | 19.06 ($\pm$ 0.3) |
| AV1451 | **66.493** | 66.923 | -0.65% | 8.0503 | **2.8920** | +64.08% | **1.616 ($\pm$ 0.65)** | 36.00 ($\pm$ 2.7) |

1. Initialize a $P \times P$ matrix $\mathbf{A}$, setting $\mathbf{A}_{jj'} = 1$ with a probability of 0.01 for all $j < j'$ and $\mathbf{A}_{jj'} = 0$ otherwise. Ensure the matrix is symmetric by assigning $\mathbf{A}_{j'j} = \mathbf{A}_{jj'}$. From the set of nodes, randomly select $H$ hub nodes and modify their corresponding rows and columns in $\mathbf{A}$ to 1 with a 99% probability or to 0 otherwise.

2. Construct another $P \times P$ matrix $\mathbf{E}$, where each element $\mathbf{E}_{jj'}$ is i.i.d.. Set $\mathbf{E}_{jj'} = 0$ if $\mathbf{A}_{jj'} = 0$. Otherwise, draw $\mathbf{E}_{jj'}$ from a uniform distribution over the intervals $[-0.75, -0.25] \cup [0.25, 0.75]$ for hub node columns and rows, and $[-0.5, -0.25] \cup [0.25, 0.5]$ for non-hub node columns and rows. Subsequently, symmetrize matrix $\mathbf{E}$ by computing $\mathbf{E} = (\mathbf{E} + \mathbf{E}^\top)/2$. Define the first network $\mathbf{\Theta}_1$ as $\mathbf{\Theta}_1 = \mathbf{E} + (0.1 - \lambda_{\min}(\mathbf{E}))\mathbf{I}$.

3. For the generation of each subsequent network ($k = 2, \ldots, K$), start with the preceding network, setting $\mathbf{\Theta}_k = \mathbf{\Theta}_{k-1}$. Then, modify $\mathbf{\Theta}_k$ by eliminating two hub nodes.

## D.4   Addendum to Subsection 4.5

In the experiments of applying GLasso to the ADNI dataset, we investigate the influence of sensitive attributes on brain networks associated with Alzheimer's disease (AD) pathology. Specifically, we focus on the amyloid accumulation network using AV45 (florbetapir) positron emission tomography (PET) data [88] and the tau accumulation network using AV1451 (flortaucipir) PET data [46]. For the amyloid network, we consider the sensitive attribute of marital status, as previous studies suggest that marriage may affect the progression of dementia due to factors such as social support, cognitive stimulation, and lifestyle habits [21, 61]. The dataset is divided into two groups based on marital status: a single group with 52 samples and a married group with 1,018 samples, creating an imbalanced and high-dimensional setting that poses challenges for network estimation. In the tau accumulation network, we explore the impact of the sensitive attribute race, which separates the dataset into two groups: the white group with 755 samples and the non-white group with 118 samples. This division allows us to investigate potential disparities in tau pathology across racial groups. Throughout the experiments, the regularization parameter $\lambda$, which controls the sparsity of the estimated networks, is fixed at 0.3 for the AV45 data and 0.2 for the AV1451 data based on empirical observations. Besides, the dataset is normalized such that it has a mean of zero and a standard deviation of one.

**Results.**   The numerical results in Table 4 demonstrate that Fair GLasso effectively reduces disparity error compared to standard GLasso, enhancing fairness while maintaining a good objective value. Figure 5 reveals notable differences in the learned network structures for AV45 results. The presence of edges between the left caudal middle frontal gyrus and right medial orbitofrontal cortex and between the left superior frontal gyrus and left superior parietal lobule in the GLasso graph suggests an increased influence of emotional factors on executive function and higher-order cognitive processes on amyloid accumulation, respectively [64, 6, 54]. Conversely, the absence of an edge between the left pars opercularis and left supramarginal gyrus in the Fair GLasso graph indicates a weaker association between language deficits and sensorimotor impairments in amyloid accumulation [18, 81, 38].

In contrast, the AV1451 results show primarily numerical differences between the two graphs, with edges remaining largely unchanged, suggesting that the tau accumulation network is robust to the sensitive attribute of race. These findings highlight the importance of considering fairness in brain imaging data analysis and the potential of Fair GLasso to uncover more equitable and unbiased patterns of amyloid and tau accumulation in Alzheimer's disease. Further research is needed to validate these findings in larger and more diverse cohorts and explore the biological mechanisms and clinical implications of the observed differences between standard GLasso and Fair GLasso graphs.

## D.5   Addendum to Subsection 4.6

Table 5 summarizes the details of credit data utilized on Fair CovGraph mentioned in Section 4.6.

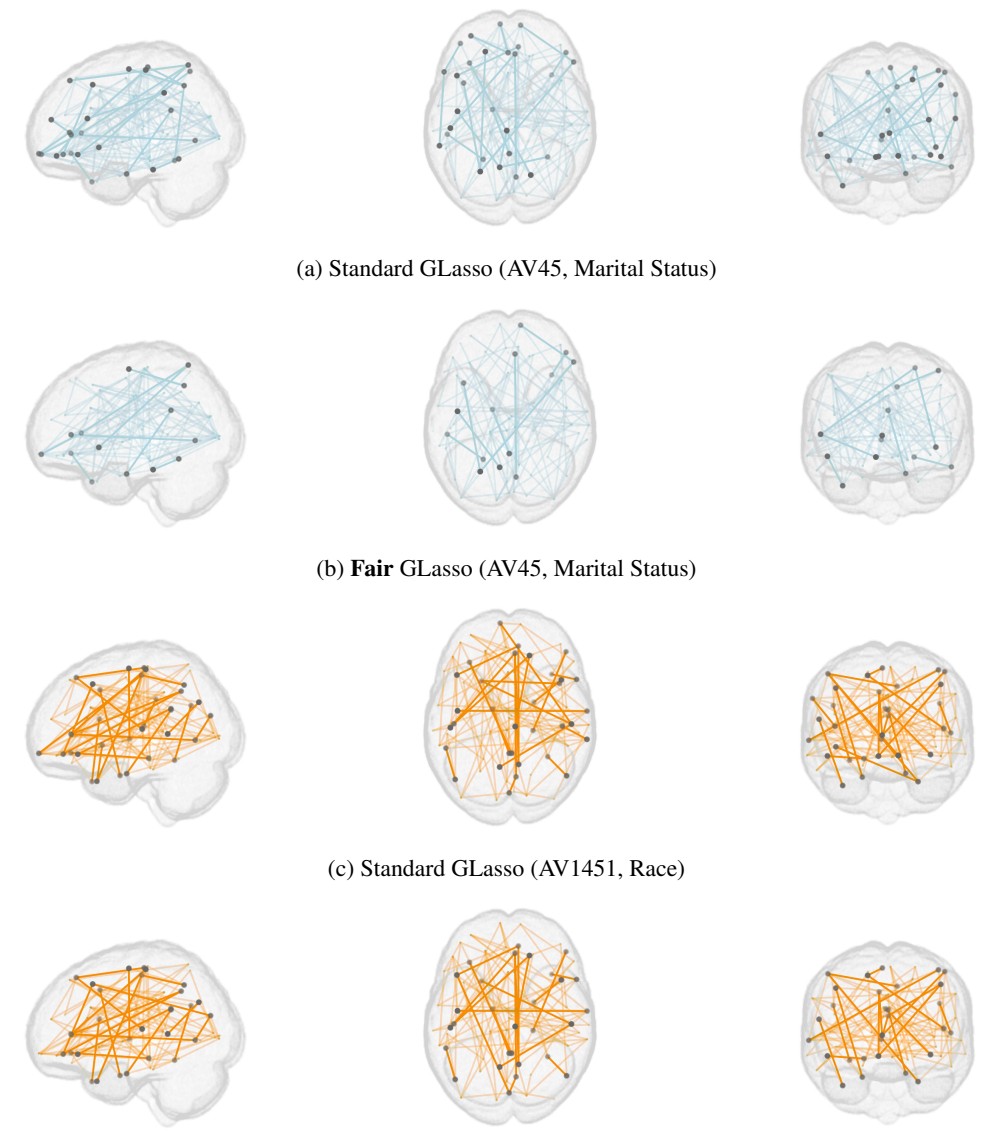

(a) Standard GLasso (AV45, Marital Status)

(b) **Fair** GLasso (AV45, Marital Status)

(c) Standard GLasso (AV1451, Race)

(d) **Fair** GLasso (AV1451, Race)

Figure 5: Subfigures (a) and (b) present a comparison of the graphs generated by standard GLasso and Fair GLasso on the ADNI dataset, considering the sensitive attribute of marital status on the AV45 biomarker. Similarly, subfigures (c) and (d) compare the graphs generated by both methods, taking into account the sensitive attribute of race on the AV1451 biomarker. To improve the clarity of the visualizations, weak edges have been removed, and edges that show significant differences in values between the two methods are highlighted. It is important to note that even though some edges may appear unchanged in the visual comparison, their actual values will differ between the standard GLasso and Fair GLasso methods.

Table 5: Distribution of the number of samples in each group in the credit dataset. "HgEd" represents "High School Graduate or Higher", and "LwEd" represents "Education below High School Level".

| Name | Size | Name | Size |
|---|---|---|---|
| Male_Singe_HgEd | 5579 | Female_Single_HgEd | 8260 |
| Male_Singe_LwEd | 974 | Female_Single_LwEd | 1151 |
| Male_Married_HgEd | 4062 | Female_Married_HgEd | 6506 |
| Male_Married_LwEd | 1128 | Female_Married_LwEd | 1963 |

## D.6 Sensitivity Analysis to Feature Size $P$

Table 6: Numerical outcomes in terms of the value of the objective function ($F_1$), the summation of the pairwise graph disparity error ($\Delta$), and the average computation time in seconds ($\pm$ standard deviation) from 10 repeated experiments. $K = 2$ and $N_k = 1000 \ \forall k \in [K]$. "$\downarrow$" means the smaller, the better, and the best value is in bold. These experiments are conducted on an Apple M2 Pro processor. Note that both $F_1$ and $\Delta$ are deterministic.

| Feature Size $P$ | $F_1 \downarrow$ | | $\%F_1 \uparrow$ | $\Delta \downarrow$ | | $\%\Delta \uparrow$ | Runtime $\downarrow$ | |
|---|---|---|---|---|---|---|---|---|
| | GM | Fair GM | | GM | Fair GM | | GM | Fair GM |
| 50 | **50.9229** | 50.9429 | -0.04% | 0.7309 | **0.3832** | +47.56% | **0.035** ($\pm$ **0.01**) | 19.86 ($\pm$ 0.24) |
| 100 | **105.089** | 105.149 | -0.06% | 2.1799 | **1.1206** | +48.60% | **0.183** ($\pm$ **0.01**) | 37.14 ($\pm$ 1.13) |
| 150 | **159.424** | 159.555 | -0.08% | 4.6926 | **1.7772** | +62.13% | **2.452** ($\pm$ **0.89**) | 48.47 ($\pm$ 4.50) |
| 200 | **215.402** | 215.558 | -0.07% | 5.7447 | **1.9693** | +65.72% | **2.034** ($\pm$ **0.11**) | 53.57 ($\pm$ 1.40) |
| 250 | **269.236** | 269.356 | -0.04% | 5.4889 | **2.1106** | +61.55% | **0.552** ($\pm$ **0.04**) | 58.71 ($\pm$ 1.55) |
| 300 | **324.733** | 324.958 | -0.07% | 8.5738 | **2.5582** | +70.16% | **1.508** ($\pm$ **0.07**) | 68.11 ($\pm$ 1.09) |
| 350 | **379.696** | 380.027 | -0.09% | 12.758 | **3.2992** | +74.14% | **3.199** ($\pm$ **0.24**) | 105.8 ($\pm$ 2.50) |
| 400 | **434.697** | 434.927 | -0.05% | 9.4141 | **2.4337** | +74.15% | **1.509** ($\pm$ **0.12**) | 107.7 ($\pm$ 3.55) |

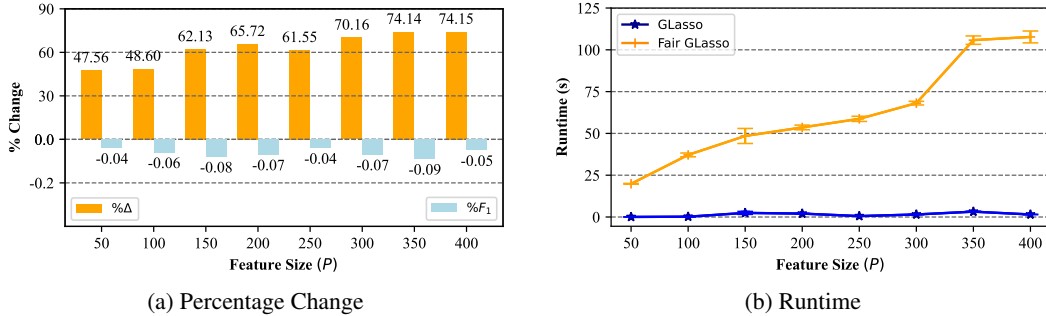

(a) Percentage Change        (b) Runtime

Figure 6: (a) Percentage change from GLasso to Fair GLasso (results from Table 6) with respect to feature size $P$. $\%F_1$ is slight, while $\%\Delta$ changes are substantial, signifying fairness improvement without significant accuracy sacrifice. (b) Runtime (mean $\pm$ std) (results from Table 6) with respect to feature size $P$.

In this section, we examine the impact of varying feature sizes $P$ on the $\%F_1$ score, $\%\Delta$ (change in accuracy), and runtime. Our experiments utilize feature sizes ranging from $P = 50$ to $P = 400$ in the GLasso algorithm applied to synthetic data. According to the procedures described in Steps 1-3 from Section D.2, we generate covariance matrices for two distinct groups: $\boldsymbol{\Sigma}_1$ featuring five diagonal blocks and $\boldsymbol{\Sigma}_2$ with four diagonal blocks.

For each feature size setting, Group 1 includes 1000 observations drawn from a multivariate normal distribution $\mathcal{N}(0, \boldsymbol{\Sigma}_1)$, and Group 2 also consists of 1000 observations from $\mathcal{N}(0, \boldsymbol{\Sigma}_2)$. The outcomes of these experiments are systematically presented in Table 6 and visually depicted in Figure 6. This structured analysis enables us to evaluate how changes in feature size affect both performance metrics and computational efficiency in our study.

By integrating both the Table 6 and Figure 6, it can be observed that as the feature size increases, although there is a rise in the pairwise graph disparity error, our proposed method still effectively reduces it, with minimal loss in the objective value. This underscores the efficacy of our approach in enhancing fairness. Regarding runtime, there is a proportional relationship between feature size and the runtime of Fair GLasso, which aligns with our theoretical analysis of algorithmic complexity.

## D.7 Sensitivity Analysis to Sample Size $N$

In this section, we conduct a sensitivity analysis with respect to the sample size $N$, while holding the feature size fixed at $P = 50$. We investigate how varying the sample size impacts the $\%F_1$ score, $\%\Delta$ (change in accuracy), and runtime. The sample sizes examined are $N_k = 100, 150, 200, ..., 400, 500$ for each group in the Fair GLasso on synthetic data.

Following the procedures outlined in Steps 1-3 in Section D.2, we generate synthetic datasets with fixed covariance structures for two distinct groups: $\boldsymbol{\Sigma}_1$ characterized by five diagonal blocks, and $\boldsymbol{\Sigma}_2$ comprising four diagonal blocks. Each dataset is generated for every specified sample size, allowing

Table 7: Numerical outcomes in terms of the value of the objective function ($F_1$), the summation of the pairwise graph disparity error ($\Delta$), and the average computation time in seconds ($\pm$ standard deviation) from 10 repeated experiments. $K = 2$ and $P = 50$. "↓" means the smaller, the better, and the best value is in bold. These experiments are conducted on an Apple M2 Pro processor. Note that both $F_1$ and $\Delta$ are deterministic.

| Sample Size $N_k$ | $F_1 \downarrow$ | | $\%F_1 \uparrow$ | $\Delta \downarrow$ | | $\%\Delta \uparrow$ | Runtime ↓ | |
| --- | --- | --- | --- | --- | --- | --- | --- | --- |
| | GM | Fair GM | | GM | Fair GM | | GM | Fair GM |
| 100 | **50.2970** | 50.3049 | -0.02% | 0.6988 | **0.3715** | +46.83% | **0.044** ($\pm$ **0.01**) | 26.89 ($\pm$ 1.32) |
| 150 | **50.6003** | 50.6043 | -0.01% | 0.3407 | **0.2042** | +40.05% | **0.037** ($\pm$ **0.01**) | 19.12 ($\pm$ 0.53) |
| 200 | **50.8234** | 50.8438 | -0.04% | 0.7194 | **0.2843** | +60.49% | **0.103** ($\pm$ **0.02**) | 19.13 ($\pm$ 0.27) |
| 250 | **50.8729** | 50.8978 | -0.05% | 0.8615 | **0.3514** | +59.21% | **0.099** ($\pm$ **0.16**) | 23.06 ($\pm$ 1.06) |
| 300 | **50.8791** | 50.8912 | -0.02% | 0.6464 | **0.3718** | +42.48% | **0.046** ($\pm$ **0.01**) | 30.18 ($\pm$ 0.99) |
| 350 | **50.9272** | 50.9448 | -0.03% | 0.7120 | **0.3660** | +48.60% | **0.018** ($\pm$ **0.00**) | 25.59 ($\pm$ 0.30) |
| 400 | **50.9186** | 50.9344 | -0.03% | 0.6675 | **0.3537** | +47.01% | **0.030** ($\pm$ **0.00**) | 23.33 ($\pm$ 0.29) |
| 500 | **50.9021** | 50.9261 | -0.05% | 0.7281 | **0.3137** | +56.91% | **0.038** ($\pm$ **0.00**) | 17.86 ($\pm$ 0.31) |

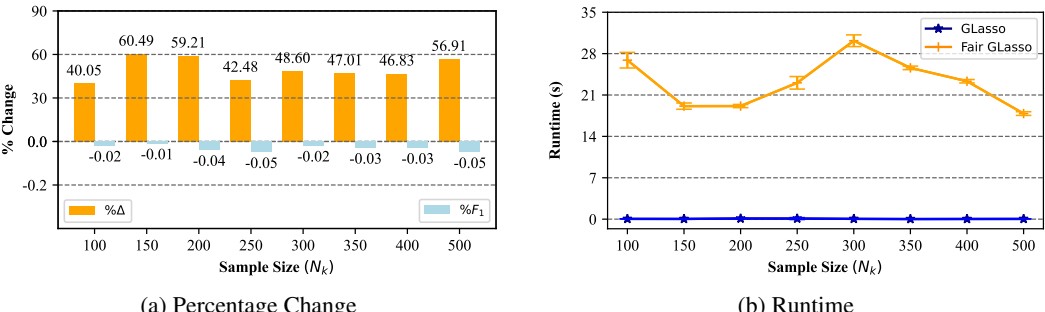

(a) Percentage Change

(b) Runtime

Figure 7: (a) Percentage change from GLasso to Fair GLasso (results from Table 7) with respect to sample size $N$. $\%F_1$ is slight, while $\%\Delta$ changes are substantial, signifying fairness improvement without significant accuracy sacrifice. (b) Runtime (mean $\pm$ std) (results from Table 7) with respect to sample size $N$.

us to systematically assess the effects of increasing $N$ on the performance metrics and computational efficiency of the algorithm.

The specific results are presented in Table 7 and visualized in Figure 7. From these, it is evident that the sample size does not significantly impact the objective value, pairwise graph disparity error, or runtime. Our proposed method consistently maintains its effectiveness across different sample sizes. This stability highlights the robustness of our approach under varying data quantities.

### D.8 Sensitivity Analysis to Sample Size Ratio $N_2/N_1$

Table 8: Numerical outcomes in terms of the value of the objective function ($F_1$), the summation of the pairwise graph disparity error ($\Delta$), and the average computation time in seconds ($\pm$ standard deviation) from 10 repeated experiments. $K = 2$, $P = 50$, and $N_2 = 100$. "↓" means the smaller, the better, and the best value is in bold. These experiments are conducted on an Apple M2 Pro processor. Note that both $F_1$ and $\Delta$ are deterministic.

| Sample Size Ratio $N_1/N_2$ | $F_1 \downarrow$ | | $\%F_1 \uparrow$ | $\Delta \downarrow$ | | $\%\Delta \uparrow$ | Runtime ↓ | |
| --- | --- | --- | --- | --- | --- | --- | --- | --- |
| | GM | Fair GM | | GM | Fair GM | | GM | Fair GM |
| 1.0 | **50.2970** | 50.3049 | -0.02% | 0.6988 | **0.3715** | +46.83% | **0.175** ($\pm$ **0.37**) | 26.77 ($\pm$ 0.59) |
| 2.0 | **50.4208** | 50.5981 | -0.35% | 4.5459 | **0.8282** | +81.78% | **0.042** ($\pm$ **0.01**) | 21.64 ($\pm$ 0.42) |
| 3.0 | **50.4427** | 50.9140 | -0.93% | 8.3116 | **0.8556** | +89.71% | **0.061** ($\pm$ **0.01**) | 16.06 ($\pm$ 0.42) |
| 4.0 | **50.3129** | 50.8348 | -1.04% | 8.6970 | **1.0065** | +88.43% | **0.033** ($\pm$ **0.01**) | 21.64 ($\pm$ 0.38) |
| 5.0 | **50.1979** | 50.8795 | -1.36% | 10.213 | **1.1157** | +89.07% | **0.049** ($\pm$ **0.02**) | 22.66 ($\pm$ 0.32) |
| 7.0 | **50.1567** | 51.1681 | -2.02% | 14.224 | **1.2931** | +90.91% | **0.033** ($\pm$ **0.01**) | 25.59 ($\pm$ 0.24) |
| 10.0 | **50.0203** | 50.9329 | -1.82% | 11.462 | **1.2407** | +89.18% | **0.035** ($\pm$ **0.00**) | 22.35 ($\pm$ 0.47) |
| 100.0 | **49.8966** | 51.2365 | -2.69% | 17.620 | **1.0912** | +93.81% | **0.033** ($\pm$ **0.01**) | 16.51 ($\pm$ 0.36) |

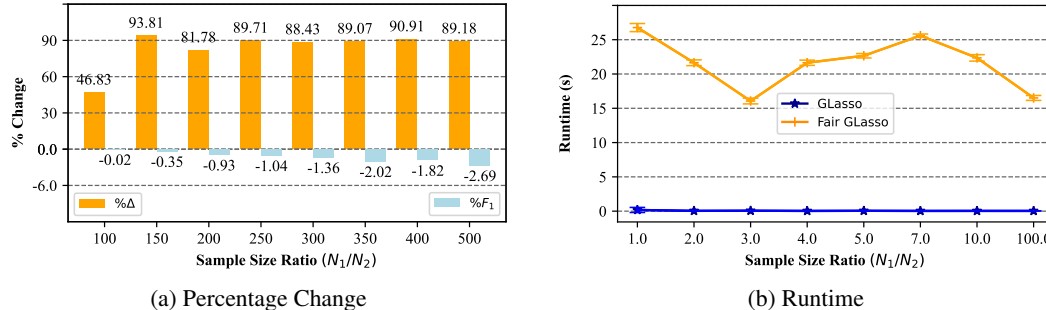

|                | (a) Percentage Change | (b) Runtime |

Figure 8: (a) Percentage change from GLasso to Fair GLasso (results from Table 7) with respect to sample size ratio $N_1/N_2$. $\%F_1$ is slight, while $\%\Delta$ changes are substantial, signifying fairness improvement without significant accuracy sacrifice. (b) Runtime (mean $\pm$ std) (results from Table 7) with respect to sample size ratio $N_1/N_2$.

We conduct a sensitivity analysis on the sample size ratio $N_1/N_2$ while keeping the feature size fixed at $P = 50$ and Group 2's sample size $N_2$ constant at 100. We examine the impact of varying $N_1/N_2$ on the $\%F_1$, $\%\Delta$, and runtime in our experiments with Fair GLasso on synthetic data.

Following the methodology outlined in Steps 1-3 from Section D.2, we generate datasets with fixed covariance structures: $\Sigma_1$ characterized by five diagonal blocks for Group 1 and $\Sigma_2$ with four diagonal blocks for Group 2. We systematically vary $N_1$ from 100 to 10,000, maintaining $N_2$ at 100, and assess how changes in the sample size ratio affect the algorithm's performance metrics and computational efficiency.

The specific results of these experiments are detailed in Table 8 and visualized in Figure 8. From this analysis, it is apparent that the sample size ratio $N_1/N_2$ does not significantly affect the objective value, pairwise graph disparity error, or runtime. Our proposed method continues to demonstrate its effectiveness consistently across varying sample size ratios. This consistency underscores the robustness of our approach, showing its reliability regardless of changes in the group imbalance between the groups.

### D.9 Sensitivity Analysis to Group Size $K$

Table 9: Numerical outcomes in terms of the value of the objective function ($F_1$), the summation of the pairwise graph disparity error ($\Delta$), and the average computation time in seconds ($\pm$ standard deviation) from 10 repeated experiments. $P = 100$ and $N_k = 1000$ $\forall k \in [K]$. "$\downarrow$" means the smaller, the better, and the best value is in bold. These experiments are conducted on an Apple M2 Pro processor. Note that both $F_1$ and $\Delta$ are deterministic.

| Group Size $K$ | $F_1 \downarrow$ | | $\%F_1 \uparrow$ | $\Delta \downarrow$ | | $\%\Delta \uparrow$ | Runtime $\downarrow$ | |
|---|---|---|---|---|---|---|---|---|
| | GM | Fair GM | | GM | Fair GM | | GM | Fair GM |
| 2 | **101.306** | 101.331 | -0.03% | 0.9451 | **0.4523** | +52.14% | **0.183 ($\pm$ 0.02)** | 28.16 ($\pm$ 1.14) |
| 3 | **102.242** | 102.441 | -0.19% | 3.7579 | **0.6341** | +83.13% | **0.155 ($\pm$ 0.06)** | 44.25 ($\pm$ 3.76) |
| 4 | **103.157** | 103.664 | -0.49% | 9.4820 | **0.5665** | +94.03% | **0.146 ($\pm$ 0.03)** | 128.3 ($\pm$ 4.95) |
| 5 | **103.856** | 104.730 | -0.84% | 18.835 | **0.4451** | +97.64% | **0.192 ($\pm$ 0.03)** | 103.2 ($\pm$ 4.62) |
| 6 | **104.489** | 105.710 | -1.17% | 31.688 | **0.4085** | +98.71% | **0.167 ($\pm$ 0.03)** | 114.5 ($\pm$ 5.21) |
| 7 | **105.113** | 106.685 | -1.50% | 48.421 | **0.4113** | +99.15% | **0.192 ($\pm$ 0.03)** | 117.6 ($\pm$ 8.08) |
| 8 | **105.806** | 107.663 | -1.75% | 68.335 | **0.7127** | +98.96% | **0.149 ($\pm$ 0.04)** | 133.2 ($\pm$ 9.25) |
| 9 | **106.458** | 108.810 | -2.21% | 92.749 | **1.5164** | +98.37% | **0.233 ($\pm$ 0.06)** | 265.7 ($\pm$ 11.2) |
| 10 | **107.112** | 109.742 | -2.46% | 121.37 | **2.1924** | +98.19% | **0.134 ($\pm$ 0.06)** | 360.9 ($\pm$ 23.0) |

In this section, we explore the impact of group size $K$ on the performance and computational efficiency of Fair GLasso. The feature size $N$ is fixed at 100, and the sample size per group $P_k$ is set at 1000. Following Steps 1-3 from Section D.2, the covariance matrix for the first group, $\Sigma_1$ is generated with 10 diagonal blocks. Each subsequent group has one fewer diagonal block in its covariance matrix, with each group sampling observations from $\mathcal{N}(0, \Sigma_k)$.

The results are detailed in Table 9 and Figure 9. Observations indicate that when the group size is less than 9, computational efficiency remains relatively stable regardless of changes in group size. However, efficiency decreases noticeably when the group size increases to 9. In terms of the objective

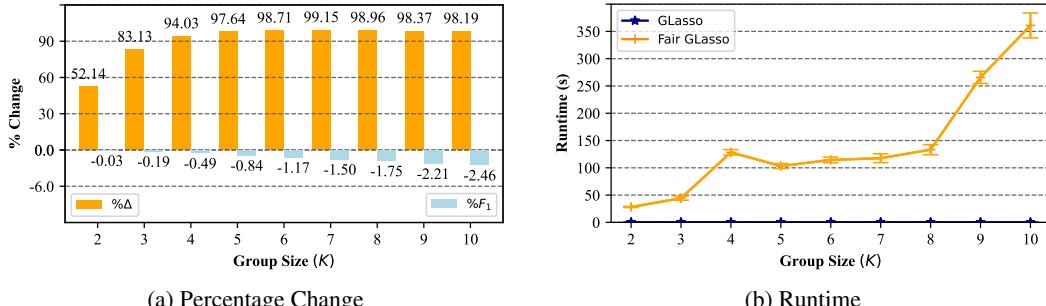

(a) Percentage Change           (b) Runtime

Figure 9: (a) Percentage change from GLasso to Fair GLasso (results from Table 6) with respect to group size $K$. $\%F_1$ is slight, while $\%\Delta$ changes are substantial, signifying fairness improvement without significant accuracy sacrifice. (b) Runtime (mean $\pm$ std) (results from Table 6) with respect to group size $K$.

Table 10: Outcomes of additional baseline with different optimization algorithms applied to GLasso and Multi-Objective Optimization (MOO), measured in terms of the value of the objective function ($F_1$), the summation of the pairwise graph disparity error ($\Delta$), and the average computation time in seconds ($\pm$ standard deviation) from 10 repeated experiments. "↓" indicates that smaller values are better. Our method applies ISTA to both GLasso and MOO (first row in each experiment). All experiments are conducted using the same runtime environment on Google Colab.

| Algorithm | | $F_1 \downarrow$ | | $\%F_1 \uparrow$ | $\Delta \downarrow$ | | $\%\Delta \uparrow$ | Runtime $\downarrow$ | |
|---|---|---|---|---|---|---|---|---|---|
| GLasso | MOO | GLasso | Fair GLasso | | GLasso | Fair GLasso | | GLasso | Fair GLasso |
| Synthetic Dataset 1 (2 Subgroups, 100 Variables, 1000 Observations in Each Group) | | | | | | | | | |
| ISTA | ISTA | 97.172 | 97.449 | -0.29% | 7.8149 | 0.5794 | +92.59% | 0.501 ($\pm$ 0.21) | 85.48 ($\pm$ 1.92) |
| ISTA | FISTA | 97.172 | 97.438 | -0.27% | 7.8149 | 0.8835 | +88.69% | 0.297 ($\pm$ 0.12) | 26.56 ($\pm$ 1.11) |
| PISTA | FISTA | 97.172 | 97.438 | -0.27% | 7.8190 | 0.9084 | +88.38% | 13.52 ($\pm$ 1.10) | 59.66 ($\pm$ 2.65) |
| GISTA | FISTA | 97.172 | 97.438 | -0.27% | 7.8149 | 0.9089 | +88.37% | 0.426 ($\pm$ 0.16) | 21.27 ($\pm$ 0.94) |
| OBN | FISTA | 97.172 | 97.438 | -0.27% | 7.8134 | 0.9112 | +88.34% | 0.483 ($\pm$ 0.16) | 22.48 ($\pm$ 0.92) |
| Synthetic Dataset 2 (2 Subgroups, 200 Variables, 2000 Observations in Each Group) | | | | | | | | | |
| ISTA | ISTA | 199.71 | 200.70 | -0.49% | 40.511 | 1.4855 | +96.33% | 2.622 ($\pm$ 1.28) | 206.7 ($\pm$ 3.27) |
| ISTA | FISTA | 199.71 | 200.68 | -0.49% | 40.511 | 1.8485 | +95.44% | 2.640 ($\pm$ 0.76) | 108.1 ($\pm$ 2.42) |
| PISTA | FISTA | 199.71 | 200.67 | -0.48% | 40.521 | 1.9474 | +95.19% | 39.16 ($\pm$ 2.30) | 178.7 ($\pm$ 3.50) |
| GISTA | FISTA | 199.71 | 200.68 | -0.48% | 40.511 | 2.0260 | +95.00% | 2.365 ($\pm$ 0.26) | 78.99 ($\pm$ 3.07) |
| OBN | FISTA | 199.71 | 200.72 | -0.50% | 40.511 | 2.4835 | +93.87% | 2.403 ($\pm$ 0.68) | 53.11 ($\pm$ 2.17) |
| Synthetic Dataset 3 (10 Subgroups, 100 Variables, 1000 Observations in Each Group) | | | | | | | | | |
| ISTA | ISTA | 95.333 | 95.603 | -0.28% | 11.394 | 0.3108 | +97.27% | 0.641 ($\pm$ 0.28) | 224.1 ($\pm$ 2.29) |
| ISTA | SOSA | 95.333 | 95.506 | -0.18% | 11.394 | 1.5133 | +86.72% | 0.626 ($\pm$ 0.19) | 143.2 ($\pm$ 2.28) |

value and pairwise graph disparity error, performance maintains a good balance, with a significant enhancement in fairness.

This conclusion aligns with our theoretical analysis of algorithmic complexity. Notably, as the group size increases, the pairwise graph disparity error also significantly rises, as shown in Table 9. Consequently, our proposed method effectively enhances fairness, albeit at the cost of sacrificing a greater portion of the objective value. This trade-off is a critical aspect of our approach, balancing computational performance with the desired ethical outcomes in machine learning applications.

### D.10    Addendum to Subsection 4.8

To address the computational complexity of Fair GMs, we explore a range of optimization methods tailored to GLasso and multi-objective optimization (MOO):

- GLasso Optimization Methods
  - Preconditioned Iterative Soft Thresholding Algorithm (PISTA): Efficiently handles large-scale sparse matrix operations [69].
  - Graphical Iterative Shrinkage Thresholding Algorithm (GISTA): Employs an iterative framework for sparsity-inducing penalty functions in high-dimensional settings [60].
  - Orthant-Based Newton Method (OBN): Uses second-order information for faster convergence in structured sparsity constraints [57].

- MOO Optimization Methods
  - Fast Iterative Shrinkage-Thresholding Algorithm (FISTA): Provides globally optimal convergence rates for MOO objectives [74].
  - Stochastic Objective Selection Approach (SOSA): Introduces a randomized selection technique for optimizing multi-objective functions under varying conditions [70].

We validate these methods through comprehensive experiments on synthetic datasets. Our first evaluation uses data with 100 variables across two subgroups, each containing 1000 observations, generated following the procedure in Appendix D.2. This experiment demonstrates that faster optimization methods improve time complexity for both GLasso and MOO while maintaining performance. All GLasso methods achieve optimal loss, while Fair GLasso variants successfully reduce pairwise graph disparity error without significant performance degradation.

To assess scalability, we extend our analysis to a larger dataset with 200 variables, maintaining the same experimental setup. Furthermore, we evaluate the efficiency of our approach with increased group complexity using synthetic data containing 100 variables across ten subgroups, each with 1000 observations. In this setting, SOSA reduces training time by approximately 36% compared to the original approach while preserving model fairness and robustness.

The numerical results presented in Table 10 confirm that our optimization strategies successfully reduce runtime while maintaining model robustness and fairness. These findings suggest promising directions for future research in balancing computational efficiency with model performance.

