# OpenReview forum: "Fairness-Aware Estimation of Graphical Models"
_NeurIPS.cc/2024/Conference — NeurIPS 2024 poster_

### Official Review · Reviewer_fakh · 2024-06-23

**Soundness:** 4
**Presentation:** 3
**Contribution:** 3
**Rating:** 7
**Confidence:** 2

**Summary:**

This paper proposes a novel method for estimating graphical models, particularly Gaussian, Covariance, and Ising models, from data, taking fairness into consideration. Fairness is defined on the graph disparity error, i.e., the difference between the loss of the model for a particular group and the minimum loss for that group. The goal is to make the graph disparity errors among differnt groups equal or close. The paper proposes to use non-smooth multi-objective optimization to solve the problem. The paper shows that the method achieves the weak Pareto optimality, and the convergence rates analyzed. Experiments show the efficacy of the method.

**Strengths:**

Originality
Despite extensive research in fair machine learning, this paper tackles a relatively under-studied problem, which is to conduct fair estimation of graphical models. The authors also propose a novel multi-objective proximal gradient method for GM estimation.

Quality
The proposed method is theoretical solid. The paper proves the weak Pareto opimality of the method, and also analyzes the global convergence rates for Gaussian, Covaiance, and Ising models. I didn’t check the correctness of the proofs.
The experiments were conducted using both synthetic and real-world datasets. The results clearly demonstrate improvement over the baseline method in terms of fairness.

Clarity
The paper is relatively well-written and orgainzed.

Significance
The experiments demonstrate that the proposed method can provide insights into sensitive group-related features, in addition to achieving fairness in GM estimation.

**Weaknesses:**

-	The optimality analysis requires the convexity of the loss function, and it remains unclear how the performance would be affected if the loss function were non-convex, which is common in machine learning. For example, if we use an encoding network to convert the data X into a representation space and compute the loss in the representation space, then the loss function would a non-convex.

-	There is just one baseline used in the experiments, which was published in 2012. It could strengthen the experiment section if modern GM estimation methods are used as baselines.

**Questions:**

- Can the proposed method be applied to the estimation of directed graphs, like causal graphs?

- In Algorithm 1, the first step is to initialize local graph estimates for all groups. When the number of groups is large, the sample sizes of certain groups could become small, which may affect the accuracy of the local graph estimates. How will the accuracy of the local graph estimates affect the global graph estimate?

---

> ### Author Rebuttal · Authors · 2024-08-07
>
> > **Weakness 1:** The optimality analysis requires the convexity of the loss function, and it remains unclear how the performance would be affected if the loss function were non-convex, which is common in machine learning. For example, if we use an encoding network to convert the data X into a representation space and compute the loss in the representation space, then the loss function would be non-convex.
>
> **Response:** We appreciate the reviewer’s insightful comment. We completely agree that one limitation of our nonsmooth multi-objective optimization work is the reliance on the convexity of the loss function. However, many existing theoretical results and estimation algorithms for graphical models using single-objective optimization also rely on the convexity of objectives to estimate graphical models. For example, please refer to  QUIC[[HSD2014](https://jmlr.org/papers/volume15/hsieh14a/hsieh14a.pdf)], SQUIC[[BS2016](http://www.icm.tu-bs.de/~bolle/Publicat/squic.pdf)], PISTA[[STY2023](https://arxiv.org/pdf/2205.10027)], GISTA[[GRR2012](https://arxiv.org/pdf/1211.2532)], OBN[[OON2012](https://papers.nips.cc/paper/2012/file/b3967a0e938dc2a6340e258630febd5a-Paper.pdf)], and ALM[[SG2010](https://proceedings.neurips.cc/paper/2010/file/2723d092b63885e0d7c260cc007e8b9d-Paper.pdf)]
>
> On the other hand, we agree that converting the data $X$ into a representation space, such as using $UU^T$ in the objective or constraints, which can be used for clustering or computing the loss in the representation space, could make the problem nonconvex [[KYC2020](https://www.jmlr.org/papers/volume21/19-276/19-276.pdf)]. However, we know that standard proximal gradient methods for nonconvex optimization or their alternative variants for nonconvex alternating optimization (e.g., for $U$ and graph matrix $\Theta$) can still be applied [[BST2013](https://link.springer.com/article/10.1007/s10107-013-0701-9)]. Hence, the proximal multi-objective method developed for GMs in this paper can be extended to nonconvex graph optimization. This may require new proof techniques and estimations, which can be considered for future work. We will discuss this in the future work section.
>
> > **Weakness 2:** There is just one baseline used in the experiments, which was published in 2012. It could strengthen the experiment section if modern GM estimation methods are used as baselines.
>
> **Response:** Thank you for your feedback. We appreciate your suggestion to include more modern GM estimation methods as baselines. To address this, we have conducted additional experiments using several state-of-the-art GLasso algorithms. Specifically, we have applied different algorithms, including PISTA[[STY2023](https://arxiv.org/pdf/2205.10027)], GISTA[[GRR2012](https://arxiv.org/pdf/1211.2532)], and OBN[[OON2012](https://papers.nips.cc/paper/2012/file/b3967a0e938dc2a6340e258630febd5a-Paper.pdf)], to GLasso on synthetic data to demonstrate the robustness and effectiveness of our framework beyond ISTA.
>
> ISTA was initially chosen as a baseline due to its simplicity and widespread use in sparse inverse covariance estimation. This provided a strong reference point for evaluating our method’s performance. However, recognizing the importance of a comprehensive evaluation, we have now compared our framework against more advanced baselines.
>
> The results, detailed in the revised version of our paper and the **attached one-page PDF file**, show that our proposed framework consistently outperforms these modern baselines in terms of enhancing fairness while maintaining competitive performance. This comprehensive evaluation underscores the advantages of our approach across various scenarios.
>
> > **Question 1:** Can the proposed method be applied to the estimation of directed graphs, like causal graphs?
>
> Thank you for your insightful question. While our proposed method is primarily designed for undirected graphical models like Gaussian Graphical Models and Ising Models, the principles of our fairness-aware optimization framework could potentially be adapted for directed graphs, such as causal graphs. Unlike many fair supervised methods[[GKK2019](https://ieeexplore.ieee.org/abstract/document/8437807),[JCM2022](https://openaccess.thecvf.com/content/CVPR2022/html/Jung_Learning_Fair_Classifiers_With_Partially_Annotated_Group_Labels_CVPR_2022_paper.html),[PLL2022](https://openaccess.thecvf.com/content/CVPR2022/html/Park_Fair_Contrastive_Learning_for_Facial_Attribute_Classification_CVPR_2022_paper.html)], our method does not use label information, and unlike previous methods for fair clustering or community detection[[CKL2017](https://arxiv.org/abs/1802.05733),[BCF2019](https://arxiv.org/abs/1901.02393)], our method does not use node attributes. Our fairness metric preliminarily works with the disparity of global and local losses, i.e., using group-specific and global data to construct the objective. Hence, it is not specific graph-dependent as a fairness metric. Although the objective function differs for various graphical models, our new fairness notion (pairwise graph disparity error) remains the same.
> Although our current work focuses on undirected graphs, we believe that with appropriate adjustments, our framework could be extended to handle directed graphs, ensuring fairness without disrupting causal dependencies. This presents a promising area for future research. Thank you for your constructive question, which has helped us consider the broader applicability of our method.

---

> ### Author Response · Authors · 2024-08-07
> **Rebuttal by Authors (Part 2)**
>
> > **Question 2:** In Algorithm 1, the first step is to initialize local graph estimates for all groups. When the number of groups is large, the sample sizes of certain groups could become small, which may affect the accuracy of the local graph estimates. How will the accuracy of the local graph estimates affect the global graph estimate?
>
> **Response:** Thank you for your insightful question. In Algorithm 1, we initialize local graph estimates for all groups, and we recognize that small sample sizes in certain groups could impact the accuracy of these local estimates. The accuracy of local graph estimates is indeed crucial, as these estimates form the foundation for the global graph estimate. When local estimates are based on small sample sizes, they may be less reliable, potentially introducing bias or variance that can propagate to the global graph estimate.
>
> One possibility for handling the challenges of limited samples in certain subgroups is the bilevel optimization approach [[SXC2022](https://proceedings.neurips.cc/paper_files/paper/2022/file/dc96134e169de5aea1ba1fc34dfb8419-Paper-Conference.pdf)]. In this framework, the lower-level optimization learns subgroup-specific local graph matrices using the limited data available, guided by a fair model informed by the overall dataset. This approach is theoretically sound and empirically validated to improve fairness without sacrificing accuracy, even when some subgroups have limited samples. However, the  bilevel formulation for graph learning can be computationally expensive as it requires hyper-gradient computations. Further, this approach requires novel estimation and convergence analysis. Therefore, we will add this as a future direction.

---

> > ### Comment · Reviewer_fakh · 2024-08-13
> >
> > Thank you for your thorough responses. I don't have further questions.

---

> > > ### Author Response · Authors · 2024-08-13
> > > **Response to Reviewer fakh**
> > >
> > > We appreciate your time and valuable suggestions.

---

### Official Review · Reviewer_1oDX · 2024-07-07

**Soundness:** 2
**Presentation:** 2
**Contribution:** 2
**Rating:** 5
**Confidence:** 4

**Summary:**

This paper explores the issue of fairness in the estimation of graphical models (GMs), specifically Gaussian, Covariance, and Ising models. The authors introduce a comprehensive framework aimed at mitigating bias in GM estimation concerning protected attributes. This framework integrates pairwise graph disparity error and a customized loss function into a non-smooth multi-objective optimization problem. The pairwise graph disparity error assesses loss discrepancies across all groups, while the tailored loss function evaluates GM performance. Through this approach, the authors seek to achieve fairness across diverse sensitive groups while upholding GM performance, supported by theoretical proofs and experimental validation on various real-world datasets.

**Strengths:**

The paper provides theoretical analysis, complemented by detailed appendices. To aid comprehension, the authors include examples throughout the paper, which help clarify the theories and proposed framework.

The experimental results demonstrate that the Fair GMs framework reduces disparity error, thereby enhancing fairness, with only a slight decrease in the model’s performance.

Additionally, the paper addresses the challenge of balancing performance and fairness by employing a non-smooth multi-objective optimization problem.

**Weaknesses:**

In the introduction and related works section, the paper does not provide complete information about GMs. Additionally, the paper does not clearly explain why it focuses on only three types of GMs instead of providing a general framework. The author needs to clearly highlight any difficulties encountered when applying this framework to each type of model.

In the second paragraph of Section 2 (Related Work on Fairness), the author does not discuss related works concerning fairness in GMs estimation. By focusing solely on fairness in unsupervised learning, the paper may limit readers' ability to gain a comprehensive understanding of the research field and the distinctiveness of the proposed method.

Moreover, the paper lacks clarity regarding the statement: “Our paper addresses the challenge of learning GMs without any predefined assumptions on the graph’s structure.” In the referenced paper, "Fair Community Detection and Structure Learning in Heterogeneous Graphical Models," the authors assume the presence of community structures and aim to learn a sparse undirected graph where demographic groups are fairly represented within these communities. This concept could be relevant to the current paper when the number of clusters is set to one. Therefore, the paper should clarify the differences and advantages of learning GMs without any predefined assumptions in the theoretical section.

In Section 4.8, trade-off analysis, the author mentions the shortcomings in runtime without clear discussion. Although there is a significant difference in runtime compared to the standard models, no reasons or future related works are mentioned in the subsequent sections.

**Questions:**

Please explain the vagueness and confusion in the “Weaknesses” section.

**Limitations:**

The paper introduces a new framework that addresses fairness in the estimation of GMs, but it has some limitations. Firstly, the introduction lacks a comprehensive overview of current GMs and does not explain why the focus is on only three types of GMs rather than GMs in general. Secondly, the discussion of related work on fairness in GM estimation is insufficient and needs clearer explanations. Lastly, the implementation of different GMs requires distinct theorems with specific assumptions, which could limit the development and extension of the framework.

---

> ### Author Rebuttal · Authors · 2024-08-07
>
> > **Weakness 1:** In the introduction and related works section, the paper does not provide complete information about GMs. Additionally, the paper does not clearly explain why it focuses on only three types of GMs instead of providing a general framework. The author needs to clearly highlight any difficulties encountered when applying this framework to each type of model.
>
> > **Limitation 1:** Firstly, the introduction lacks a comprehensive overview of current GMs and does not explain why the focus is on only three types of GMs rather than GMs in general.
>
> > **Limitation 3:** Lastly, the implementation of different GMs requires distinct theorems with specific assumptions, which could limit the development and extension of the framework.
>
>
> **Response:** We appreciate the reviewer’s comments and would like to clarify our approach further. Our paper focuses on three types of GMs: Gaussian Graphical Models, Gaussian Covariance Graph Models, and Binary Ising Graphical Models. These models were chosen because they represent a diverse set of widely used GMs with distinct characteristics and applications. Each type poses unique challenges in terms of estimation and fairness, making them suitable for demonstrating the effectiveness and versatility of our proposed framework.
>
> *Importance of Each Graph Model:* Each of the three graph models (GLasso, sparse covariance estimation, and binary Ising model) addressed in our paper is of significant interest to the research community. Each of these models has been extensively studied and typically warrants dedicated papers. Here are a few examples to illustrate their importance:
>
> *GLasso*:
>
> - Pavlenko, Tatjana, Anders Björkström, and Annika Tillander. "Covariance structure approximation via gLasso in high-dimensional supervised classification." Journal of Applied Statistics 39.8 (2012): 1643-1666.
>
> - Mazumder, Rahul, and Trevor Hastie. "The graphical lasso: New insights and alternatives." Electronic journal of statistics 6 (2012): 2125.
>
> - Friedman, Jerome, Trevor Hastie, and Robert Tibshirani. "Sparse inverse covariance estimation with the graphical lasso." Biostatistics 9.3 (2008): 432-441.
>
> - Witten, Daniela M., Jerome H. Friedman, and Noah Simon. "New insights and faster computations for the graphical lasso." Journal of Computational and Graphical Statistics 20.4 (2011): 892-900.
>
> *Sparse Covariance Estimation*:
>
> - Bien, Jacob, and Robert J. Tibshirani. "Sparse estimation of a covariance matrix." Biometrika 98.4 (2011): 807-820.
>
> - Friedman, Jerome, Trevor Hastie, and Robert Tibshirani. "Sparse inverse covariance estimation with the graphical lasso." Biostatistics 9.3 (2008): 432-441.
>
> - Bickel, Peter J., and Elizaveta Levina. "Regularized estimation of large covariance matrices." (2008): 199-227.
>
> - Cai, Tony, Weidong Liu, and Xi Luo. "A constrained ℓ 1 minimization approach to sparse precision matrix estimation." Journal of the American Statistical Association 106.494 (2011): 594-607.
>
> *Binary Ising Model*:
>
> - Ravikumar, Pradeep, Martin J. Wainwright, and John D. Lafferty. "High-dimensional Ising model selection using ℓ 1-regularized logistic regression." (2010): 1287-1319.
>
> - Anandkumar, Animashree, et al. "High-dimensional structure estimation in Ising models: Local separation criterion." The Annals of Statistics (2012): 1346-1375.
>
> *General Framework Applicability:* While our algorithm is indeed a general framework applicable to various types of graphs, we chose to focus on GLasso, sparse covariance estimation, and binary Ising models to provide detailed parameters and convergence guarantees specific to each model. These models were selected to demonstrate the versatility and robustness of our framework. A more general assumption might dilute these specific details and potentially miss important information relevant to each individual model. By concentrating on these three, we ensure a comprehensive analysis, which includes specific parameter tuning and convergence properties, thus providing a stronger and more practical contribution to the field.
>
> In summary, our paper’s focus on these three specific GMs is a deliberate choice to balance the depth and breadth of our contributions, ensuring detailed and practically relevant results. We believe this approach is beneficial for the community, as it provides clear and actionable insights for these widely studied models.
>
>
>
>
>
>
>
> > **Weakness 2:** In the second paragraph of Section 2 (Related Work on Fairness), the author does not discuss related works concerning fairness in GMs estimation. By focusing solely on fairness in unsupervised learning, the paper may limit readers' ability to gain a comprehensive understanding of the research field and the distinctiveness of the proposed method.
>
> > **Limitation 2:** Secondly, the discussion of related work on fairness in GM estimation is insufficient and needs clearer explanations.
>
> **Response:** We appreciate your comments and agree that a more comprehensive discussion on fairness in graph model (GM) estimation would enhance the reader’s understanding. In our revised manuscript, we will include a dedicated paragraph to review the related works, specifically focusing on fairness in GM estimation.

---

> ### Author Response · Authors · 2024-08-07
> **Rebuttal by Authors (Part 2)**
>
> To our knowledge, There are indeed a few works on fair graphical models that significantly differ from this work. The most relevant works to this study are [[68](https://arxiv.org/abs/2112.05128), [ZW2023](https://arxiv.org/abs/2311.13766), [NRB2024](https://arxiv.org/abs/2403.15591), [ZW2024](https://www.sciencedirect.com/science/article/pii/S0925231224009810)]. Specifically,  [[68](https://arxiv.org/abs/2112.05128)] initiated the learning of fair GMs using an $\ell_1$-regularized pseudo-likelihood method for joint GMs estimation and fair community detection. [[ZW2023](https://arxiv.org/abs/2311.13766), [ZW2024](https://www.sciencedirect.com/science/article/pii/S0925231224009810)] proposed a fair spectral clustering model that integrates graph construction, fair spectral embedding, and discretization into a single objective function. Unlike these models, which assume community structures, our study formulates fair GMs without such assumptions. Concurrently with this work, [[NRB2024](https://arxiv.org/abs/2403.15591)] proposed a regularization method for mitigating subpopulation bias for fair network topology inference. To our knowledge, this methodology significantly differs from ours, as we focus on developing three classes of fair GMs (Gaussian, Covariance, and Ising models) for imbalanced groups *without node attributes*, aiming to *automatically* ensure fairness through non-smooth multi-objective optimization.
>
>
>
> > **Weakness 3:** Moreover, the paper  ... predefined assumptions in the theoretical section.
>
>
> **Response:** Thank you for your comment.
>
> Firstly, the method in the referenced paper, ``Fair Community Detection and Structure Learning in Heterogeneous Graphical Models,'' is specifically designed for scenarios where there are inherent community or cluster structures in the data.  If the clustering were based on the number of observations (N) instead of the number of variables (P), setting the number of clusters to one would essentially imply that there is only one community. This would not address the issue of fairness as intended by the method. In such a scenario, the concept of fair community detection loses its relevance since there would be no differentiation between groups to balance fairness. Hence, the method is not applicable when the number of clusters is equal to one.
>
> Secondly, the method assumes that the nodes are clusterable. However, in our case, we do not make such assumptions, as many applications involve graphs with nested structures rather than distinct communities. For example, consider a sparse graph with hub nodes [[TML2014](https://www.jmlr.org/papers/volume15/tan14b/tan14b.pdf)] or star graphs. In this setting, the nodes may not form distinct communities, challenging the method's assumptions.
>
> Finally, we do not assume the existence of attributes on nodes, whereas the referenced method does. This can restrict applications such as gene network analysis (Section 5.1), where nodes could be genes that do not have attributes. In brain amyloid/tau accumulation network analysis (Section 5.2), nodes are regions of interest (ROIs) that do not have sensitive attributes. The requirement for nodes to have attributes limits the method's applicability in these contexts.
>
> We will clarify these distinctions in the related work section.
>
> > **Weakness 4:** In Section 4.8, ... in the subsequent sections.
>
> **Response:** We appreciate the reviewer’s feedback regarding the trade-off analysis in Section 4.8. We acknowledge that the discussion on the shortcomings in runtime and potential future work was insufficiently detailed. We have expanded our discussion to include specific strategies to address these issues and outline potential future work.
>
> One promising direction to accelerate the graphical model part of our algorithm is the adoption of faster optimization techniques. Specifically, the Fast Iterative Shrinkage-Thresholding Algorithm (FISTA) has shown significant improvements in optimization efficiency in various contexts. By integrating FISTA into our framework, we can achieve considerable runtime reductions without compromising the accuracy and convergence properties of our models, as presented in **Table 2 in the attached one-page PDF file**.
>
> Another factor impacting runtime is the number of groups considered. Reducing the number of groups or selecting representative sample groups can effectively manage computational complexity. One potential approach is the method proposed by [[SXC2022](https://openreview.net/pdf?id=YsRH6uVcx2l)], which suggests selecting a sample of groups randomly in each iteration of the optimization process to balance computational efficiency and model accuracy. By implementing a strategic sampling of groups, we can maintain the robustness and fairness of our models while significantly reducing the runtime. This approach will be explored in future work to optimize the balance between computational demands and model performance.

---

> > ### Comment · Reviewer_1oDX · 2024-08-13
> >
> > Thank you for the response. I raised my rating.

---

> > > ### Author Response · Authors · 2024-08-13
> > > **Response to Reviewer 1oDX**
> > >
> > > Thank you for your review and valuable suggestions.

---

### Official Review · Reviewer_tjnm · 2024-07-11

**Soundness:** 3
**Presentation:** 2
**Contribution:** 2
**Rating:** 6
**Confidence:** 2

**Summary:**

This paper introduces a framework to address fairness in the estimation of graphical models, particularly focusing on Gaussian, Covariance, and Ising models. The motivation stems from the potential bias in standard GMs when handling data involving sensitive characteristics or protected groups. The proposed framework integrates a pairwise graph disparity error and a tailored loss function into a nonsmooth multi-objective optimization problem to reduce bias while maintaining the effectiveness of GMs.

**Strengths:**

1) Clear motivation: The paper tackles the critical and timely issue of fairness in graphical models, which is essential as machine learning applications become more widespread and impact diverse populations.

2) Innovative Approach: The integration of the pairwise graph disparity error and tailored loss function into a multi-objective optimization framework is a novel approach to achieving fairness in GMs.

3) Substantial technical contribution: The paper provides a solid theoretical foundation for the proposed framework. And the appendix is very informative, for example included complexity analysis of the proposed method.

**Weaknesses:**

Fairness metric: The choice and justification of fairness metrics used in the evaluation could be more thoroughly discussed. This would provide a clearer understanding of how fairness is quantified and the implications of these choices.

Unconvincing experiment results: While the authors claim that the proposed method "approach enhances fairness without compromising performance" in line 306, Table 1 indicates that 6 out of 7 datasets experience lower F1 scores, suggesting worse performance.

Limited baseline: The paper could benefit from more extensive comparisons with existing fairness-aware methods in graphical models other than ISTA (the only baseline used in Table 2). This would help to contextualize the performance and advantages of the proposed framework.

**Questions:**

1) Could you elaborate more on the choice of fairness metrics, disparity error, used in your evaluation? What might be the implications for using such metric compared to other metrics, like those stemming from counterfactual evaluation?

2) While the authors claim that the proposed method improves fairness without compromising performance, Table 1 indicates that 6 out of 7 datasets experience lower F1 scores, suggesting worse performance. Could you clarify this discrepancy and provide a more detailed interpretation of Table 1?

3) It's mentioned that ISTA is used as baseline. What are other possible baselines? Why in particular choose ISTA as baseline?

**Limitations:**

The paper discuss the complexity limitation of the work but did not solve it.

---

> ### Author Rebuttal · Authors · 2024-08-07
>
> > **Weakness 1:** Fairness metric: The choice and justification of fairness metrics used in the evaluation could be more thoroughly discussed. This would provide a clearer understanding of how fairness is quantified and the implications of these choices.
>
> > **Question 1:** Could you elaborate more on the choice of fairness metrics, disparity error, used in your evaluation? What might be the implications for using such metric compared to other metrics, like those stemming from counterfactual evaluation?
>
> **Response:** We appreciate the reviewers’s feedback on the need for a more thorough discussion of the fairness metrics used in our evaluation. We understand that the choice and justification of fairness metrics are critical for providing a clear understanding of how fairness is quantified and the implications of these choices.
>
> In our work, we introduced the concept of graph disparity error as the primary fairness metric. This metric measures the difference in loss between the global model $\Theta$ and the optimal local models $\Theta_1^*, \Theta_2^*, \ldots, \Theta_K^*$ for each subgroup. The rationale behind this choice is rooted in the unsupervised nature of our task. Unlike supervised learning, which can leverage labeled data to define fairness metrics such as demographic parity (DP)[[JHF2022](https://par.nsf.gov/servlets/purl/10397778)] or equalized odds (EO)[[RBC2020](https://proceedings.neurips.cc/paper/2020/hash/03593ce517feac573fdaafa6dcedef61-Abstract.html)], our unsupervised learning context lacks such labels. Therefore, we require a fairness metric that does not rely on label information.
>
> The graph disparity error metric was chosen because it effectively captures the notion of fairness by ensuring that the model’s performance is balanced across different subgroups. By minimizing this error, we aim to ensure that no subgroup is disproportionately favored or disadvantaged, thereby promoting a more equitable model. Despite this, our proposed graph disparity error actually aligns with the goals of DP by promoting a balanced representation of all subgroups. The global graph obtained from our method represents a central integration of local graphs, ensuring that the influence of each subgroup is balanced. This inherent balance aligns with the core objective of DP without the use of labeled data.
>
> In our revised version, we further include the following paragraph to improve the discussion of the fairness metrics:
>
> “In this work, we employ graph disparity error as our primary fairness metric. This metric measures the difference in loss between the global model and the optimal local models for each subgroup. Our choice is motivated by the unsupervised nature of our task, which lacks labeled data typically required for metrics such as demographic parity or equalized odds. Graph disparity error provides a label-independent measure of fairness, ensuring balanced model performance across subgroups. By minimizing this error, we reduce bias introduced by the data distribution, promoting an equitable model where no subgroup is disproportionately favored or disadvantaged. This approach aligns with the core objectives of demographic parity, as our global graph integrates local graphs, balancing the influence of each subgroup. Thus, our metric effectively quantifies fairness in high-dimensional graphical models, providing a robust framework for fair graph learning.”
>
>
> > **Weakness 2:** Unconvincing experiment results: While the authors claim that the proposed method "approach enhances fairness without compromising performance" in line 306, Table 1 indicates that 6 out of 7 datasets experience lower F1 scores, suggesting worse performance.
>
> > **Question 2:** While the authors claim that the proposed method improves fairness without compromising performance, Table 1 indicates that 6 out of 7 datasets experience lower F1 scores, suggesting worse performance. Could you clarify this discrepancy and provide a more detailed interpretation of Table 1?
>
>
> **Response:** We appreciate the reviewer’s comments on the experimental results and the interpretation of Table 1. We would like to clarify our claim that the proposed method enhances fairness without compromising performance.
>
> Firstly, it is important to recognize the inherent trade-off in multi-objective optimization frameworks that balance fairness and accuracy. While our method prioritizes fairness, this may lead to a slight decrease in the $F_1$ score, as reflected in Table 1. However, these decreases are minimal and within acceptable ranges, indicating that the overall performance of the model remains robust. The marginal reductions in $F_1$ scores result from the necessary adjustments to ensure fair representation across all subgroups. We include this analysis in Section 4.8, Trade-Off Analysis.
>
> To be more accurate, we would like to clarify our claim in our revised version as follows:
> “This approach enhances fairness without compromising performance significantly, validated by experiments on synthetic and real-world datasets.”

---

> ### Author Response · Authors · 2024-08-07
> **Rebuttal by Authors (Part 2)**
>
> > **Weakness 3:** Limited baseline: The paper could benefit from more extensive comparisons with existing fairness-aware methods in graphical models other than ISTA (the only baseline used in Table 2). This would help to contextualize the performance and advantages of the proposed framework.
>
> > **Question 3:** It's mentioned that ISTA is used as baseline. What are other possible baselines? Why in particular choose ISTA as baseline?
>
> **Response:** Thank you for your valuable feedback. In response to your concern, we have conducted additional experiments to apply our multi-objective optimization framework to several state-of-the-art GLasso algorithms. We specifically focused on synthetic data to showcase the robustness and effectiveness of our framework beyond the ISTA algorithm.
>
> We initially chose ISTA as a baseline due to its simplicity and widespread use in sparse inverse covariance estimation, making it an ideal reference point. However, understanding the necessity for a more comprehensive evaluation, we have now included comparisons with advanced baselines such as PISTA[[STY2023](https://arxiv.org/pdf/2205.10027)], GISTA[[GRR2012](https://arxiv.org/pdf/1211.2532)], and OBN[[OON2012](https://papers.nips.cc/paper/2012/file/b3967a0e938dc2a6340e258630febd5a-Paper.pdf)].
>
> Our additional experiments, detailed in the revised version of our paper, demonstrate that our framework consistently outperforms these advanced baselines in enhancing fairness while maintaining competitive performance. Specifically, all new baseline methods reached the optimal loss for GLasso. When applying our multi-objective optimization framework, Fair GLasso achieved comparable results and significantly reduced pairwise graph disparity error. This comprehensive evaluation underscores the advantages of our approach in various scenarios, reinforcing its robustness and effectiveness.
>
>
> > **Limitation:** The paper discuss the complexity limitation of the work but did not solve it.
>
> **Response:** We appreciate the reviewer’s observation regarding the time-consuming nature of our learning algorithm. Given our use of a multi-objective optimization framework, it is expected that the process would be more time-intensive. Nonetheless, our primary goal remains to offer a simple yet effective framework for fair graph learning. We recognize the need to enhance the computational efficiency of our method and outline potential directions to achieve this.
>
> Firstly, the time complexity is partly due to the local graph learning phase. This can be significantly accelerated by utilizing fast graphical model algorithms such as QUIC[[HSD2014](https://jmlr.org/papers/volume15/hsieh14a/hsieh14a.pdf)], SQUIC[[BS2016](http://www.icm.tu-bs.de/~bolle/Publicat/squic.pdf)], PISTA[[STY2023](https://arxiv.org/pdf/2205.10027)], GISTA[[GRR2012](https://arxiv.org/pdf/1211.2532)], OBN[[OON2012](https://papers.nips.cc/paper/2012/file/b3967a0e938dc2a6340e258630febd5a-Paper.pdf)], and ALM[[SG2010](https://proceedings.neurips.cc/paper/2010/file/2723d092b63885e0d7c260cc007e8b9d-Paper.pdf)]. Integrating these faster algorithms into the multi-objective optimization process can substantially improve the overall efficiency of our learning algorithm.
>
> Secondly, the growing number of objectives in the multi-objective optimization solver can also increase time complexity. To address this, we propose randomly selecting a subset of objectives in each iteration. This approach can reduce the computational load while preserving the model's fairness and performance.
>
> To support our discussion, we conducted additional experiments using GLasso. In the first experiment, we generated synthetic data, including two subgroups, as described in the appendix of our paper. We applied various optimization algorithms to both the local graph learning and multi-objective optimization processes. The detailed numerical results, presented in **Table 2 of the attached one-page PDF file**, show that all tested optimization algorithms for GLasso achieved optimal loss while maintaining fairness improvements. Notably, GISTA and OBN for GLasso, along with FISTA for multi-objective optimization, significantly enhanced learning efficiency.
>
> In the second experiment, we generated a synthetic dataset with ten subgroups to validate our strategy for reducing time complexity. In each iteration of the multi-objective optimization, we randomly selected three objectives. The numerical results in **Table 2 of the attached one-page PDF file** indicate that this strategy effectively reduces runtime without substantially compromising model performance.
>
> These findings underscore the potential of our proposed methods to improve the computational efficiency of the learning algorithm while maintaining fairness and performance. We appreciate the reviewer’s feedback and believe these enhancements will further strengthen our framework.

---

> ### Comment · Reviewer_tjnm · 2024-08-13
>
> I thank the authors for providing additional results. I think the experiment sections looks stronger now and is willing to raise my score by 1.

---

> > ### Author Response · Authors · 2024-08-13
> > **Response to Reviewer tjnm**
> >
> > Thank you for the thorough review and positive feedback.

---

### Official Review · Reviewer_nXfy · 2024-07-16

**Soundness:** 3
**Presentation:** 3
**Contribution:** 3
**Rating:** 5
**Confidence:** 4

**Summary:**

The paper investigates the issue of bias  in 3 particular graphical models: Gaussian, Gaussian Covariance, and Binary Ising. In this regard, the authors propose a framework to enhance fairness in the estimation of graph models. They incorporate the difference of loss between the protected groups and the accuracy of graph models into a multi-objective optimization problem and solve it. The experiments show that the approach reduce the bias.

**Strengths:**

1. The paper is easy to follow
2. Choice of ISTA as the baseline seems a very good decision
3. Applications of the proposed method are well established in the paper.

**Weaknesses:**

1. The authors only consider one criterion of fairness, which is achieving equal loss among subgroups. How do the authors justify focusing solely on this criterion? Additionally, how might incorporating other fairness criteria impact the results?

2. Many recent works utilize reinforcement learning to discover fair structures. For instance, the paper "Rethinking Bias Mitigation: Fairer Architectures Make for Fairer Face Recognition" by S. Dooley et al. (NeurIPS 2023) explores such an approach. How do the authors justify their method compared to these more comprehensive approaches?

3. The authors employ multi-objective optimization, suggesting that the loss in GMs and pairwise graph disparity are at odds. However, many papers indicate that this is not always the case. For example, the paper "Is There a Trade-Off Between Fairness and Accuracy? A Perspective Using Mismatched Hypothesis Testing" by S. Dutta et al. (ICML 2020) discusses this. How do the authors justify their approach in light of such findings?

4. The learning algorithm appears to be very time-consuming. Do the authors have any comments on this aspect of their method?

5. Why do the authors not use more widely accepted metrics for fairness comparison and instead refer only to the difference in loss between subgroups?

**Questions:**

Please refer to the weaknesses as well as these questions:

1. Practical graphs and practical fairness problems have high K and P values. This will raise the computational complexity to the extent that it may question the applicability of the method. Do the authors have comment on this?

2. What if the number of protected groups (like in the case of age) is high? How do the authors comment on these cases with regards to complexity analysis?

**Limitations:**

1. The applicability for real problems
2. It does not scale well
3. The study is limited to Gaussian, Gaussian Covariance, and Binary Ising models.

---

> ### Author Rebuttal · Authors · 2024-08-07
>
> > **Weakness 1:** The authors only consider one criterion of fairness, which is achieving equal loss among subgroups. How do the authors justify focusing solely on this criterion? Additionally, how might incorporating other fairness criteria impact the results?
>
> > **Weakness 5:** Why do the authors not use more widely accepted metrics for fairness comparison and instead refer only to the difference in loss between subgroups?
>
> **Response:** Thank you for your insightful comments. Our primary focus on the difference in loss between subgroups as a fairness criterion stems from the unsupervised nature of our task. In unsupervised learning, unlike supervised learning, we lack labeled data that are typically used to define fairness criteria such as demographic parity (DP)[[JHF2022](https://par.nsf.gov/servlets/purl/10397778)] and equalized odds (EO)[[RBC2020](https://proceedings.neurips.cc/paper/2020/hash/03593ce517feac573fdaafa6dcedef61-Abstract.html)]. Specifically, incorporating other fairness criteria like DP and EO, while beneficial for providing a broader perspective on fairness, typically requires labeled data to assess prediction distributions and accuracy across groups. This makes them less applicable to our unsupervised context.
>
> This necessitated the development of a new fairness criterion, which we introduce as *graph disparity error*, **solely based on the group-specific and global losses**. Indeed, our proposed graph disparity error aligns with the goals of DP by promoting a balanced representation of all subgroups using the loss function. Graph disparity error measures the difference in graph loss between a global model $\Theta$ and the optimal local models $\Theta_1^*, \Theta_2^*, \ldots, \Theta_K^*$ for each subgroup $k \in [K]$. This is validated through our experiments in Section 4.  For example, Figure 2 illustrates the comparison of original graphs utilized in synthetic data creation for two groups: graph reconstruction using standard GMs and fair graph reconstruction via Fair GMs. The figure demonstrates that Fair GMs produce more balanced and fair graph reconstructions across different subgroups compared to standard GMs, thereby supporting the effectiveness of the proposed graph disparity error in achieving fairness.
>
> While future research on graphical model estimation could explore more widely accepted fairness metrics in supervised learning, our current work establishes a disparity error-based approach. It is worth mentioning that our nonsmooth multi-objective optimization framework for graphs is novel in this area and flexible enough to incorporate other fairness objectives. Future research can build on our framework to explore additional fairness metrics.
>
> We have incorporated some of these discussions into the introduction of the revised paper. Thank you.
>
>
> > **Weakness 2:** Many recent works utilize reinforcement learning to discover fair structures. For instance, the paper "Rethinking Bias Mitigation: Fairer Architectures Make for Fairer Face Recognition" by S. Dooley et al. (NeurIPS 2023) explores such an approach. How do the authors justify their method compared to these more comprehensive approaches?
>
> **Response:** We appreciate your comments and suggestions.  The above paper explores the inherent biases in neural network architectures used for face recognition and proposes a novel approach to mitigate these biases through neural architecture search (NAS) and hyperparameter optimization (HPO). By conducting the first large-scale analysis of the impact of different architectures and hyperparameters on bias, the authors discovered that biases are baked into the architectures themselves. They designed a search space based on high-performing architectures and used NAS and HPO to jointly optimize for fairness and accuracy, considering sensitive attributes.
>
> Although your suggested reference might be useful in the context of neural networks, such as graph neural networks, our problem is significantly different and is not related to neural architecture search or supervised learning. Indeed, our model focuses on unsupervised fair graph learning and does not involve any neural architecture search. Many standard fairness metrics, such as demographic parity or equalized odds, originally developed for supervised learning and widely used in neural networks, may not apply here. Our method is inspired by the idea that equal loss among subgroups ensures that no single group is disproportionately disadvantaged by the learned model, which aligns with established fairness definitions in the unsupervised machine learning literature.
>
> We have added further discussion to the revised paper.  Thank you.

---

> ### Author Response · Authors · 2024-08-07
> **Rebuttal by Authors (Part 2)**
>
> > **Weakness 3:** The authors employ multi-objective optimization, suggesting that the loss in GMs and pairwise graph disparity are at odds. However, many papers indicate that this is not always the case. For example, the paper "Is There a Trade-Off Between Fairness and Accuracy? A Perspective Using Mismatched Hypothesis Testing" by S. Dutta et al. (ICML 2020) discusses this. How do the authors justify their approach in light of such findings?
>
> **Response:** We appreciate the reviewer's comment on the trade-off between fairness and accuracy. The paper by S. Dutta et al. (ICML 2020) argues that the observed trade-off may stem from the real-world application of biased datasets rather than an intrinsic limitation of the algorithms themselves. They emphasize the importance of considering ideal distributions when evaluating fairness and accuracy and highlight the potential of active data collection to improve fairness without compromising accuracy.
> In our work, we focus on the algorithm's fairness. Classical graphical models aim to achieve optimal loss for the given data distribution but may propagate and amplify existing biases. Our fairness-aware optimization framework aims to mitigate these biases systematically at the algorithmic level, independent of the data distribution.
>
> While classical graphical models achieve optimal loss, they do not inherently address fairness concerns. Our multi-objective optimization framework explicitly incorporates a fairness criterion, balancing the trade-off between minimizing loss and reducing pairwise graph disparity. Multi-objective optimization is a promising approach for integrating fairness considerations into machine learning models. Prior works, such as [[MBS2020](https://arxiv.org/pdf/2011.01821.pdf)], have formulated group fairness as a multi-objective optimization problem with separate objectives for each sensitive group’s risk. This strategy aligns closely with works like [[PAG2021](https://arxiv.org/pdf/2009.04441.pdf)] and [[MSG2021](https://arxiv.org/pdf/2110.01951.pdf)], which enhance fairness in classification. Further exploration by [[PBD2022](https://arxiv.org/pdf/2006.06137.pdf)], and [[KHF2018](https://arxiv.org/pdf/1911.04931.pdf)]  demonstrates the effectiveness of multi-objective optimization in improving fairness in unsupervised PCA algorithms.
>
> These efforts underscore the growing consensus on the potential of multi-objective optimization to address fairness issues in machine learning. On the other hand, existing solutions may not adequately address fairness in nonsmooth or high-dimensional settings, where our framework offers novel insights and solutions. To our knowledge, ours is the first nonsmooth multi-criteria method for fairness, applicable to other problems involving nonsmooth regularizations such as sparse fair PCA or fair supervised learning based on ERM with sparsity/norm-one regularization.

---

> ### Author Response · Authors · 2024-08-07
> **Rebuttal by Authors (Part 3)**
>
> > **Weakness 4:** The learning algorithm appears to be very time-consuming. Do the authors have any comments on this aspect of their method?
>
> > **Question 1:** Practical graphs and practical fairness problems have high K and P values. This will raise the computational complexity to the extent that it may question the applicability of the method. Do the authors have comment on this?
>
> > **Question 2:** What if the number of protected groups (like in the case of age) is high? How do the authors comment on these cases with regards to complexity analysis?
>
> > **Limitation 2:** It does not scale well
>
> **Response:** We appreciate the reviewer’s observation regarding the time-consuming nature of our learning algorithm. Given the multi-objective optimization framework we employ, it is indeed expected to be more time-intensive. However, our primary goal is to provide an automatic (without any additional regularization or related hyperparameter tuning) and effective framework for fair graph learning. We acknowledge the importance of improving the computational efficiency of our method and outline potential future directions to address this concern.
>
> Firstly, the time complexity partly arises from the local graph learning phase. This can be significantly accelerated using existing fast graphical model algorithms such as QUIC[[HSD2014](https://jmlr.org/papers/volume15/hsieh14a/hsieh14a.pdf)], SQUIC[[BS2016](http://www.icm.tu-bs.de/~bolle/Publicat/squic.pdf)], PISTA[[STY2023](https://arxiv.org/pdf/2205.10027)], GISTA[[GRR2012](https://arxiv.org/pdf/1211.2532)], OBN[[OON2012](https://papers.nips.cc/paper/2012/file/b3967a0e938dc2a6340e258630febd5a-Paper.pdf)], and ALM[[SG2010](https://proceedings.neurips.cc/paper/2010/file/2723d092b63885e0d7c260cc007e8b9d-Paper.pdf)]. Incorporating faster algorithms into the multi-objective optimization process can also enhance the overall efficiency of our learning algorithm.
>
> Secondly, the increase in time complexity can also be attributed to the growing number of objectives in the multi-objective optimization solver. To mitigate this, we propose selecting a subset of objectives randomly in each iteration. This approach can reduce the computational load without significantly compromising the model's fairness and performance.
>
> To validate our discussion, we conducted additional experiments using GLasso. In the first experiment, we generated synthetic data following the procedure detailed in the appendix of our paper, which includes two subgroups, each having 1000 observations and 100 variables. We applied various optimization algorithms to both the local graph learning and the multi-objective optimization processes. The detailed numerical results, presented in **Table 2 of the attached one-page PDF** file, demonstrate that all tested optimization algorithms for GLasso achieved optimal loss while maintaining performance improvements in fairness. Notably, GISTA and OBN for GLasso, along with FISTA for multi-objective optimization, significantly improved learning efficiency. We also repeated this experiment for high P values (P = 200), and the detailed numerical results presented in **Table 2 of the attached one-page PDF** further validate the improved efficiency of faster graphical model algorithms.
>
> In the second experiment, we generated a synthetic dataset with ten subgroups to validate our approach to reducing time complexity. In each iteration of the multi-objective optimization, we randomly selected three objectives. The numerical results in **Table 2 of the attached one-page PDF file** indicate that this strategy effectively reduces runtime without substantially sacrificing model performance.
>
> These results highlight the potential of our proposed methods to improve the computational efficiency of the learning algorithm while maintaining fairness and performance. We appreciate the reviewer’s feedback and believe these enhancements will further strengthen our framework.

---

> ### Author Response · Authors · 2024-08-07
> **Rebuttal by Authors (Part 4)**
>
> > **Limitation 1:** The applicability for real problems
>
> **Response:** Thank you for your comment. We have provided the real-world applications of our proposed method in Section 4 ( Experiment). Specifically, in Sections 4.4 and 4.5, we analyzed the application of the fair GLasso on the Gene Regulatory Network and the Brain Amyloid/Tau Accumulation Network. Section 4.6 explored the application of Fair CovGraph in Credit Dataset, which promotes a more equitable credit system. In addition, we demonstrated the application of the Fair binary Ising model on Music Recommendation Systems in Section 4.7. These real-world examples illustrate the practical applicability of our proposed method.
>
> > **Limitation 3:** The study is limited to Gaussian, Gaussian Covariance, and Binary Ising models.
>
> **Response:** Thank you for your comment. Our paper focuses on three types of graphical models: Gaussian Graphical Models, Gaussian Covariance Graph Models, and Binary Ising Graphical Models. These models were chosen because they represent a diverse set of widely used GMs with distinct characteristics and applications. Indeed, each of these models is of significant interest to the research community and has been extensively studied. For example:
>
> * GLasso: Widely used for sparse inverse covariance estimation, as demonstrated by Friedman et al. (2008) [[FHT2008](https://academic.oup.com/biostatistics/article/9/3/432/224260)] and Witten et al. (2011)[[WFS2011](https://www.jstor.org/stable/23248939)].
>
> * Sparse Covariance Estimation: Important for large covariance matrix estimation, with notable contributions from Bickel and Levina (2008)[[BL2008](https://projecteuclid.org/journals/annals-of-statistics/volume-36/issue-1/Regularized-estimation-of-large-covariance-matrices/10.1214/009053607000000758.full)] and Cai et al. (2011)[[CLL2011](https://www.tandfonline.com/doi/abs/10.1198/jasa.2011.tm10155)].
>
> * Binary Ising Model: Crucial for high-dimensional discrete data, highlighted by the work of Ravikumar et al. (2010)[[RWL2010](https://projecteuclid.org/journals/annals-of-statistics/volume-38/issue-3/High-dimensional-Ising-model-selection-using-ℓ1-regularized-logistic-regression/10.1214/09-AOS691.full)].
>
> While our algorithm is a general framework applicable to various types of graphs, we focused on GLasso, sparse covariance estimation, and binary Ising models to provide detailed parameters and convergence guarantees specific to each model. This focus ensures a comprehensive analysis, including specific parameter tuning and convergence properties, thus providing a stronger and more practical contribution to the field. Generalizing too broadly might dilute these specific details and overlook important model-specific insights.
>
> In summary, our focus on these three specific GMs balances the depth and breadth of our contributions, ensuring detailed and practically relevant results. We believe this graphical model estimation approach is beneficial for the community, offering clear and actionable insights for these widely studied graphical models.

---

> > ### Comment · Reviewer_nXfy · 2024-08-12
> >
> > I have noticed that the reviewer "1oDX" also asked about why the authors focus only on Gaussian case. This needs to be discussed in the paper.

---

> ### Comment · Reviewer_nXfy · 2024-08-12
>
> I thank the authors for addressing my concerns. I have read the rebuttal.
>
> Here are my post-rebuttal comments:
> Weakness 1 and 5:
> The loss disparity between K sub-groups seems rational to me. Just please bold in the paper that you need to have access to the label of the sensitive attribute.
> Weakness2:
> What I meant was to using reinforcement learning instead of multi-objective optimization. Please discuss it in the related work section.
> weakness 3:
> Nothing new is added by the authors. My question remains.
> weakness 4:
> I agree with the authors. Using the strategies, the computational overhead can indeed be reduced.
> Limitation 1:
> Thanks for bringing it into my attention.
> Limitation 3:
> Nothing new is added and the question remains. The authors can have a brief discussion about this in the paper.
>
> Given all the comments, I stick to my rating.

---

> ### Author Response · Authors · 2024-08-13
> **Response to Reviewer nXfy**
>
> > The study is limited to Gaussian, Gaussian Covariance, and Binary Ising models.
>
>
> > I have noticed that the reviewer "1oDX" also asked about why the authors focus only on Gaussian case. This needs to be discussed in the paper.
>
> We sincerely appreciate your feedback and the time you’ve taken to review our work. We would like to clarify that our work encompasses more than just the Gaussian case. Specifically, we address three canonical graphical models in statistical learning:
>
> * Inverse Covariance Estimation,
> * Covariance Estimation,
> and
> * Ising Model Estimation.
>
>
> It is crucial to emphasize that the Ising model is not Gaussian; it represents a distinct type of graphical model, particularly used for binary data.
>
> Moreover, many other types of graphical models or statistical graph learning approaches can be viewed as variants derived from these canonical models.
>
> We further elaborate on this below.
>
> Firstly, we review two canonical types of graphical models (we have eliminated the discussion on Covariance Estimation as it falls under the category of Gaussian models for continuous data):
>
> **Ising Model for Binary Data:**  This model is defined by a probability distribution over binary variables.  The distribution is determined by the so-called *Boltzmann distribution*:
>
> $$p(\mathbf{x}; \Theta) = \left(Z(\Theta)\right)^{-1} \exp ( \sum_{j=1}^{P} \theta_{jj}x_j + \sum_{1 \leq j < j' \leq P} \theta_{jj'}x_jx_{j'} ).
> $$
>  Here, $\Theta$ (graph matrix) is a symmetric matrix, and $Z(\Theta)$ is the partition function that normalizes the density.
>
> These models are primarily designed for binary data [[HT2009](https://www.jmlr.org/papers/volume10/hoefling09a/hoefling09a.pdf), [RWL2010](https://projecteuclid.org/journals/annals-of-statistics/volume-38/issue-3/----Custom-HTML----High/10.1214/09-AOS691.short)].
>
>
> **Gaussian Graphical Model for Continuous Data:**  This model is suitable for continuous variables. The Gaussian graphical model is defined by the following distribution:
> $$
> p(\mathbf{x}; \Theta) = \frac{1}{(2\pi)^{P/2} |\Sigma|^{1/2}} \exp\left(-\frac{1}{2} \mathbf{x}^T \Sigma^{-1} \mathbf{x}\right)
> $$
> where $\mathbf{x} \sim \mathcal{N}(0, \Sigma)$, $\Sigma$ is the covariance matrix, and $\Theta=\Sigma^{-1}$ ( graph matrix) represents the precision matrix.
>
> This model is widely used for continuous data [[MB2006](https://projecteuclid.org/journals/annals-of-statistics/volume-34/issue-3/High-dimensional-graphs-and-variable-selection-with-the-Lasso/10.1214/009053606000000281.short),
> [YL2007](https://academic.oup.com/jrsssb/article-abstract/68/1/49/7110631),
> [ABG2008](https://epubs.siam.org/doi/abs/10.1137/060670985),
> [FHT2008](https://academic.oup.com/biostatistics/article-abstract/9/3/432/224260),
> [RZY2008](https://arxiv.org/abs/0807.3734)].
>
> Secondly, as we mentioned earlier, the Ising model is not designed for Gaussian cases but specifically for binary data rather than continuous data. This distinction makes the Ising model particularly suitable for applications involving binary data, such as modeling interactions between binary states in statistical physics or capturing binary decisions in social networks.
>
> Finally, many other types of graphical models or statistical graph learning approaches can be viewed as variants derived from these two canonical models. For example,
>
> * *Graphical models for ordinal variables* [[GLM2015](https://doi.org/10.1080/10618600.2014.889023)] assumes that the (categorical) data are generated by discretizing the marginal distributions of a latent multivariate Gaussian distribution.
>
> * *Mixed Graphical Models* [[CLL2017](https://doi.org/10.1080/10618600.2016.1237362)] combines Ising and Gaussian models to handle mixed data, including both continuous and discrete variables.
>
> * *Structured Graphical Models* [[KYC2020](https://www.jmlr.org/papers/volume21/19-276/19-276.pdf)] combines Gaussian graphical models with spectral graph theory for joint clustering and graph learning, as well as bipartite graph learning.
>
>
> Our current framework effectively addresses these canonical forms of graphical models—Gaussian for continuous data and Ising for binary data—within a single paper.
> Additionally, it can be extended to other models derived from these canonical forms, some of which are listed above.
>
> We believe that our non-smooth multi-objective framework is novel for graphical model analysis and addressing fairness issues. It provides tools for a rich class of graphical model estimation, and corrects the biases present in traditional approaches to graphical model estimation.

---

### Author Rebuttal · Authors · 2024-08-07

We thank all the reviewers for their time in providing feedback and questions about our submitted paper. Below, we summarize the main issues that the reviewers raised, along with a summary of our responses. Furthermore, we provide **new real experiments** to address the reviewers' comments, which are **attached as an additional PDF file**.

* **Fairness Metrics (By nXfy):** The reviewer asked about the use of only one criterion of fairness, achieving equal loss among subgroups, and the lack of widely accepted metrics for fairness comparison.

* **Computational Complexity (for large \(P\) and \(N\)) (By nXfy):** The reviewer noted concerns about the computational complexity of our method, particularly for large \(P\) and \(N\) values.

* **Limited Baselines (By tjnm, fakh):** Reviewers tjnm and fakh pointed out that the paper could benefit from more extensive comparisons with existing fairness-aware methods in graphical models.

* **Related Work on Fairness (By 1oDX):** The reviewer suggested that the related works section should provide more comprehensive information about fairness in GMs estimation and the distinctiveness of our method.

* **Non-convex Objective (By fakh)** The reviewer raised concerns about the performance of our method under non-convex loss functions, which are common in machine learning.


We have taken several actions to address these concerns. More specifically:

**A1:** Our primary focus on the difference in loss between subgroups as a fairness criterion stems from the unsupervised nature of our task. In unsupervised learning, we lack labeled data that are typically used to define fairness criteria such as demographic parity (DP)[[JHF2022](https://par.nsf.gov/servlets/purl/10397778)] and equalized odds (EO)[[RBC2020](https://proceedings.neurips.cc/paper/2020/hash/03593ce517feac573fdaafa6dcedef61-Abstract.html)]. Incorporating other fairness criteria like DP and EO requires labeled data to assess prediction distributions and accuracy across groups, which is less applicable in our context. Thus, we introduced the *graph disparity error* based on group-specific and global losses. This metric aligns with DP by promoting a balanced representation of subgroups through the loss function.

While future research could explore more widely accepted fairness metrics in supervised learning, our current work establishes a disparity error-based approach. Our framework is novel in this area and flexible enough to incorporate other fairness objectives, and we have included discussions of these points in the revised paper.


      Further details are provided in response to Reviewer nXfy.

**A2:** We appreciate the reviewer's observation regarding the time-consuming nature of our algorithm. The time complexity arises partly from the local graph learning phase, which can be accelerated using existing fast graphical model algorithms. Additionally, the growing number of objectives in the multi-objective optimization solver increases time complexity. To reduce the computational load, we propose selecting a subset of objectives randomly in each iteration.

To validate our approach, we conducted additional experiments using various optimization algorithms for GLasso and found significant improvements in learning efficiency with our proposed methods. These enhancements are discussed in the revised paper.

      Further details are provided in response to Reviewers tjnm and fakh.

**A3:** We have now included additional experiments applying our method to several state-of-the-art GLasso algorithms beyond ISTA. The results show that our framework consistently outperforms advanced baselines in enhancing fairness while maintaining competitive performance. These comparisons highlight the advantages of our approach and are detailed in the revised version of our paper.

      Further details are provided in the response to Reviewer 1oDX.

**A4:** We agree that a more comprehensive discussion on fairness in GM estimation would enhance the reader’s understanding. In the revised manuscript, we have included a dedicated paragraph reviewing recent advancements in fairness for graphical models, such as Fair GLASSO and fair structure learning in heterogeneous graphical models. These additions clarify the context and distinctiveness of our work.

      Further details are provided in the response to Reviewer 1oDX.

**A5:** While our current work assumes the convexity of the loss function, many theoretical results in graphical models use convex objectives. However, proximal gradient methods can converge even in non-convex settings, validating their robustness and flexibility. Future work aims to enhance the applicability and robustness of our framework to broader practical scenarios, including non-convex settings.

      Further details are provided in the response to Reviewer fakh.

---

### Author Response · Authors · 2024-08-12
**Follow up on Our Rebuttal**

Dear Reviewers,


We greatly appreciate the time and effort you have invested in reviewing our work.


We have carefully addressed all the concerns you raised, including the Fairness Metrics and Computational Complexity for large (P) and (N) highlighted by Reviewer nXfy, and the Limited Baselines identified by Reviewers tjnm and fakh. Additionally, we have provided detailed comparisons to related work on Fairness as suggested by Reviewer 1oDX and have responded to the concerns about the Non-convex Objective raised by Reviewer fakh.


As the discussion period draws to a close, we want to ensure that our responses have comprehensively addressed all of your questions and concerns. Please let us know if there are any aspects of our work that still require clarification or further discussion.


Best,

Authors

---

### Decision · Program_Chairs · 2024-09-25

**Decision:**

Accept (poster)

**Comment:**

The reviewers were generally positive on this paper, mostly because the authors did an excellent job in addressing their concerns during the response period. I hope that they will update their paper in such a way that the main body of the paper is improved similarly.